# DELAY FLOW MATCHING

**Bolin Zhao[1,2]∗, Xiaoyu Zhang[2,3]∗, Yuting Dong[1,2], Xin Lu[3]†, Wei Lin[1,2,4,5]†, Qunxi Zhu[2,4,5]†**

[1]School of Mathematical Sciences, SCMS, SCAM, and CCSB, Fudan University, China
[2]Research Institute of Intelligent Complex Systems, Fudan University, China
[3]College of Systems Engineering, National University of Defense Technology, China
[4]State Key Laboratory of Medical Neurobiology and MOE Frontiers Center for Brain Science, Institutes of Brain Science, Fudan University, China
[5]Shanghai Artificial Intelligence Laboratory, China.

{blzhao21, ytdong22}@m.fudan.edu.cn  {wlin, qxzhu}@fudan.edu.cn
{xyzhangphd, xin.lu.lab}@outlook.com

## ABSTRACT

Flow matching (FM) based on Ordinary Differential Equations (ODEs) has achieved significant success in generative tasks. However, it faces several inherent limitations, including an inability to model trajectory intersections, capture delay dynamics, and handle transfer between heterogeneous distributions. These limitations often result in a significant mismatch between the modeled transfer process and real-world phenomena, particularly when key coupling or inherent structural information between distributions must be preserved. To address these issues, we propose Delay Flow Matching (DFM), a new FM framework based on Delay Differential Equations (DDEs). Theoretically, we show that DFM possesses *universal approximation capability* for continuous transfer maps. By incorporating delay terms into the vector field, DFM enables trajectory intersections and better captures delay dynamics. Moreover, by designing appropriate initial functions, DFM ensures accurate transfer between heterogeneous distributions. Consequently, our framework preserves essential coupling relationships and achieves more flexible distribution transfer strategies. We validate DFM's effectiveness across synthetic datasets, single-cell data, and image-generation tasks.

## 1 INTRODUCTION

Generative modeling is a key and rapidly advancing field in machine learning, focusing on learning transformations between different distributions. It underpins a wide range of applications across many tasks in diverse fields, including image generation (Nichol et al., 2021), molecule design (Sanchez-Lengeling & Aspuru-Guzik, 2018), and single-cell trajectory inference (Saelens et al., 2019). Traditional approaches, such as Variational Autoencoders (Kingma, 2013; Rezende et al., 2014), Generative Adversarial Networks (Goodfellow et al., 2014), and Normalizing Flows (Dinh et al., 2014; 2022; Papamakarios et al., 2021), have achieved notable success, yet they still face challenges like instability and computational inefficiency.

Recently, diffusion models (Sohl-Dickstein et al., 2015; Song & Ermon, 2019; Ho et al., 2020; Song et al., 2020a; Yang et al., 2023) have garnered considerable attention for their ability to model complex data distributions via a diffusion process. These models incrementally add noise to data in a forward process and then learn to reverse this process in order to regenerate data samples, which can be understood as learning stochastic dynamics that interpolate between the prior distribution and the target data distribution (Song et al., 2020b; 2021). Additionally, Flow Matching (FM), a simulation-free method for training continuous normalizing flows (Chen et al., 2018), is proposed to model the transformation between distributions as the flow maps of Neural ODEs (Chen et al., 2018; Grathwohl et al., 2018). FM achieves efficient training by directly regressing on an explicitly

---

∗Equal contribution.
†Corresponding authors.

constructed conditional vector field. Concurrently, the stochastic interpolant (Albergo & Vanden-Eijnden, 2022) and rectified flow (Liu et al., 2023) are introduced, both of which employ flow maps for matching distributions, though from distinct conceptual frameworks.

As discussed above, most existing continuous generative models for distribution transfer rely on ODEs or SDEs. However, ODE-based models have limited representational capacity, restricting their ability to capture a broad range of distribution transfer strategies (Dupont et al., 2019). In contrast, Delay Differential Equations (DDEs) have been widely adopted to model various real-world systems, including neural dynamics (Campbell, 2007), electro-optical systems (Chembo Kouomou et al., 2005), population dynamics (Lotka, 1925), and many biological network motifs (Glass et al., 2021), where delayed feedback mechanisms naturally give rise to DDE-based formulations. To address the limitations of ODEs in dynamic modeling, Neural DDEs (Zhu et al., 2020) and their variants (Ji & Orosz, 2024; Zhu et al., 2022) are proposed, which explicitly incorporate the effects of historical states and thus exhibit a significantly enhanced representational capacity. Despite these advancements, however, there is currently no approach that utilizes the probability flows of DDEs for distribution transport.

**Contributions**. We present Delay Flow Matching (DFM), a generalized framework that enables more flexible and precise distribution transport strategies using DDEs. The key contributions of this study are summarized as follows:

1. **Development of DFM**: We introduce DFM, a novel generative model framework based on Neural DDEs, which overcomes the inherent limitations of ODE-based models by incorporating delay terms in the vector field and designing appropriate initial functions. DFM can model a broader range of transport strategies, including those with trajectory intersections, and achieves more precise transport between heterogeneous distributions. It also naturally adapts to the probability flow generated by delay dynamical systems.

2. **Theoretical insights**: We rigorously prove that DFM can universally approximate any continuous transport map between source and target distributions. In contrast, ODE-based models cannot represent certain simple transport maps, such as those involving trajectory intersections, and cannot achieve exact transport between heterogeneous distributions.

3. **Integration with advanced techniques**: DFM can be seamlessly integrated with existing methods, such as keypoint-guided optimal transport, enabling more effective alignment with known coupling information during transport.

4. **Empirical validation**: We validate DFM's effectiveness on both synthetic and real-world datasets. DFM accurately recovers underlying delay dynamics from snapshot data, significantly outperforms existing methods in single-cell trajectory inference, and surpasses ODE-based FM in image generation tasks.

## 2 PRELIMINARIES

### 2.1 DDEs AND PROBABILITY FLOWS

DDEs are widely used to model systems in which the time evolution depends not only on the current state but also on past states, such as those governed by delayed feedback. For time-dependent DDEs with a single delay term, the general formulation is given by:

$$\frac{\mathrm{d}\boldsymbol{x}(t)}{\mathrm{d}t} = \boldsymbol{u}[t, \boldsymbol{x}(t), \boldsymbol{x}(t-\tau)], t \in [0, T], \tag{1}$$
$$\boldsymbol{x}(h) = \boldsymbol{\psi}(h), h \in [-\tau, 0],$$

where $\boldsymbol{u}[t, \boldsymbol{x}(t), \boldsymbol{x}(t-\tau)] : [0, T] \times \mathbb{R}^d \times \mathbb{R}^d \to \mathbb{R}^d$ is a smooth vector field which is abbreviated as $\boldsymbol{u}(t, \boldsymbol{x}, \boldsymbol{x}_\tau)$ in the following, $\boldsymbol{\psi}(h)$ represents the continuous initial function. We further denote $\boldsymbol{\psi}(h; \boldsymbol{x}_0)$ as the initial function that takes the value $\boldsymbol{x}_0$ at time $t = 0$, i.e. $\boldsymbol{\psi}(0; \boldsymbol{x}_0) = \boldsymbol{x}_0$.

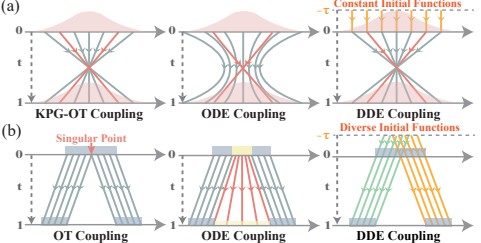

Figure 1: Comparison of ODE and DDE coupling. (a) A Gaussian distribution is mapped to itself via KPG-OT, where the red lines and arrows denote the prescribed source-target keypoint pairs, enforcing $x \to -x$. The resulting trajectory intersections cannot be realized by ODE flows, whereas the DDE $\dot{x} = -2x(t-1)$ with constant initial functions achieves exact coupling. (b) The transport of $\mathcal{U}(-2, 2)$ to $\frac{1}{2}\mathcal{U}(-3, -1) + \frac{1}{2}\mathcal{U}(1, 3)$ violates connectivity preservation of ODE flows, forcing mass near 0 into $(-1, 1)$. DDEs achieve precise transport by assigning distinct initial functions.

For a given initial distribution $\boldsymbol{x}_0 \sim p_0(\boldsymbol{x}_0)$ and the corresponding initial functions $\boldsymbol{\psi}(t; \boldsymbol{x}_0)$, the above DDE (1) induces the associated probability flows $p(\boldsymbol{x}, t|\boldsymbol{\psi}) : \mathbb{R}^d \times [0, T] \to \mathbb{R}^+$, satisfying the delay Fokker-Planck equation (refer to Appendix B) (Guillouzic et al., 1999; Frank, 2005):

$$\frac{\partial p(\boldsymbol{x}, t|\boldsymbol{\psi})}{\partial t} = -\nabla \cdot \left\{ \int \mathrm{d}\boldsymbol{x}_\tau [\boldsymbol{u}(t, \boldsymbol{x}, \boldsymbol{x}_\tau) \cdot p(\boldsymbol{x}, t; \boldsymbol{x}_\tau, t - \tau|\boldsymbol{\psi})] \right\}, \tag{2}$$

where $p(\boldsymbol{x}, t|\boldsymbol{\psi})$ is the probability density given the initial function, $p(\boldsymbol{x}, t; \boldsymbol{x}_\tau, t - \tau|\boldsymbol{\psi})$ is the joint probability density of being at $\boldsymbol{x}$ at time $t$ and at $\boldsymbol{x}_\tau$ at time $t-\tau$. The specific relationships and distinctions between the probability flows of ODEs and DDEs are elaborated in Appendix A.

## 2.2 Optimal Transport

Optimal transport (OT) provides a mathematical framework for transforming one probability distribution to another in the most cost-efficient manner (Villani et al., 2009; Santambrogio, 2015; Chen, 2016; Peyré et al., 2019). It seeks the optimal strategy for reallocating mass while minimizing the transportation cost. Formally, given two separable metric spaces $\mathcal{X}$ and $\mathcal{Y}$ with probability measures $\mu$ on $\mathcal{X}$ and $\nu$ on $\mathcal{Y}$, the objective is to determine a transport plan $\pi^*$ that minimizes the total transport cost, as defined in the Kantorovich problem (Kantorovich, 1942):

$$C(\mu, \nu) = \inf_{\pi \in \Pi(\mu, \nu)} \int_{\mathcal{X} \times \mathcal{Y}} c(\boldsymbol{x}, \boldsymbol{y}) \mathrm{d}\pi(\boldsymbol{x}, \boldsymbol{y}), \ \ \Pi(\mu, \nu) = \left\{ \pi : P_\#^{\boldsymbol{x}}(\pi) = \mu, P_\#^{\boldsymbol{y}}(\pi) = \nu \right\}, \tag{3}$$

where $c(\boldsymbol{x}, \boldsymbol{y})$ denotes the transportation cost of moving a unit mass from $\boldsymbol{x}$ to $\boldsymbol{y}$, $P_\#^{\boldsymbol{x}}(\pi)$ and $P_\#^{\boldsymbol{y}}(\pi)$ denote the marginal distributions of $\pi$ with respect to $\boldsymbol{x}$ and $\boldsymbol{y}$. In most cases, the transport cost between two points is defined as the squared Euclidean distance. The minimum total transport cost in this scenario corresponds to the squared 2-Wasserstein distance between the two distributions:

$$W_2(\mu, \nu)^2 = \inf_{\pi \in \Pi(\mu, \nu)} \int \|\boldsymbol{x} - \boldsymbol{y}\|^2 \mathrm{d}\pi(\boldsymbol{x}, \boldsymbol{y}). \tag{4}$$

## 2.3 Keypoint-Guided Optimal Transport

Traditional OT, which focuses solely on minimizing transport costs, often neglects other crucial constraints, resulting in suboptimal matching strategies. Thus, it is important to use a small set of well-matched source-target keypoint pairs, $\mathcal{K} = \{(\boldsymbol{x}_k, \boldsymbol{y}_k)\}_{k=1}^K$, to semi-supervise the transport strategy, especially when inherent structures and features of data need to be preserved.

To ensure correct transport with these keypoints, Keypoint-guided Optimal Transport (KPG-OT) is proposed (Gu et al., 2022; 2023), which guarantees that the relationships between each data point and the keypoints are preserved during transport. Formally, the objective can be expressed as:

$$\inf_{\tilde{\pi} \in \tilde{\Pi}(\mu, \nu)} \int_{\mathcal{X} \times \mathcal{Y}} g(\boldsymbol{x}, \boldsymbol{y}) \mathrm{d}(w \circ \tilde{\pi})(\boldsymbol{x}, \boldsymbol{y}), \ \ \tilde{\Pi}(\mu, \nu) = \left\{ \tilde{\pi} : P_\#^{\boldsymbol{x}}(w \circ \tilde{\pi}) = \mu, P_\#^{\boldsymbol{y}}(w \circ \tilde{\pi}) = \mu \right\}, \tag{5}$$

where $w \circ \tilde{\pi} = w(\boldsymbol{x}, \boldsymbol{y})\tilde{\pi}(\boldsymbol{x}, \boldsymbol{y})$ is the keypoint-masked transport plan, $w$ denotes the mask function. For a pair of keypoints $(\boldsymbol{x}_k, \boldsymbol{y}_k) \in \mathcal{K}$, the mask function is defined as: $w(\boldsymbol{x}_k, \boldsymbol{y}_k) = 1, w(\boldsymbol{x}_k, \boldsymbol{y}) = 0$ for $\boldsymbol{y} \neq \boldsymbol{y}_k, w(\boldsymbol{x}, \boldsymbol{y}_k) = 0$ for $\boldsymbol{x} \neq \boldsymbol{x}_k$, and $w(\boldsymbol{x}, \boldsymbol{y}) = 1$ if neither $\boldsymbol{x}$ nor $\boldsymbol{y}$ matches any keypoint. Obviously, the mask function defined above exactly preserves the matching of the keypoints in $\mathcal{K}$.

The cost function in Eq. (5) is defined as $g(\boldsymbol{x}, \boldsymbol{y}) = d[R^{\mathrm{s}}(\boldsymbol{x}), R^{\mathrm{t}}(\boldsymbol{y})]$, where $d$ denotes the Jensen-Shannon divergence. $R^{\mathrm{s}}(\boldsymbol{x}) \in (0, 1)^K$ (resp. $R^{\mathrm{t}}(\boldsymbol{y}) \in (0, 1)^K$) captures the relationship between $\boldsymbol{x}$ (resp. $\boldsymbol{y}$) and all source points (resp. target points) in $\mathcal{K}$, with its $k$-th dimension representing the relation score between $\boldsymbol{x}$ (resp. $\boldsymbol{y}$) and the $k$-th source keypoint $\boldsymbol{x}_k$ (resp. target keypoint $\boldsymbol{y}_k$):

$$R_k^{\mathrm{s}}(\boldsymbol{x}) = \frac{\mathrm{e}^{-c(\boldsymbol{x}, \boldsymbol{x}_k)/\tau}}{\sum_{i=1}^K \mathrm{e}^{-c(\boldsymbol{x}, \boldsymbol{x}_i)/\tau}}, R_k^{\mathrm{t}}(\boldsymbol{y}) = \frac{\mathrm{e}^{-c(\boldsymbol{y}, \boldsymbol{y}_k)/\tau}}{\sum_{i=1}^K \mathrm{e}^{-c(\boldsymbol{y}, \boldsymbol{y}_i)/\tau}}, \tag{6}$$

where $\tau$ is the temperature hyperparameter. Note that $g$ quantifies the similarity distance between the two relationship vectors outlined above. Therefore, minimizing the cost function in Eq. (5) effectively promotes the preservation of the relationship between each point and the keypoints.

## 2.4 ODE-BASED FLOW MATCHING

Flow Matching (FM) (Lipman et al., 2022) is a simulation-free training method for CNF, which trains the parameterized vector field $\boldsymbol{v}(t, \boldsymbol{x}; \theta)$ by regression on the target vector field $u(t, \boldsymbol{x})$ that generates the probability paths $p(\boldsymbol{x}, t)$ with marginals $p_0 = q_0$ and $p_1 = q_1$. The training objective is

$$\mathcal{L}_{\text{FM}}(\theta) = \mathbb{E}_{t, p(\boldsymbol{x}, t)} \| \boldsymbol{v}(t, \boldsymbol{x}; \theta) - \boldsymbol{u}(t, \boldsymbol{x}) \|^2, \tag{7}$$

where $t \sim \mathcal{U}(0, 1)$ follows the uniform distribution.

However, the explicit forms of $\boldsymbol{u}(t, \boldsymbol{x})$ and $p(\boldsymbol{x}, t)$ are intractable to compute. To address this, Conditional FM (CFM) (Tong et al., 2023a; Pooladian et al., 2023) introduces a latent variable $\boldsymbol{z}$ to construct the target probability path as a mixture of conditional probability paths $p(\boldsymbol{x}, t) = \mathbb{E}_{q(\boldsymbol{z})}[p(\boldsymbol{x}, t | \boldsymbol{z})]$ and modifies the training objective to:

$$\mathcal{L}_{\text{CFM}}(\theta) = \mathbb{E}_{t, q(\boldsymbol{z}), p(\boldsymbol{x}, t | \boldsymbol{z})} \| \boldsymbol{v}(t, \boldsymbol{x}; \theta) - \boldsymbol{u}(t, \boldsymbol{x} | \boldsymbol{z}) \|^2, \tag{8}$$

where $\boldsymbol{u}(t, \boldsymbol{x} | \boldsymbol{z})$ denotes the conditional vector field generating the conditional probability path $p(\boldsymbol{x}, t | \boldsymbol{z})$. Then, it can be shown that the gradient w.r.t $\theta$ of the CFM objective (8) is the same as that of the FM objective (7). Typically, we set the latent condition $\boldsymbol{z} := (\boldsymbol{x}_0, \boldsymbol{x}_1)$, with $q(\boldsymbol{z})$ being the coupling between distributions $q_0(\boldsymbol{x}_0)$ and $q_1(\boldsymbol{x}_1)$, based on a coupling strategy such as OT or KPG-OT. The conditional probability path is then modeled as a Gaussian flow:

$$p(\boldsymbol{x}, t | \boldsymbol{z}) = \mathcal{N}(\boldsymbol{x} | t \boldsymbol{x}_1 + (1 - t) \boldsymbol{x}_0, \sigma^2), \tag{9}$$

with the corresponding conditional vector field:

$$\boldsymbol{u}(t, \boldsymbol{x} | \boldsymbol{z}) = \boldsymbol{x}_1 - \boldsymbol{x}_0. \tag{10}$$

## 3 LIMITATIONS OF ODE-BASED FLOW MATCHING

Current continuous-time generative models utilize the parameterized vector field of ODEs to facilitate distribution transport. However, the inherent constraints of ODEs limit their ability to model specific transport strategies, leading to significant discrepancies between the modeled and true transport processes. In this section, we theoretically demonstrate the limitations of the ODE-based FM frameworks. Note that the following propositions are stated within the classical FM framework (Lipman et al., 2022; Liu et al., 2023). Recent ODE-based FM extensions enhance representational capacity and mitigate these limitations; we discuss these variants further in Section 5.

**Proposition 3.1** (Restriction on trajectory intersections). *Suppose that the target transport strategy, guided by certain key points or specific coupling constraints, inevitably induces trajectory intersections in the augmented phase space $(t, \boldsymbol{x})$ during the transport process. If the ODE-based FM is governed by a vector field that is Lipschitz continuous in $\boldsymbol{x}$ and continuous in $t$, then the corresponding flow map cannot precisely preserve the transport strategy.*

*Remark* 3.2. As demonstrated by Liu et al. (2023), when trajectory intersections occur in the transport strategy, FM tends to learn a *rectified flow*, with the targets of the trajectories being "rewired" at the intersection points. As a result, the given transport strategy cannot be accurately preserved. A simple example is shown in Fig. 1 (a).

**Proposition 3.3** (Heterogeneity in distributions). *Assume that the source (resp., target) distribution $q_0$ (resp., $q_1$) is supported on $M$ (resp., $N$) disjoint compact sets $\{U_i^0\}_{i=1}^M$ (resp., $\{U_j^1\}_{j=1}^N$), where each $U_i^0 \in \mathbb{R}^d$ (resp., $U_j^1 \in \mathbb{R}^d$) is a path-connected set with non-zero measure and $q_0(\boldsymbol{x}_0) > 0$ (resp., $q_1(\boldsymbol{x}_1) > 0$) for all $\boldsymbol{x}_0 \in \bigcup_{i=1}^M U_i^0$ (resp., $\boldsymbol{x}_1 \in \bigcup_{j=1}^N U_j^1$). If $M < N$, then for any transport map $T : \bigcup_{i=1}^M U_i^0 \to \bigcup_{j=1}^N U_j^1$ satisfying $T_\# q_0 = q_1$, the following holds: if the flow map induced by a Neural ODE is equal to the transport map $T$, then the associated vector field cannot be Lipchitz continuous.*

*Remark* 3.4. The conclusion emphasizes that accurate transport using an ODE-based FM requires regularity assumptions on distributions, which are often violated in real-world scenarios. For example, in single-cell dynamics, cells differentiate from one type into multiple types over time, leading to heterogeneous distributions before and after differentiation. A simple example is shown in Fig. 1 (b).

Moreover, when snapshot data are generated by a delay dynamical system, ODE-based FM fails to recover the true vector field with the delay term, resulting in inaccurate distribution transfer, interpolation, and extrapolation predictions.

## 4 DELAY FLOW MATCHING

To overcome the limitations of ODE-based FM, we introduce the Delay FM (DFM) framework, a new class of generative models based on Neural DDEs for distribution transport. Since the vector field incorporates delay terms, DFM allows trajectory intersections, enabling more accurate transport for tasks requiring keypoint-guided strategies. Additionally, by designing appropriate initial functions, DFM addresses singularities caused by distributional heterogeneity. A comparison between DDE-based and ODE-based FM under various scenarios is shown in Table 1.

Table 1: Comparison of ODE-based and DDE-based FM. DFM can handle not only KPG-OT coupling and trajectory intersections but also distribution heterogeneity by employing diverse initial functions. Furthermore, DFM can accurately model and recover the dynamics of delay dynamical systems.

| Methods | KPG-OT | Intersection | Heterogeneity | Delay Dynamics |
|---|---|---|---|---|
| FM | ✗ | ✗ | ✗ | ✗ |
| CFM | ✗ | ✗ | ✗ | ✗ |
| DFM(C) | ✓ | ✓ | ✗ | ✓ |
| DFM(D) | ✓ | ✓ | ✓ | ✓ |

### 4.1 FORMULATION

Consider a target probability flow $p(\boldsymbol{x}, t)$, generated by a vector field with a single delay term $\boldsymbol{u}(t, \boldsymbol{x}, \boldsymbol{x}_\tau)$ and initial functions $\boldsymbol{\psi} \sim q^\circ(\boldsymbol{\psi})$. The vector field and initial functions naturally define a joint probability flow at time $t$ and $t - \tau$, denoted as $p(\boldsymbol{x}, t; \boldsymbol{x}_\tau, t - \tau)$, which satisfies

$$\int p(\boldsymbol{x}, t; \boldsymbol{x}_\tau, t - \tau) \mathrm{d}\boldsymbol{x}_\tau = p(\boldsymbol{x}, t). \quad (11)$$

Note that $p(\boldsymbol{x}, t)$ and $p(\boldsymbol{x}, t; \boldsymbol{x}_\tau, t - \tau)$ can be modeled as a mixture of conditional distributions $p(\boldsymbol{x}, t|\boldsymbol{\psi})$ and $p(\boldsymbol{x}, t; \boldsymbol{x}_\tau, t - \tau|\boldsymbol{\psi})$, respectively, as follow:

$$p(\boldsymbol{x}, t) = \mathbb{E}_{q^\circ(\boldsymbol{\psi})} p(\boldsymbol{x}, t|\boldsymbol{\psi}),$$

$$p(\boldsymbol{x}, t; \boldsymbol{x}_\tau, t - \tau) = \mathbb{E}_{q^\circ(\boldsymbol{\psi})} p(\boldsymbol{x}, t; \boldsymbol{x}_\tau, t - \tau|\boldsymbol{\psi}),$$
$$(12)$$

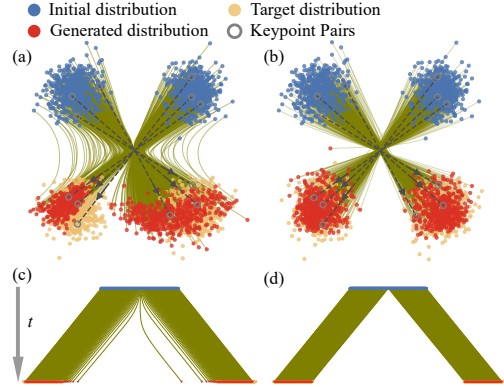

Figure 2: (a, b) show the generation results of CFM (a) and DFM (C) (b) under the KPG-OT coupling on Gaussian mixture data. (c, d) show the generation results of CFM (c) and DFM (D) (d) for heterogeneous source and target distributions.

where $p(\boldsymbol{x}, t|\boldsymbol{\psi})$ and $p(\boldsymbol{x}, t; \boldsymbol{x}_\tau, t - \tau|\boldsymbol{\psi})$ represent the probability flow generated by $\boldsymbol{u}(t, \boldsymbol{x}, \boldsymbol{x}_\tau)$ and the initial function $\boldsymbol{\psi}$, and they satisfy the delay Fokker-Planck equation (2). Based on the joint probability flow, we aim to learn a parameterized vector field $\boldsymbol{v}(t, \boldsymbol{x}, \boldsymbol{x}_\tau; \theta)$, which can generate the target probability flow, by minimizing the following regression objective against the target vector field:

$$\mathcal{L}_{\mathrm{DFM}}(\theta) = \mathbb{E}_{t, q^\circ(\boldsymbol{\psi}), p(\boldsymbol{x}, t; \boldsymbol{x}_\tau, t - \tau|\boldsymbol{\psi})} ||\boldsymbol{v}(t, \boldsymbol{x}, \boldsymbol{x}_\tau; \theta) - \boldsymbol{u}(t, \boldsymbol{x}, \boldsymbol{x}_\tau)||^2. \quad (13)$$

In most cases, both $p(\boldsymbol{x}, t; \boldsymbol{x}_\tau, t - \tau|\boldsymbol{\psi})$ and $\boldsymbol{u}(t, \boldsymbol{x}, \boldsymbol{x}_\tau)$ are computationally intractable. To address this, inspired by CFM, we introduce a latent variable $\boldsymbol{z}$ and further decompose the target probability path and joint probability path conditioned on the initial function into mixtures of simple probability paths and simple joint probability paths conditioned on both $\boldsymbol{\psi}$ and $\boldsymbol{z}$, respectively, as follow:

$$p(\boldsymbol{x}, t|\boldsymbol{\psi}) = \mathbb{E}_{q(\boldsymbol{z})} p(\boldsymbol{x}, t|\boldsymbol{z}, \boldsymbol{\psi}), \quad p(\boldsymbol{x}, t; \boldsymbol{x}_\tau, t - \tau|\boldsymbol{\psi}) = \mathbb{E}_{q(\boldsymbol{z})} p(\boldsymbol{x}, t; \boldsymbol{x}_\tau, t - \tau|\boldsymbol{z}, \boldsymbol{\psi}), \quad (14)$$

where $p(\boldsymbol{x}, t|\boldsymbol{z}, \boldsymbol{\psi}) = \int p(\boldsymbol{x}, t; \boldsymbol{x}_\tau, t - \tau|\boldsymbol{z}, \boldsymbol{\psi}) \mathrm{d}\boldsymbol{x}_\tau$. We can define the marginal vector field by marginalizing over the conditional vector field $\boldsymbol{u}(t, \boldsymbol{x}, \boldsymbol{x}_\tau|\boldsymbol{z})$ as follow:

$$\boldsymbol{u}(t, \boldsymbol{x}, \boldsymbol{x}_\tau) = \mathbb{E}_{q(\boldsymbol{z})} \frac{\boldsymbol{u}(t, \boldsymbol{x}, \boldsymbol{x}_\tau|\boldsymbol{z}) p(\boldsymbol{x}, t; \boldsymbol{x}_\tau, t - \tau|\boldsymbol{z}, \boldsymbol{\psi})}{p(\boldsymbol{x}, t; \boldsymbol{x}_\tau, t - \tau|\boldsymbol{\psi})}, \quad (15)$$

where $\boldsymbol{u}(t, \boldsymbol{x}, \boldsymbol{x}_\tau|\boldsymbol{z})$ represents the conditional vector field that generates $p(\boldsymbol{x}, t|\boldsymbol{z}, \boldsymbol{\psi})$ and $p(\boldsymbol{x}, t; \boldsymbol{x}_\tau, t - \tau|\boldsymbol{z}, \boldsymbol{\psi})$. Under this setup, we can derive the following result.

**Proposition 4.1.** *The marginal vector field $\boldsymbol{u}(t, \boldsymbol{x}, \boldsymbol{x}_\tau)$ given in Eq. (15), together with the selected initial function $\boldsymbol{\psi}$, generates the probability path $p(\boldsymbol{x}, t|\boldsymbol{\psi})$ and the joint probability path $p(\boldsymbol{x}, t; \boldsymbol{x}_\tau, t - \tau|\boldsymbol{\psi})$ in Eq. (14).*

Then, we can train the parameterized vector field by minimizing the following Delay Conditional FM (DCFM) objective:

$$\mathcal{L}_{\text{DCFM}}(\theta) = \mathbb{E}_{t, q(\boldsymbol{z}), q^\circ(\boldsymbol{\psi}), p(\boldsymbol{x}, t; \boldsymbol{x}_\tau, t - \tau|\boldsymbol{z}, \boldsymbol{\psi})} ||\boldsymbol{v}(t, \boldsymbol{x}, \boldsymbol{x}_\tau; \theta) - \boldsymbol{u}(t, \boldsymbol{x}, \boldsymbol{x}_\tau|\boldsymbol{z})||^2, \tag{16}$$

because the following proposition holds.

**Proposition 4.2.** *Given that $p(\boldsymbol{x}, t; \boldsymbol{x}_\tau, t - \tau|\boldsymbol{\psi}) > 0$ for all $\boldsymbol{x}, \boldsymbol{x}_\tau \in \mathbb{R}^d$ and $t \in [0, 1]$, up to a constant independent of $\boldsymbol{\theta}$, $\mathcal{L}_{DCFM}(\boldsymbol{\theta})$ and $\mathcal{L}_{DFM}(\boldsymbol{\theta})$ are equal, which further implies that $\nabla_{\boldsymbol{\theta}} \mathcal{L}_{DCFM}(\boldsymbol{\theta}) = \nabla_{\boldsymbol{\theta}} \mathcal{L}_{DFM}(\boldsymbol{\theta})$.*

### 4.2 SELECTION OF THE LATENT VARIABLE $\boldsymbol{z}$

#### 4.2.1 TRANSPORT BETWEEN TWO MEASURES

As outlined in Section 2.4, ODE-based FM typically selects latent variables as two endpoints $\boldsymbol{z} := (\boldsymbol{x}_0, \boldsymbol{x}_1)$ jointly sampled from the source and target distributions. In contrast, DFM requires the construction of a conditional joint probability density path for $\boldsymbol{x}(t)$ and $\boldsymbol{x}(t - \tau)$ based on the latent variable. Hence, we define the latent variable $\boldsymbol{z}$ as a entire path $\boldsymbol{\gamma}(t; \boldsymbol{x}_0, \boldsymbol{x}_1)$ connecting $\boldsymbol{x}_0$ and $\boldsymbol{x}_1$.

Formally, we define the probability distribution of the latent variable as $q[\boldsymbol{\gamma}(t; \boldsymbol{x}_0, \boldsymbol{x}_1)] := \pi(\boldsymbol{x}_0, \boldsymbol{x}_1)\mathcal{P}(\boldsymbol{\gamma}; \boldsymbol{x}_0, \boldsymbol{x}_1)$, where $\pi(\boldsymbol{x}_0, \boldsymbol{x}_1)$ represents the OT or KPG-OT coupling between the source and target distributions, and $\mathcal{P}(\boldsymbol{\gamma}; \boldsymbol{x}_0, \boldsymbol{x}_1)$ denotes the path measure pinned at $\boldsymbol{x}_0$ and $\boldsymbol{x}_1$. In practice, we can simply construct the path measure as a Dirac Delta distribution at a given path $\boldsymbol{\gamma}^*$ connecting $\boldsymbol{x}_0$ and $\boldsymbol{x}_1$:

$$\mathcal{P}(\boldsymbol{\gamma}; \boldsymbol{x}_0, \boldsymbol{x}_1) = \delta(\boldsymbol{\gamma} - \boldsymbol{\gamma}^*), \; \boldsymbol{\gamma}_0^* = \boldsymbol{x}_0, \boldsymbol{\gamma}_1^* = \boldsymbol{x}_1, \tag{17}$$

where $\boldsymbol{\gamma}^*$ is constructed using a specific interpolation method. For instance, it can be taken as the linear interpolation $\boldsymbol{\gamma}_t^* = (1 - t)\boldsymbol{x}_0 + t\boldsymbol{x}_1$, or alternatively, as a geodesic interpolation on the data manifold, based on appropriate manifold learning techniques.

After sampling a path $\boldsymbol{\gamma}(t; \boldsymbol{x}_0, \boldsymbol{x}_1) \sim q(\boldsymbol{z})$, the conditional joint probability density naturally degenerates into the following form:

$$p(\boldsymbol{x}, t; \boldsymbol{x}_\tau, t - \tau|\boldsymbol{\gamma}) = \delta[\boldsymbol{x} - \boldsymbol{\gamma}(t)]\delta[\boldsymbol{x}_\tau - \boldsymbol{\gamma}(t - \tau)], \tag{18}$$

which is exactly generated by the conditional vector field:

$$\boldsymbol{u}[t, \boldsymbol{x}, \boldsymbol{x}_\tau|\boldsymbol{\gamma}(t; \boldsymbol{x}_0, \boldsymbol{x}_1)] = \frac{\partial \boldsymbol{\gamma}(t; \boldsymbol{x}_0, \boldsymbol{x}_1)}{\partial t}. \tag{19}$$

#### 4.2.2 TRANSPORT BETWEEN MULTIPLE PROBABILITY MEASURES OVER TIME

For tasks with multiple target probability distributions over time $\{q_{t_j}\}_{j=0}^J$, where $t_0 = 0, t_J = T$, we define the latent variable $\boldsymbol{z} := \boldsymbol{\gamma}\left(t; \{\boldsymbol{x}_{t_j}\}_{j=0}^J\right)$ as a trajectory passing through $\boldsymbol{x}_{t_j}$ at time $t_j$, sampled from

$$q\left[\boldsymbol{\gamma}\left(t; \{\boldsymbol{x}_{t_j}\}_{j=0}^J\right)\right] := \mathcal{P}\left(\boldsymbol{\gamma}; \{\boldsymbol{x}_{t_j}\}_{j=0}^J\right) \prod_{i=0}^{J-1} \pi(\boldsymbol{x}_{t_i}, \boldsymbol{x}_{t_{i+1}}), \tag{20}$$

where $\{\boldsymbol{x}_{t_j}\}_{j=0}^J \sim \prod_{i=0}^{J-1} \pi(\boldsymbol{x}_{t_i}, \boldsymbol{x}_{t_{i+1}})$, with $\pi(\boldsymbol{x}_{t_i}, \boldsymbol{x}_{t_{i+1}})$ representing the OT or KPG-OT coupling between adjacent probability distributions at time $t_i$ and $t_{i+1}$, and $\mathcal{P}(\boldsymbol{\gamma}; \{\boldsymbol{x}_{t_j}\}_{j=0}^J)$ denotes the path measure pinned at $\{\boldsymbol{x}_{t_j}\}_{j=0}^J$. Similarly, we construct the path measure as:

$$\mathcal{P}\left(\boldsymbol{\gamma}; \{\boldsymbol{x}_{t_j}\}_{j=0}^J\right) = \delta(\boldsymbol{\gamma} - \boldsymbol{\gamma}^*), \; \boldsymbol{\gamma}_{t_j}^* = \boldsymbol{x}_{t_j}, \tag{21}$$

where $\boldsymbol{\gamma}^*$ represents a path passing through $\boldsymbol{x}_{t_j}$ at time $t_j$, which can be constructed using the cubic spline (CSpline) interpolation method. With this path as the latent variable, the resulting conditional joint probability path and conditional vector field are identical to those in Eq. (18) and Eq. (19), respectively.

### 4.3 SELECTION OF THE INITIAL FUNCTION

#### 4.3.1 DFM WITH CONSTANT INITIAL FUNCTIONS

In general, the initial function can be simply chosen as a constant (DFM(C)), i.e. $q^\circ(\boldsymbol{\psi}) = \delta(\boldsymbol{\psi} - \boldsymbol{\psi}^*)$, where $\boldsymbol{\psi}^*(t; \boldsymbol{x}_0) \equiv \boldsymbol{x}_0$ for $t \in [-\tau, 0]$ and $\boldsymbol{x}_0 \sim q_0$. Under this setting, we can theoretically prove that DFM can approximate any continuous transport strategy to arbitrary precision.

**Proposition 4.3** (Universal approximating capability of DDE's flow map). *For any given $\epsilon > 0$ and continuous transport map $F : \mathbb{R}^d \to \mathbb{R}^d$, which push-forward the source distribution $q_0$ to the target distribution $q_1$, i.e. $F_\# q_0 = q_1$, if there exists a neural network $\boldsymbol{f}(\boldsymbol{x}; \theta)$ such that $\|\boldsymbol{f}(\boldsymbol{x}; \theta) - [F(\boldsymbol{x}) - \boldsymbol{x}]\| < \epsilon$, for all $\boldsymbol{x} \in \mathbb{R}^d$, then we can construct a vector field with a single delay term $\boldsymbol{v}(t, \boldsymbol{x}, \boldsymbol{x}_\tau; \theta)$, where the corresponding flow map $G(\boldsymbol{x}; \theta)$, under the constant initial function condition, satisfies $\|G(\boldsymbol{x}; \theta) - F(\boldsymbol{x})\| < \epsilon$, for all $\boldsymbol{x} \in \mathbb{R}^d$.*

This indicates the exceptional representational capacity of DFM, enabling the modeling of a wider range of transport processes than ODE-based FM.

#### 4.3.2 DFM WITH DIVERSE INITIAL FUNCTIONS

In Section 3, we rigorously prove that ODE-based FM cannot effectively handle tasks with heterogeneous source and target distributions. Here, we demonstrate that DFM addresses this by designing diverse initial functions (DFM(D)).

Specifically, we employ clustering methods, such as Gaussian Mixture Model, DBSCAN, to partition the source dataset $\boldsymbol{X}_0 \sim q_0$ (resp. target dataset $\boldsymbol{X}_1 \sim q_1$) into $M$ (resp. $N$) mutually exclusive subsets, denoted as $\boldsymbol{X}_0^{(1)}, ..., \boldsymbol{X}_0^{(M)}$ (resp. $\boldsymbol{X}_1^{(1)}, ..., \boldsymbol{X}_1^{(N)}$). We can assign a normalized mass to each subset $\boldsymbol{X}_0^{(m)}$ (resp. $\boldsymbol{X}_1^{(n)}$) as $\rho_0^{(m)} = |\boldsymbol{X}_0^{(m)}|/|\boldsymbol{X}_0|$ (resp., $\rho_1^{(n)} = |\boldsymbol{X}_1^{(n)}|/|\boldsymbol{X}_1|$). If the endpoints $(\boldsymbol{x}_0, \boldsymbol{x}_1)$ of a sampled trajectory $\boldsymbol{\gamma}$ are drawn from $\boldsymbol{X}_0^{(m)}$ and $\boldsymbol{X}_1^{(n)}$, we assign it an initial function $\boldsymbol{\psi}_{mn}^*$ which has a constant time derivative $C_{mn}$, i.e.:

$$\frac{\mathrm{d}\boldsymbol{\psi}_{mn}^*(t; \boldsymbol{x}_0)}{\mathrm{d}t} = C_{mn}, \ \boldsymbol{\psi}_{mn}^*(0; \boldsymbol{x}_0) = \boldsymbol{x}_0, \tag{22}$$

where $t \in [-\tau, 0]$. In this case, $q^\circ(\boldsymbol{\psi})$ is a discrete distribution which satisfies:

$$\sum_{n=1}^N q^\circ(\psi_{mn}^*) = \rho_0^{(m)}, \quad \sum_{m=1}^M q^\circ(\psi_{mn}^*) = \rho_1^{(n)}. \tag{23}$$

$p(\boldsymbol{x}_0, 0 | \boldsymbol{\psi} = \psi_{mn}^*) = q_0(\boldsymbol{x}_0 | \boldsymbol{\psi} = \psi_{mn}^*)$ represents empirical data distribution available as data points in $\boldsymbol{X}_0^{(m)}$ whose corresponding transport target is in $\boldsymbol{X}_1^{(n)}$. By coupling the source and target data through OT or KPG-OT, we can obtain the transport target for any initial point. This enables the construction of $q^\circ(\psi)$ as described above, from which we can sample to obtain the corresponding initial function.

In summary, we design distinct initial functions for different subsets of the source and target data, guiding the vector field from different initial subsets to corresponding target subsets, thereby effectively handling distributional heterogeneity.

### 4.4 GENERATION PROCESS BASED ON NEURAL DDEs

After training, we sample data from $q_0$ and initial functions from $q^\circ(\psi)$, then generate the target data by solving the forward pass of the trained vector field with a single delay term using a piecewise ODE solver, as in Neural DDEs (Zhu et al., 2020).

## 5 RELATED WORKS

A number of recent methods have been proposed to address the limitations of ODE-based FM. To enable crossing trajectories, Constant Acceleration Flow (Park et al., 2024) models acceleration and conditions it on an initial velocity; Hierarchical Rectified Flow (Zhang et al., 2025) achieves trajectory

crossings by hierarchically coupling multiple ODEs; and Augmented Bridge Matching (De Bortoli et al., 2023) incorporates information about the initial point directly into the vector field. Diversified Flow Matching (Shrestha & Fu, 2025) takes a different perspective by training interpolation strategies that construct non-intersecting target trajectories, thereby indirectly mitigating trajectory-crossing issues. To address the singularities caused by distributional heterogeneity, Switched Flow Matching (Zhu & Lin, 2024) and Variational Rectified Flow Matching (Guo & Schwing, 2025) introduce additional latent variables that allow the model to represent multi-modal transport paths, alleviating the degeneracies that arise under heterogeneous target distributions.

The methods discussed above introduce various extensions to classical FM, enabling the model to accommodate more transport strategies. While these approaches either allow trajectory intersections during the transport process or alleviate singularities arising from distributional heterogeneity, most of them cannot simultaneously resolve both issues. Moreover, they remain confined to the ODE-based generative framework, typically resolving limitations by introducing additional conditioning variables into the vector field or by redesigning the target transport process.

In contrast, our proposed DFM departs from the ODE paradigm and introduces a fundamentally new DDE-based generative framework. By modeling the vector field directly in the original phase space, without introducing auxiliary latent variables or specially engineered transport paths, DFM inherently accommodates both trajectory intersections and distributional heterogeneity within a unified mechanism. Furthermore, DFM naturally captures transport dynamics governed by intrinsic delays, a capability that is entirely out of reach for those ODE-based extensions.

## 6 EXPERIMENTS

We demonstrate the advantages of DFM over ODE-based FM across various tasks, including reconstructing delay dynamics from snapshots, inferring differentiation trajectories from single-cell RNA sequencing (scRNA-seq) data, and image generation. The experimental details and additional results can be found in Appendix D. A systematic ablation study is shown in Appendix F.

### 6.1 SIMPLE SYNTHETIC ILLUSTRATIONS

To more clearly illustrate the DFM workflow, we present two simple and concrete running examples, demonstrating the procedure of DFM(C) and DFM(D), respectively.

**2-d Gaussian mixture dataset.** As shown in Fig. 2 (a,b), we aim to transport a two-component Gaussian mixture to its reflection-symmetric counterpart while respecting 6 keypoint pairs (i.e., the optimal mapping $\boldsymbol{x} \to -\boldsymbol{x}$). Using KPG-OT, we first obtain pointwise couplings and construct the linear interpolation paths $\gamma = t \times \boldsymbol{x}_0 + (1-t) \times \boldsymbol{x}_1$, which unavoidably produce trajectory crossings. We select the constant initial function with $\tau = 1$, and train the delay vector field. Notably, this toy problem admits an exact DDE solution: $\dot{\boldsymbol{x}} = -2\boldsymbol{x}(t-1) = -2\boldsymbol{x}_0$. The learned DFM(C) exhibits the same behavior: trajectories select different directions at the crossing by leveraging historical states, yielding accurate keypoint-respecting transport.

**1-d example with distributional heterogeneity.** As shown in Fig. 2 (c,d), we aim to realize the transport $\mathcal{U}(-1, 1) \to \frac{1}{2}\mathcal{U}(-3, -2) + \frac{1}{2}\mathcal{U}(2, 3)$. Notably, we can use $\dot{x} = x(t) - x(t-1)$ to achieve precise OT transport by assigning different initial functions: ① $x(t) = x_0 - t, x_0 \in [-1, 0]$ and ② $x(t) = x_0 + t, x_0 \in (0, 1]$. During training, we employed the OT coupling, which maps $[-1, 0]$ to $[-3, -2]$ and $(0, 1]$ to $[2, 3]$, and construct the linear interpolation paths. In applying DFM(D), initial function ① is selected for $x_1 \in [-3, -2]$, while ② is chosen for $x_1 \in [2, 3]$, with $\tau = 1$. The trained DFM(D) achieves accurate transfer between heterogeneous distributions (Fig. 2 (d)).

### 6.2 SYNTHETIC DATASETS OF DELAY DYNAMICAL SYSTEMS

**Biological autoregulation motif**. DDEs provide a simplified framework for modeling biological network motifs (Glass et al., 2021). We consider recovering the delay dynamics from snapshot data generated by the autoregulation model: $\dot{x}(t) = \frac{\eta}{1+x^n(t-\tau)}$ with $\tau = 1, n = 2, \eta = 5$, which produces damped oscillations (Fig. 3 (a,b)). Using $1,000$ initial values sampled from $\mathcal{U}(0.2, 1.2)$ and constant initial functions, we generate trajectories and collect snapshots at $t=0, 1, \ldots, 7$ for training. During

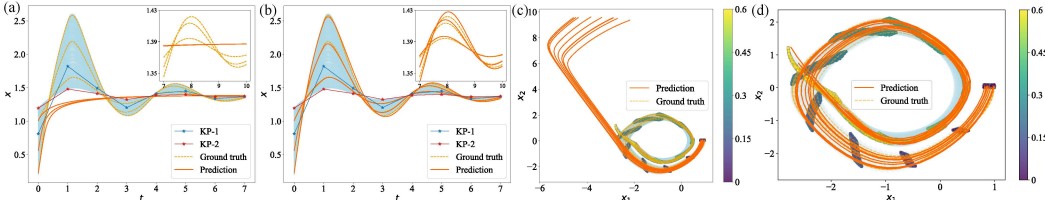

Figure 3: Comparison of predicted trajectories of CFM and DFM trained on snapshots of delay dynamical systems. (a-b) Interpolation and extrapolation (insets) results on the biological autoregulation motif dataset using KP-CFM (a) and KP-DFM(C) (b), both with two keypoints. (c-d) Interpolation results on the spiral DDE using OT-CFM (c) and OT-DFM(C) (d).

training, KPG-OT (KP-) matches minibatches at adjacent time steps using two known keyoints. After training, forward integration from $t=0$ shows that KP-CFM fails to recover the dynamics, while KP-DFM(C) with $\tau=1$ accurately captures and extrapolates the oscillatory behavior (insets in Fig. 3 (a,b)). Results with other delays and error metrics are provided in Appendix D.1 and F.

**Spiral DDE**. We next consider a 2-d DDE (Zhu et al., 2020): $\dot{\boldsymbol{x}}(t) = \boldsymbol{A} \tanh[\boldsymbol{x}(t) + \boldsymbol{x}(t-\tau)]$ with $\tau=0.5$ and $\boldsymbol{A} \in \mathbb{R}^{2\times2}$, which produces crossing spiral trajectories (Fig. 3 (c,d)). Snapshots are taken every 0.05 in $t \in [0, 0.6]$ and coupled using minibatch-OT (OT-) between adjacent steps. After training, OT-DFM(C) with $\tau=0.5$ successfully reproduces the dynamics, while OT-CFM fails around the crossing region. DFM also generalizes well across a range of $\tau$ values (Appendix D.2, F).

## 6.3 TRAJECTORY INFERENCE OF SINGLE-CELL

We investigate the inference of differentiation trajectories from real scRNA-seq data, where heterogeneity increases as cells transition from a single type to multiple types during development, complicating modeling with ODE-based approaches. To evaluate performance, we train on all time points except one intermediate time, then sample from the initial distribution and perform forward integration to predict distributions at each time point. Predictions are assessed via unsupervised leave-one-out validation (L) against the held-out intermediate distribution and supervised final-time validation (F) against the final distribution. Trajectory accuracy is measured using the 2-Wasserstein distance ($W_2$) and Maximum Mean Discrepancy with a Gaussian kernel (MMD(G)).

Table 2: Trajectory inference results on the scRNA-seq dataset in mouse hematopoiesis and the single-cell qPCR iPSC dataset with bifurcation. All results are averages of 10 runs.

| Methods | Mouse hematopoiesis | | | | qPCR iPSC | | | |
|---|---|---|---|---|---|---|---|---|
| | $W_2$(L) | MMD(L) | $W_2$(F) | MMD(F) | $W_2$(L) | MMD(L) | $W_2$(F) | MMD(F) |
| TIGON | 0.519 | 0.563 | 0.264 | 0.155 | 0.733 | 0.791 | 0.695 | 0.405 |
| MIOFlow | 0.514 | 0.629 | 0.220 | 0.056 | 0.770 | 1.039 | 0.345 | 0.155 |
| OT-CFM | 0.378 | 0.357 | 0.192 | 0.047 | 0.579 | 0.492 | 0.226 | 0.030 |
| OT-DFM(C) | 0.379 | 0.384 | 0.136 | 0.021 | 0.553 | 0.447 | 0.234 | 0.041 |
| OT-DFM(D) | **0.372** | **0.341** | **0.095** | **0.010** | **0.532** | **0.399** | **0.213** | **0.027** |

**Mouse hematopoiesis dataset**. We evaluate DFM on scRNA-seq data from mouse hematopoiesis (Weinreb et al., 2020), focusing on cells differentiating into neutrophils (Neu) and monocytes (Mo) at Day 2, 4, and 6. Models are trained on Day 2 and 6 data; after training, Day 2 samples are integrated forward to predict distributions at Days 4 and 6. Geodesic interpolation is used to construct trajectories. Due to the heterogeneity between pre- and post-differentiation distributions, trajectories inferred by ODE-based models (TIGON (Sha et al., 2024), MIOFlow (Huguet et al., 2022), OT-CFM) partially deviate from the data manifold, falling between Neu and Mo fates (Fig. 4 (b-e)). In contrast, OT-DFM(D), using distinct initial functions for Neu and Mo (Appendix D.3), preserves alignment with the data manifold (Fig. 4 (f)) and achieves more accurate predictions (Table 2).

**qPCR iPSC dataset**. We further apply DFM to model the bifurcation process of iPSCs in cardiomyocytes (Bargaje et al., 2017). Data from Day 2, 3, 4, and 5 are selected, with bifurcation observed from Day 3, where progenitor cells differentiate into mesodermal (M) and endodermal (En) fates. Models are trained on all time points except Day 3. After training, samples from Day 2 are forward-integrated to infer trajectories and predict distributions at Day 3 and 5. Due to distributional heterogeneity at the

bifurcation point, ODE-based methods and OT-DFM(C) produce trajectories that partially misalign with the data manifold (Fig. 4 (h–k)). In contrast, OT-DFM(D), by assigning separate initial functions for each fate (Appendix D.4), preserves trajectory alignment with the corresponding manifolds and yields more accurate predictions at both Day 3 and 5 compared to other methods (Table 2).

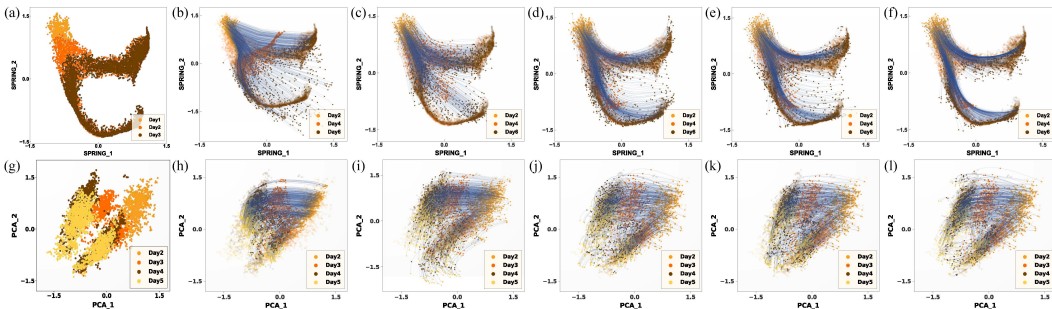

Figure 4: Comparisons between predicted trajectories of MIOFlow (b, h), TIGON (c, i), OT-CFM (d, j), OT-DFM(C) (e, k), and OT-DFM(D) (f, l) on scRNA-seq dataset in mouse hematopoiesis (a-f) and qPCR iPSC dataset (g-l). (a) and (g) illustrate the true data distribution.

## 6.4 IMAGE GENERATION

**MNIST dataset**. We design a *Semi-paired Image-to-Image Translation* task on the MNIST dataset, where the source domain consists of original images, and the target domain includes their negative images (the normalized pixel values transformed via $x \mapsto 1-x$). To provide

Table 3: Comparison of FID between KP-CFM ($\tau = 0$) and KP-DFM(C) with different time delay on the MNIST dataset.

| $\tau$ | 0 (CFM) | 0.125 | 0.250 | 0.500 | 1.000 |
|---|---|---|---|---|---|
| FID | 45.020 | 28.497 | 11.747 | 12.653 | 12.031 |

partial supervision, 10% of the training data are paired with their negative counterparts as keypoints. During training, minibatches are independently sampled from the source and target distributions, and coupled via KPG-OT (KP-). The task aims to transform source images into their negatives, where, under linear interpolation, all transport paths intersect at an image with uniform pixel values of 0.5. While KP-CFM struggles with this transformation, KP-DFM(C) effectively handles it. As shown in Table 3, across varying delays $\tau$, KP-DFM(C) consistently achieves lower FID scores than KP-CFM.

**CIFAR-10 dataset**. We further evaluate DFM on the CIFAR-10 dataset under a more general generation setup (Zhu & Lin, 2024), where the source distribution is a two-component Gaussian mixture. DFM employs trainable initial functions with distinct constant time derivatives, enabling generation from different mixture components to specific image classes. We compare

Table 4: Comparison of FID between CFM and DFM on the CIFAR-10 dataset.

| NFE | 10 | 20 | 30 | 40 | Adap. |
|---|---|---|---|---|---|
| I-CFM | 108.291 | 94.629 | 91.404 | 90.254 | 88.306 |
| OT-CFM | 78.165 | 27.512 | 16.409 | 12.026 | 6.162 |
| I-DFM(D) | **54.064** | **18.248** | **11.429** | **9.008** | **4.980** |
| OT-DFM(D) | 54.222 | 18.598 | 11.894 | 9.287 | 5.191 |

CFM and DFM(D) under two coupling strategies: independent coupling (I-) and OT coupling (OT-). As shown in Table 4, I-CFM struggles with mode heterogeneity, while I-DFM(D) generates higher-quality images. With OT coupling, OT-DFM(D) consistently outperforms OT-CFM, especially when the number of function evaluations (NFE) is small. Additional results are provided in Appendix D.6.

## 7 CONCLUSION

We introduce DFM, a novel continuous-time generative framework based on Neural DDEs. Through theoretical analysis, we highlight the limitations of ODE-based generative models, particularly their inability to capture certain transport strategies and preserve critical coupling information. In contrast, DFM offers universal approximation for arbitrary continuous transport strategies, addressing these shortcomings effectively. DFM also overcomes the challenge of transport between heterogeneous distributions by incorporating task-specific initial functions. Furthermore, it is naturally suited for modeling delay dynamical systems, a feature beyond the capability of ODEs. Extensive experiments on both synthetic and real-world datasets demonstrate that DFM achieves significantly more precise and versatile distribution transport strategies compared to FM.

ACKNOWLEDGMENTS

Q. Zhu is supported by the National Key R&D Program of China (No. 2025YFA1016503), by the National Natural Science Foundation of China (NSFC) (Nos. T2541024, 62406072, and 12171350), and by the STCSM (No. 23YF1402500). W. Lin is supported by the NSFC (Nos. 11925103, 12531018), the STCSM (Nos. 2021SHZDZX0103, 22JC1402500, 22JC1401402, 25JS2810400), and the SMEC (Nos. 2023ZKZD04, 2023KEJI05-72). X. Lu is supported by the National Science and Technology Major Project for Brain Science and Brain-like Intelligence Technology (No. 2025ZD0215700), the National Natural Science Foundation of China (Nos. 72025405, 72421002, 92467302, 72474223, 72301285), the Hunan Science and Technology Plan Project (Nos. 2023JJ40685, 2024RC3133), and the Major Program of Xiangjiang Laboratory (No. 24XJJCYJ01001).

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

The structure of the appendix is as follows:

- Appendix A provides the connection between the probability flows of ODEs and DDEs.
- Appendix B provides the formal proofs for the theoretical results presented in the main text.
- Appendix C provides pseudocode for DFM(C) and DFM(D).
- Appendix D includes the experimental setup details and additional experimental results.
- Appendix E provides the comparison of the training and inference time required by DFM and CFM.
- Appendix F provides the sensitivity anaysis of time delay parameter $\tau$ on both synthetic and real-world datasets.
- Appendix G compares DFM with other ODE-based generative methods designed to address trajectory intersections and distributional heterogeneity, further highlighting the advantages of the DDE-based framework.

## A  CONNECTION BETWEEN THE PROBABILITY FLOWS OF ODES AND DDES

Based on the vector field $\boldsymbol{u}[t, \boldsymbol{x}(t), \boldsymbol{x}(t-\tau)]$ in Eq. (1), we can define the conditional average drift (CAD) without delay terms as follows (Guillouzic et al., 1999):

$$\overline{\boldsymbol{u}}(t, \boldsymbol{x}|\boldsymbol{\psi}) = \int \mathrm{d}\boldsymbol{x}_\tau [\boldsymbol{u}(t, \boldsymbol{x}, \boldsymbol{x}_\tau) p(\boldsymbol{x}_\tau, t-\tau|\boldsymbol{x}, t; \boldsymbol{\psi})], \tag{24}$$

where $p(\boldsymbol{x}_\tau, t - \tau|\boldsymbol{x}, t; \boldsymbol{\psi})$ denotes the conditional probability given $\boldsymbol{x}(t) = \boldsymbol{x}$. Note that $p(\boldsymbol{x}, t; \boldsymbol{x}_\tau, t-\tau|\boldsymbol{\psi}) = p(\boldsymbol{x}_\tau, t-\tau|\boldsymbol{x}, t; \boldsymbol{\psi})p(\boldsymbol{x}, t|\boldsymbol{\psi})$, so Eq. (2) is equivalent to:

$$\frac{\partial p(\boldsymbol{x}, t|\boldsymbol{\psi})}{\partial t} = -\nabla \cdot \{\overline{\boldsymbol{u}}(t, \boldsymbol{x}|\boldsymbol{\psi})p(\boldsymbol{x}, t|\boldsymbol{\psi})\}, \tag{25}$$

which is precisely the continuity equation satisfied by the probability flows of ODEs.

*Remark* A.1. This implies that a non-delayed vector field can be constructed to match the probability flow of the delayed vector field in Eq. (1). However, the integration in Eq. (24) eliminates the coupling information between the states at times $t$ and $t - \tau$. Consequently, while the ODE preserves the marginal probability flow, it fails to maintain the coupling relationship during distribution transfer, as illustrated by a simple example in Fig. 1 (a).

## B  PROOFS OF THEORETICAL RESULTS

### B.1  FOKKER-PLANCK EQUATION OF DDES

**Theorem B.1** (Fokker-Planck equation of DDEs, (Guillouzic et al., 1999; Frank, 2005))**.** *Consider a time-dependent DDE with a single delay term:*

$$\frac{\mathrm{d}\boldsymbol{x}(t)}{\mathrm{d}t} = \boldsymbol{u}[t, \boldsymbol{x}(t), \boldsymbol{x}(t-\tau)], \quad t \in [0, T],$$
$$\boldsymbol{x}(h) = \boldsymbol{\psi}(h), \quad h \in [-\tau, 0], \tag{26}$$

*where $\boldsymbol{u}[t, \boldsymbol{x}(t), \boldsymbol{x}(t-\tau)] : [0, T] \times \mathbb{R}^d \times \mathbb{R}^d \to \mathbb{R}^d$ is a smooth vector field which is abbreviated as $\boldsymbol{u}(t, \boldsymbol{x}, \boldsymbol{x}_\tau)$ in the following, $\boldsymbol{\psi}(h)$ represents the continuous initial function. The associated probability flows $p(\boldsymbol{x}, t|\boldsymbol{\psi}) : \mathbb{R}^d \times [0, T] \to \mathbb{R}^+$ satisfy the delay Fokker-Planck equation:*

$$\frac{\partial p(\boldsymbol{x}, t|\boldsymbol{\psi})}{\partial t} = -\nabla \cdot \left\{ \int \mathrm{d}\boldsymbol{x}_\tau [\boldsymbol{u}(t, \boldsymbol{x}, \boldsymbol{x}_\tau) p(\boldsymbol{x}, t; \boldsymbol{x}_\tau, t-\tau|\boldsymbol{\psi})] \right\}, \tag{27}$$

*where $p(\boldsymbol{x}, t|\boldsymbol{\psi})$ is the probability density at time $t$ given the initial function, while $p(\boldsymbol{x}, t; \boldsymbol{x}_\tau, t-\tau|\boldsymbol{\psi})$ is the joint probability density representing the likelihood of the system being at $\boldsymbol{x}$ at time $t$ and at $\boldsymbol{x}_\tau$ at time $t - \tau$.*

*Proof.* The proof follows a similar approach as presented in Guillouzic et al. (1999). Without loss of generality, we assume that $d = 1$ and the state variable $x \in [a, b]$ in Eq. (26). Consider an arbitrary $C^2$ function $F(x)$ defined on the interval $[a, b]$, i.e. $F \in C^2([a, b])$, which satisfies the following conditions:

$$\lim_{x \to a} F(x) = \lim_{x \to b} F(x) = 0, \tag{28}$$

$$\lim_{x \to a} \frac{\mathrm{d}}{\mathrm{d}x} F(x) = \lim_{x \to b} \frac{\mathrm{d}}{\mathrm{d}x} F(x) = 0. \tag{29}$$

Then, by applying the Taylor expansion, we obtain:

$$\mathrm{d}F[x(t)] = F[x(t) + \mathrm{d}x(t)] - F[x(t)] = \left\{ u[t, x(t), x(t-\tau)] \frac{\mathrm{d}}{\mathrm{d}x} F[x(t)] \right\} \mathrm{d}t. \tag{30}$$

The ensemble average (average over realizations) of $\mathrm{d}F[x(t)]$ can be written as

$$\left\langle \frac{\mathrm{d}}{\mathrm{d}t} F[x(t)] \right\rangle = \left\langle u[t, x(t), x(t-\tau)] \frac{\mathrm{d}}{\mathrm{d}x} F[x(t)] \right\rangle. \tag{31}$$

We denote $p(x, t; x_\tau, t - \tau | \psi) \mathrm{d}x \mathrm{d}x_\tau$ as the probability that $x(t) \in [x, x + \mathrm{d}x]$ and $x(t - \tau) \in [x_\tau, x_\tau + \mathrm{d}x_\tau]$ given the initial function $\psi$. Then, Eq. (31) is equivalent to

$$\int_a^b \mathrm{d}x F(x) \int_a^b \mathrm{d}x_\tau \frac{\partial}{\partial t} p(x, t; x_\tau, t - \tau | \psi)$$
$$= \int_a^b \mathrm{d}x \frac{\mathrm{d}}{\mathrm{d}x} F[x(t)] \int_a^b \mathrm{d}x_\tau u(t, x, x_\tau) p(x, t; x_\tau, t - \tau | \psi). \tag{32}$$

By applying the integration by parts formula to the right-hand side, we obtain:

$$\int_a^b \mathrm{d}x F(x) \int_a^b \mathrm{d}x_\tau \frac{\partial}{\partial t} p(x, t; x_\tau, t - \tau | \psi)$$
$$= \int_a^b \mathrm{d}x F(x) \int_a^b \mathrm{d}x_\tau \left\{ -\frac{\partial}{\partial x} [u(t, x, x_\tau) p(x, t; x_\tau, t - \tau | \psi)] \right\}, \tag{33}$$

where the surface terms are neglected due to Eq. (28) and Eq. (29). Since $F(x)$ is arbitrary, Eq. (33) leads to

$$\frac{\partial}{\partial t} p(x, t | \psi) = -\frac{\partial}{\partial x} \left\{ \int_a^b \mathrm{d}x_\tau [u(t, x, x_\tau) p(x, t; x_\tau, t - \tau | \psi)] \right\}. \tag{34}$$

□

## B.2 PROOFS OF PROPOSITIONS IN THE MAIN TEXT

**Proposition 3.1** (Restriction on trajectory intersections). *Suppose that the target transport strategy, guided by certain key points or specific coupling constraints, inevitably induces trajectory intersections in the augmented phase space $(t, \boldsymbol{x})$ during the transport process. If the ODE-based FM is governed by a vector field that is Lipschitz continuous in $\boldsymbol{x}$ and continuous in $t$, then the corresponding flow map cannot precisely preserve the transport strategy.*

*Proof of Proposition 3.1.* Consider a time-dependent Neural ODE corresponding to a ODE-based FM with the following form:

$$\frac{\mathrm{d}\boldsymbol{x}(t)}{\mathrm{d}t} = \boldsymbol{v}[t, \boldsymbol{x}(t); \theta], \quad t \in [0, T],$$
$$\boldsymbol{x}(0) = \boldsymbol{x}_0, \tag{35}$$

where $\boldsymbol{v}[t, \boldsymbol{x}(t); \theta] : [0, T] \times \mathbb{R}^d \to \mathbb{R}^d$ is a parameterized vector field. Since the vector field is Lipschitz continuous in $\boldsymbol{x}$ and continuous in $t$, by the Picard-Lindelöf Theorem, the corresponding initial value problem (35) has a unique solution on the interval $[0, T]$. Suppose that there exist two distinct solutions $\boldsymbol{x}^1(t)$ and $\boldsymbol{x}^2(t)$ corresponding to different initial values $\boldsymbol{x}_0^1$ and $\boldsymbol{x}_0^2$, which intersect at $(t^*, \boldsymbol{x}^*)$. By the uniqueness of the solution, these two trajectories must correspond to the same solution, leading to a contradiction. Therefore, given a transport strategy, which inevitably leads to trajectory intersections during the transport process, the flow map corresponding to the ODE-based FM cannot precisely ensure the transport strategy. □

**Proposition 3.3** (Heterogeneity in distributions). *Assume that the source (resp., target) distribution $q_0$ (resp., $q_1$) is supported on $M$ (resp., $N$) disjoint compact sets $\{U_i^0\}_{i=1}^M$ (resp., $\{U_j^1\}_{j=1}^N$), where each $U_i^0 \in \mathbb{R}^d$ (resp., $U_j^1 \in \mathbb{R}^d$) is a path-connected set with non-zero measure and $q_0(\boldsymbol{x}_0) > 0$ (resp., $q_1(\boldsymbol{x}_1) > 0$) for all $\boldsymbol{x}_0 \in \bigcup_{i=1}^M U_i^0$ (resp., $\boldsymbol{x}_1 \in \bigcup_{j=1}^N U_j^1$). If $M < N$, then for any transport map $T : \bigcup_{i=1}^M U_i^0 \to \bigcup_{j=1}^N U_j^1$ satisfying $T_\# q_0 = q_1$, the following holds: if the flow map induced by a Neural ODE is equal to the transport map $T$, then the associated vector field cannot be Lipschitz continuous.*

*Proof of Proposition 3.3.* Assume there exists a parameterized vector field $\boldsymbol{v}(t, \boldsymbol{x}; \theta)$ whose flow map $\boldsymbol{\phi}_T$ coincides with the transport map $T$, i.e., $\boldsymbol{\phi}_T(\boldsymbol{x}) = T(\boldsymbol{x})$ for all $\boldsymbol{x} \in \bigcup_{i=1}^M U_i^0$.

First, consider the case $M < N$. Then there must exist $m \in \{1, 2, \cdots, M\}$, and $n_1, n_2 \in \{1, 2, \cdots, N\}$ such that $q_1[T(U_m^0) \cap U_{n_1}^1] > 0$ and $q_1[T(U_m^0) \cap U_{n_2}^1] > 0$. Otherwise, for each $m \in \{1, 2, \cdots, M\}$, there exists a unique $n \in \{1, 2, \cdots, N\}$ such that $q_1[T(U_m^0) \cap U_{n'}^1] = 0$ for all $n' \neq n$. Since $M < N$, there exists $n^* \in \{1, 2, \cdots, N\}$ such that $\sum_{i=1}^M q_1[T(U_i^0) \cap U_{n^*}^1] = 0$, which implies

$$q_1(U_{n^*}^1) = q_1 \left[ T\left( \bigcup_{i=1}^M U_i^0 \right) \cap U_{n^*}^1 \right] = q_1 \left[ \bigcup_{i=1}^M T(U_i^0) \cap U_{n^*}^1 \right] = 0, \tag{36}$$

a contradiction.

Hence, we can choose $\boldsymbol{x}_{m,1}, \boldsymbol{x}_{m,2} \in U_m^0$ such that $T(\boldsymbol{x}_{m,1}) \in U_{n_1}^1$, $T(\boldsymbol{x}_{m,2}) \in U_{n_2}^1$. By the path-connectedness of $U_m^0$, there exists a continuous path $\gamma : [0,1] \to U_m^0$ with $\gamma(0) = \boldsymbol{x}_{m,1}$ and $\gamma(1) = \boldsymbol{x}_{m,2}$. Then, for any $\epsilon > 0$, there exist $\boldsymbol{x}_s, \tilde{\boldsymbol{x}}_s \in \gamma$ such that $|\boldsymbol{x}_s - \tilde{\boldsymbol{x}}_s| < \epsilon$, and $T(\boldsymbol{x}_s)$ and $T(\tilde{\boldsymbol{x}}_s)$ lie in distinct subsets of $\{U_j^1\}_{j=1}^N$. Since each $U_j^1$ is a compact subset of $\mathbb{R}^d$, it follows that for any $j_1, j_2 \in \{1, 2, \cdots, N\}$, $j_1 \neq j_2$,

$$\mathrm{dist}\left(U_{j_1}^1, U_{j_2}^1\right) = \inf\left\{ |\boldsymbol{x} - \boldsymbol{y}| \mid \boldsymbol{x} \in U_{j_1}^1, \boldsymbol{y} \in U_{j_2}^1 \right\} > 0. \tag{37}$$

Let

$$d = \min_{j_1 \neq j_2} \left\{ \mathrm{dist}\left(U_{j_1}^1, U_{j_2}^1\right) \right\}_{j_1, j_2} > 0. \tag{38}$$

Then we have

$$\frac{|\boldsymbol{\phi}_T(\boldsymbol{x}_s) - \boldsymbol{\phi}_T(\tilde{\boldsymbol{x}}_s)|}{|\boldsymbol{x}_s - \tilde{\boldsymbol{x}}_s|} = \frac{|T(\boldsymbol{x}_s) - T(\tilde{\boldsymbol{x}}_s)|}{|\boldsymbol{x}_s - \tilde{\boldsymbol{x}}_s|} > \frac{d}{\epsilon}. \tag{39}$$

On the other hand, if $\boldsymbol{v}(t, \boldsymbol{x}; \theta)$ is Lipschitz continuous with constant $L \geq 0$, then for any $\boldsymbol{x}, \boldsymbol{y} \in \mathbb{R}^d$,

$$|\boldsymbol{v}(t, \boldsymbol{x}; \theta) - \boldsymbol{v}(t, \boldsymbol{y}; \theta)| \leq L|\boldsymbol{x} - \boldsymbol{y}|, \tag{40}$$

By Grönwall's inequality, the corresponding flow satisfies

$$|\boldsymbol{\phi}_t(\boldsymbol{x}) - \boldsymbol{\phi}_t(\boldsymbol{y})| = \left| \boldsymbol{x} - \boldsymbol{y} + \int_0^t \boldsymbol{v}[s, \boldsymbol{x}(s); \theta] - \boldsymbol{v}[s, \boldsymbol{y}(s); \theta] \mathrm{d}s \right|$$

$$\leq |\boldsymbol{x} - \boldsymbol{y}| + \int_0^t |\boldsymbol{v}[s, \boldsymbol{x}(s); \theta] - \boldsymbol{v}[s, \boldsymbol{y}(s); \theta]| \mathrm{d}s \tag{41}$$

$$\leq |\boldsymbol{x} - \boldsymbol{y}| + \int_0^t L|\boldsymbol{\phi}_s(\boldsymbol{x}) - \boldsymbol{\phi}_s(\boldsymbol{y})| \mathrm{d}s.$$

Thus,

$$|\boldsymbol{\phi}_T(\boldsymbol{x}) - \boldsymbol{\phi}_T(\boldsymbol{y})| \leq \mathrm{e}^{LT}|\boldsymbol{x} - \boldsymbol{y}|, \tag{42}$$

holds for any $\boldsymbol{x}, \boldsymbol{y} \in \mathbb{R}^d$. Now, choose $\epsilon < \frac{d}{\mathrm{e}^{LT}}$. Then from (39), there exist $\boldsymbol{x}_s, \tilde{\boldsymbol{x}}_s \in U_m^0$ such that

$$|\boldsymbol{\phi}_T(\boldsymbol{x}_s) - \boldsymbol{\phi}_T(\tilde{\boldsymbol{x}}_s)| > \frac{d}{\epsilon}|\boldsymbol{x}_s - \tilde{\boldsymbol{x}}_s| > \mathrm{e}^{LT}|\boldsymbol{x}_s - \tilde{\boldsymbol{x}}_s|, \tag{43}$$

which contradicts the Lipschitz bound.

Therefore, any parameterized vector field realizing such a transport map cannot be Lipschitz continuous. This completes the proof. $\square$

**Proposition 4.1.** *The marginal vector field $\boldsymbol{u}(t, \boldsymbol{x}, \boldsymbol{x}_\tau)$ given in Eq. (15), together with the selected initial function $\boldsymbol{\psi}$, generates the probability path $p(\boldsymbol{x}, t|\boldsymbol{\psi})$ and the joint probability path $p(\boldsymbol{x}, t; \boldsymbol{x}_\tau, t - \tau|\boldsymbol{\psi})$ in Eq. (14).*

*Proof of Proposition 4.1.* Given that $\boldsymbol{u}(t, \boldsymbol{x}, \boldsymbol{x}_\tau|\boldsymbol{z})$ is the conditional vector field that generates $p(\boldsymbol{x}, t|\boldsymbol{z}, \boldsymbol{\psi})$ and $p(\boldsymbol{x}, t; \boldsymbol{x}_\tau, t - \tau|\boldsymbol{z}, \boldsymbol{\psi})$, it is evident that the following delay Fokker-Planck equation holds:

$$\frac{\partial p(\boldsymbol{x}, t|\boldsymbol{z}, \boldsymbol{\psi})}{\partial t} = -\nabla \cdot \left\{ \int \mathrm{d}\boldsymbol{x}_\tau [\boldsymbol{u}(t, \boldsymbol{x}, \boldsymbol{x}_\tau|\boldsymbol{z}) p(\boldsymbol{x}, t; \boldsymbol{x}_\tau, t - \tau|\boldsymbol{z}, \boldsymbol{\psi})] \right\}. \tag{44}$$

To verify this proposition, we only need to check that the marginal vector field $\boldsymbol{u}(t, \boldsymbol{x}, \boldsymbol{x}_\tau)$ given by Eq. (15), the marginal probability density path $p(\boldsymbol{x}, t|\boldsymbol{\psi})$ and the joint probability density $p(\boldsymbol{x}, t; \boldsymbol{x}_\tau, t - \tau|\boldsymbol{\psi})$ satisfy the following delay Fokker-Planck equation:

$$\frac{\partial p(\boldsymbol{x}, t|\boldsymbol{\psi})}{\partial t} = -\nabla \cdot \left\{ \int \mathrm{d}\boldsymbol{x}_\tau [\boldsymbol{u}(t, \boldsymbol{x}, \boldsymbol{x}_\tau) p(\boldsymbol{x}, t; \boldsymbol{x}_\tau, t - \tau|\boldsymbol{\psi})] \right\}. \tag{45}$$

Assuming that all the functions involved satisfy the regularity conditions necessary for the interchange of integration and differentiation, we have that

$$\frac{\partial p(\boldsymbol{x}, t|\boldsymbol{\psi})}{\partial t}$$
$$= \frac{\partial}{\partial t} \int p(\boldsymbol{x}, t|\boldsymbol{z}, \boldsymbol{\psi}) q(\boldsymbol{z}) \mathrm{d}\boldsymbol{z}$$
$$= \int \left[ \frac{\partial}{\partial t} p(\boldsymbol{x}, t|\boldsymbol{z}, \boldsymbol{\psi}) \right] q(\boldsymbol{z}) \mathrm{d}\boldsymbol{z}$$
$$= -\int \left( \nabla \cdot \left\{ \int \mathrm{d}\boldsymbol{x}_\tau [\boldsymbol{u}(t, \boldsymbol{x}, \boldsymbol{x}_\tau|\boldsymbol{z}) p(\boldsymbol{x}, t; \boldsymbol{x}_\tau, t - \tau|\boldsymbol{z}, \boldsymbol{\psi})] \right\} \right) q(\boldsymbol{z}) \mathrm{d}\boldsymbol{z}$$
$$= -\nabla \cdot \left\{ \int \mathrm{d}\boldsymbol{x}_\tau \left[ \int \boldsymbol{u}(t, \boldsymbol{x}, \boldsymbol{x}_\tau|\boldsymbol{z}) p(\boldsymbol{x}, t; \boldsymbol{x}_\tau, t - \tau|\boldsymbol{z}, \boldsymbol{\psi}) q(\boldsymbol{z}) \mathrm{d}\boldsymbol{z} \right] \right\}$$
$$= -\nabla \cdot \left\{ \int \mathrm{d}\boldsymbol{x}_\tau \left[ \int \frac{\boldsymbol{u}(t, \boldsymbol{x}, \boldsymbol{x}_\tau|\boldsymbol{z}) p(\boldsymbol{x}, t; \boldsymbol{x}_\tau, t - \tau|\boldsymbol{z}, \boldsymbol{\psi})}{p(\boldsymbol{x}, t; \boldsymbol{x}_\tau, t - \tau|\boldsymbol{\psi})} q(\boldsymbol{z}) \mathrm{d}\boldsymbol{z} \cdot p(\boldsymbol{x}, t; \boldsymbol{x}_\tau, t - \tau|\boldsymbol{\psi}) \right] \right\}$$
$$= -\nabla \cdot \left\{ \int \mathrm{d}\boldsymbol{x}_\tau [\boldsymbol{u}(t, \boldsymbol{x}, \boldsymbol{x}_\tau) p(\boldsymbol{x}, t; \boldsymbol{x}_\tau, t - \tau|\boldsymbol{\psi})] \right\}. \tag{46}$$

$\square$

**Proposition 4.2.** *Given that $p(\boldsymbol{x}, t; \boldsymbol{x}_\tau, t - \tau|\boldsymbol{\psi}) > 0$ for all $\boldsymbol{x}, \boldsymbol{x}_\tau \in \mathbb{R}^d$ and $t \in [0, 1]$, up to a constant independent of $\boldsymbol{\theta}$, $\mathcal{L}_{DCFM}(\boldsymbol{\theta})$ and $\mathcal{L}_{DFM}(\boldsymbol{\theta})$ are equal, which further implies that $\nabla_{\boldsymbol{\theta}} \mathcal{L}_{DCFM}(\boldsymbol{\theta}) = \nabla_{\boldsymbol{\theta}} \mathcal{L}_{DFM}(\boldsymbol{\theta})$.*

*Proof of Proposition 4.2.* To guarantee the existence of all integrals and the validity of changing the order of integration (as justified by Fubini's Theorem), we assume that $p(\boldsymbol{x}, t|\boldsymbol{z}, \boldsymbol{\psi})$, $p(\boldsymbol{x}, t; \boldsymbol{x}_\tau, t - \tau|\boldsymbol{z}, \boldsymbol{\psi})$ decay to zero sufficiently fast as $\|\boldsymbol{x}\| \to \infty$ and $\|\boldsymbol{x}_\tau\| \to \infty$, and further assume that $\boldsymbol{u}(t, \boldsymbol{x}, \boldsymbol{x}_\tau|\boldsymbol{z}), \boldsymbol{v}(t, \boldsymbol{x}, \boldsymbol{x}_\tau; \theta), \nabla_{\boldsymbol{\theta}} \boldsymbol{v}(t, \boldsymbol{x}, \boldsymbol{x}_\tau; \theta)$ are bounded. Note that $t \sim \mathcal{U}(0, 1)$ and $\boldsymbol{\psi} \sim q^\circ(\boldsymbol{\psi})$ are both independent of $\boldsymbol{\theta}$, so we fixed $t$ and $\boldsymbol{\psi}$ in the following analysis. Using the bilinearity of the Euclidean norm and the independence of $\boldsymbol{u}(t, \boldsymbol{x}, \boldsymbol{x}_\tau|\boldsymbol{z})$ from $\boldsymbol{\theta}$, we have:

$$\nabla_{\boldsymbol{\theta}} \mathbb{E}_{p(\boldsymbol{x}, t; \boldsymbol{x}_\tau, t - \tau|\boldsymbol{\psi})} \|\boldsymbol{v}(t, \boldsymbol{x}, \boldsymbol{x}_\tau; \theta) - \boldsymbol{u}(t, \boldsymbol{x}, \boldsymbol{x}_\tau)\|^2$$
$$= \nabla_{\boldsymbol{\theta}} \mathbb{E}_{p(\boldsymbol{x}, t; \boldsymbol{x}_\tau, t - \tau|\boldsymbol{\psi})} \left( \|\boldsymbol{v}(t, \boldsymbol{x}, \boldsymbol{x}_\tau; \theta)\|^2 - 2 \langle \boldsymbol{v}(t, \boldsymbol{x}, \boldsymbol{x}_\tau; \theta), \boldsymbol{u}(t, \boldsymbol{x}, \boldsymbol{x}_\tau) \rangle + \|\boldsymbol{u}(t, \boldsymbol{x}, \boldsymbol{x}_\tau)\|^2 \right) \tag{47}$$
$$= \nabla_{\boldsymbol{\theta}} \mathbb{E}_{p(\boldsymbol{x}, t; \boldsymbol{x}_\tau, t - \tau|\boldsymbol{\psi})} \left( \|\boldsymbol{v}(t, \boldsymbol{x}, \boldsymbol{x}_\tau; \theta)\|^2 - 2 \langle \boldsymbol{v}(t, \boldsymbol{x}, \boldsymbol{x}_\tau; \theta), \boldsymbol{u}(t, \boldsymbol{x}, \boldsymbol{x}_\tau) \rangle \right),$$

and

$$\nabla_{\boldsymbol{\theta}} \mathbb{E}_{q(\boldsymbol{z}), p(\boldsymbol{x}, t; \boldsymbol{x}_\tau, t - \tau|\boldsymbol{z}, \boldsymbol{\psi})} \|\boldsymbol{v}(t, \boldsymbol{x}, \boldsymbol{x}_\tau; \theta) - \boldsymbol{u}(t, \boldsymbol{x}, \boldsymbol{x}_\tau|\boldsymbol{z})\|^2$$
$$= \nabla_{\boldsymbol{\theta}} \mathbb{E}_{q(\boldsymbol{z}), p(\boldsymbol{x}, t; \boldsymbol{x}_\tau, t - \tau|\boldsymbol{z}, \boldsymbol{\psi})} (\|\boldsymbol{v}(t, \boldsymbol{x}, \boldsymbol{x}_\tau; \theta)\|^2 - 2 \langle \boldsymbol{v}(t, \boldsymbol{x}, \boldsymbol{x}_\tau; \theta), \boldsymbol{u}(t, \boldsymbol{x}, \boldsymbol{x}_\tau|\boldsymbol{z}) \rangle +$$
$$\|\boldsymbol{u}(t, \boldsymbol{x}, \boldsymbol{x}_\tau|\boldsymbol{z})\|^2) \tag{48}$$
$$= \nabla_{\boldsymbol{\theta}} \mathbb{E}_{q(\boldsymbol{z}), p(\boldsymbol{x}, t; \boldsymbol{x}_\tau, t - \tau|\boldsymbol{z}, \boldsymbol{\psi})} \left( \|\boldsymbol{v}(t, \boldsymbol{x}, \boldsymbol{x}_\tau; \theta)\|^2 - 2 \langle \boldsymbol{v}(t, \boldsymbol{x}, \boldsymbol{x}_\tau; \theta), \boldsymbol{u}(t, \boldsymbol{x}, \boldsymbol{x}_\tau|\boldsymbol{z}) \rangle \right).$$

Next,

$$\mathbb{E}_{p(\boldsymbol{x},t;\boldsymbol{x}_\tau,t-\tau|\boldsymbol{\psi})}\|\boldsymbol{v}(t,\boldsymbol{x},\boldsymbol{x}_\tau;\theta)\|^2$$

$$= \iint \|\boldsymbol{v}(t,\boldsymbol{x},\boldsymbol{x}_\tau;\theta)\|^2 p(\boldsymbol{x},t;\boldsymbol{x}_\tau,t-\tau|\boldsymbol{\psi})\mathrm{d}\boldsymbol{x}\mathrm{d}\boldsymbol{x}_\tau$$

$$= \iiint \|\boldsymbol{v}(t,\boldsymbol{x},\boldsymbol{x}_\tau;\theta)\|^2 p(\boldsymbol{x},t;\boldsymbol{x}_\tau,t-\tau|\boldsymbol{z},\boldsymbol{\psi})q(\boldsymbol{z})\mathrm{d}\boldsymbol{z}\mathrm{d}\boldsymbol{x}\mathrm{d}\boldsymbol{x}_\tau \qquad (49)$$

$$= \mathbb{E}_{q(\boldsymbol{z}),p(\boldsymbol{x},t;\boldsymbol{x}_\tau,t-\tau|\boldsymbol{z},\boldsymbol{\psi})}\|\boldsymbol{v}(t,\boldsymbol{x},\boldsymbol{x}_\tau;\theta)\|^2.$$

Finally,

$$\mathbb{E}_{p(\boldsymbol{x},t;\boldsymbol{x}_\tau,t-\tau|\boldsymbol{\psi})}\langle\boldsymbol{v}(t,\boldsymbol{x},\boldsymbol{x}_\tau;\theta),\boldsymbol{u}(t,\boldsymbol{x},\boldsymbol{x}_\tau)\rangle$$

$$= \iint \left\langle \boldsymbol{v}(t,\boldsymbol{x},\boldsymbol{x}_\tau;\theta), \frac{\int \boldsymbol{u}(t,\boldsymbol{x},\boldsymbol{x}_\tau|\boldsymbol{z})p(\boldsymbol{x},t;\boldsymbol{x}_\tau,t-\tau|\boldsymbol{z},\boldsymbol{\psi})q(\boldsymbol{z})\mathrm{d}\boldsymbol{z}}{p(\boldsymbol{x},t;\boldsymbol{x}_\tau,t-\tau|\boldsymbol{\psi})} \right\rangle p(\boldsymbol{x},t;\boldsymbol{x}_\tau,t-\tau|\boldsymbol{\psi})\mathrm{d}\boldsymbol{x}\mathrm{d}\boldsymbol{x}_\tau$$

$$= \iint \left\langle \boldsymbol{v}(t,\boldsymbol{x},\boldsymbol{x}_\tau;\theta), \int \boldsymbol{u}(t,\boldsymbol{x},\boldsymbol{x}_\tau|\boldsymbol{z})p(\boldsymbol{x},t;\boldsymbol{x}_\tau,t-\tau|\boldsymbol{z},\boldsymbol{\psi})q(\boldsymbol{z})\mathrm{d}\boldsymbol{z} \right\rangle \mathrm{d}\boldsymbol{x}\mathrm{d}\boldsymbol{x}_\tau$$

$$= \iiint \langle\boldsymbol{v}(t,\boldsymbol{x},\boldsymbol{x}_\tau;\theta),\boldsymbol{u}(t,\boldsymbol{x},\boldsymbol{x}_\tau|\boldsymbol{z})\rangle p(\boldsymbol{x},t;\boldsymbol{x}_\tau,t-\tau|\boldsymbol{z},\boldsymbol{\psi})q(\boldsymbol{z})\mathrm{d}\boldsymbol{z}\mathrm{d}\boldsymbol{x}\mathrm{d}\boldsymbol{x}_\tau$$

$$= \mathbb{E}_{q(\boldsymbol{z}),p(\boldsymbol{x},t;\boldsymbol{x}_\tau,t-\tau|\boldsymbol{z},\boldsymbol{\psi})}\langle\boldsymbol{v}(t,\boldsymbol{x},\boldsymbol{x}_\tau;\theta),\boldsymbol{u}(t,\boldsymbol{x},\boldsymbol{x}_\tau|\boldsymbol{z})\rangle.$$

$$(50)$$

From Eq. (49) and Eq. (50), it follows that Eq. (47) is equal to Eq. (48) for any latent variable $\boldsymbol{z}$, which can be further deduced that $\nabla_{\boldsymbol{\theta}}\mathcal{L}_{\mathrm{DCFM}}(\boldsymbol{\theta}) = \nabla_{\boldsymbol{\theta}}\mathcal{L}_{\mathrm{DFM}}(\boldsymbol{\theta})$. $\qquad\square$

**Proposition 4.3** (Universal approximating capability of DDE's flow map). *For any given $\epsilon > 0$ and continuous transport map $F : \mathbb{R}^d \to \mathbb{R}^d$, which push-forward the source distribution $q_0$ to the target distribution $q_1$, i.e. $F_{\#}q_0 = q_1$, if there exists a neural network $\boldsymbol{f}(\boldsymbol{x};\theta)$ such that $||\boldsymbol{f}(\boldsymbol{x};\theta) - [F(\boldsymbol{x}) - \boldsymbol{x}]|| < \epsilon$, for all $\boldsymbol{x} \in \mathbb{R}^d$, then we can construct a vector field with a single delay term $\boldsymbol{v}(t,\boldsymbol{x},\boldsymbol{x}_\tau;\theta)$, where the corresponding flow map $G(\boldsymbol{x};\theta)$, under the constant initial function condition, satisfies $||G(\boldsymbol{x};\theta) - F(\boldsymbol{x})|| < \epsilon$, for all $\boldsymbol{x} \in \mathbb{R}^d$.*

*Proof of Proposition 4.3.* The proof is straightforward. We consider the following DDE with a constant initial function and a parameterized vector field with a single delay term:

$$\frac{\mathrm{d}\boldsymbol{x}(t)}{\mathrm{d}t} = \boldsymbol{v}[\boldsymbol{x}(t-\tau);\theta], \quad t \in [0,1],$$

$$\boldsymbol{x}(t) \equiv \boldsymbol{x}_0, \quad t \in [-\tau,0], \qquad (51)$$

where the time delay $\tau = 1$, and the vector field depends solely on the delay term, independent of the current state and time, which is a special degenerate case of DFM. In this case, for any point sampled from the initial distribution $\boldsymbol{x}_0 \sim q_0$, the corresponding vector field $\boldsymbol{v}[\boldsymbol{x}(t-\tau);\theta] = \boldsymbol{v}(\boldsymbol{x}_0;\theta)$ remains constant over the time interval $t \in [0,1]$, which means that the flow map $G : \mathbb{R}^d \to \mathbb{R}^d$ associated with the above DDE will map $\boldsymbol{x}_0$ to $\boldsymbol{x}_0 + \boldsymbol{v}[\boldsymbol{x}_0;\theta] \cdot 1$. For any given $\epsilon > 0$, suppose there exists a neural network $\boldsymbol{f}(\boldsymbol{x};\theta)$ such that $||\boldsymbol{f}(\boldsymbol{x};\theta) - [F(\boldsymbol{x}) - \boldsymbol{x}]|| < \epsilon$, for all $\boldsymbol{x} \in \mathbb{R}^d$. Let $\boldsymbol{v}(\boldsymbol{x}_0;\theta) = \boldsymbol{f}(\boldsymbol{x}_0;\theta)$. Then it follows that $||G(\boldsymbol{x}_0;\theta) - F(\boldsymbol{x}_0)|| = ||\boldsymbol{x}_0 + \boldsymbol{v}(\boldsymbol{x}_0;\theta) \cdot 1 - F(\boldsymbol{x}_0)|| \le ||\boldsymbol{v}(\boldsymbol{x}_0;\theta) - \boldsymbol{f}(\boldsymbol{x}_0;\theta)|| + ||\boldsymbol{f}(\boldsymbol{x}_0;\theta) - [F(\boldsymbol{x}_0) - \boldsymbol{x}_0]|| < \epsilon$. This implies that the flow map of the Neural DDEs approximately peserves the target transport strategy, and naturally, it can approximately push-forward $q_0$ to $q_1$. $\qquad\square$

## C  ALGORITHMS

To more clearly illustrate the overall workflow of our proposed DFM, we provide pseudocode for different DFM methods: Algorithm 1 for DFM(C) and Algorithm 2 for DFM(D).

## D  EXPERIMENTAL SETUP DETAILS AND ADDITIONAL RESULTS

In this section, we provide a detailed explanation of the experimental settings for different datasets described in the main text and present additional experimental results. The experimental details for the delay dynamical systems and single-cell datasets are summarized in Table 5).

---

**Algorithm 1** `Delay Flow Matching with Constant Initial Functions`

```
# Input:  Data={X_0, X_1}, the source data X_0 ∼ q_0 and the target data
X_1 ∼ q_1; Time delay τ
# Output:  Model v(t, x, x_τ; θ) for the vector field with a single
delay term x_τ = x(t − τ)
Initialize Model:  v(t, x, x_τ; θ) # Arbitrary neural network structure
Select Coupling Strategy:  KPG-OT/OT/Independent
Select Interpolation Method:  Linear/Geodesic/CSpline # Consider
the linear interpolation below
Initialize Initial Function:  ψ*(t; x_0) ≡ x_0 for t ∈ [−τ, 0]
While not converge:
    Sample x_0 ⊂ X_0, x_1 ⊂ X_1 # Generate batch data
    Optimizer.zero_grad()
    x_0, x_1 = Coupling_Strategy(x_0, x_1) # Construct coupling
    t = torch.rand(batchsize)  # Randomly sample t ∈ [0, 1]
    x(t) = (1 − t) × x_0 + t × x_1 # Sample state at t based on the linear
interpolation
    x(t − τ) = (1 − t + τ) × x_0 + (t − τ) × x_1 for t ∈ [τ, 1] # Sample delay state
at t − τ based on the linear interpolation
    x(t − τ) = x_0 for t ∈ [0, τ) # Sample delay state at t − τ based on
the constant initial function
    Loss = { v[t, x(t), x(t − τ); θ] − (x_1 − x_0) }.pow(2).mean()
    Loss.backward()
    Optimizer.step()
return Model
```

---

Table 5: The experimental setup details for delay dynamical systems and single-cell datasets.

| | Setup | Autoregulation | Spiral DDE | Mouse hematopoiesis | iPSCs |
|---|---|---|---|---|---|
| Data | Dimension | 1 | 2 | 2 | 4 |
| Structure | Hidden layer | 3 | 3 | 3 | 3 |
| | Hidden neuron | 64 | 64 | 64 | 64 |
| | Activation | Tanh | Tanh | SELU | SELU |
| | Time input | False | False | True | True |
| | Delay term input | True (DFM) / False (CFM) | True (DFM) / False (CFM) | True (DFM) / False (CFM) | True (DFM) / False (CFM) |
| Training | Coupling | KPG-OT | OT | OT | OT |
| | Initial function (DFM) | Constant | Constant | Diverse | Diverse |
| | Latent variable $\gamma$ | CSpline | CSpline | Geodesic | CSpline |
| | Batch size | 256 | 256 | 128 | 128 |
| | Iteration | 2k | 2k | 10k | 2k |
| | Optimizer | Adam | Adam | Adam | Adam |
| | Learning rate | $10^{-3}$ | $10^{-3}$ | $10^{-3}$ | $10^{-3}$ |
| Results | Figures & tables | Figs. 3 & Tab. 6, 7, 8 | Figs. 3 & Tab.9, 10 | Fig. 4 & Tab. 2 | Fig. 4 & Tab. 2 |

## D.1 BIOLOGICAL AUTOREGULATION MOTIF

**Dataset generation**. The autoregulation motif, which is one of the most common biological network motifs, can be modeled as the following 1-d DDE:

$$\dot{x}(t) = \frac{\eta}{1 + x^n(t − \tau)}, \tag{52}$$

where $\tau = 1$, $n = 2$, $\eta = 5$. As shown in Fig. 3, the dynamics exhibit damped oscillations. We sample $1,000$ initial points from the uniform distribution $\mathcal{U}(0.2, 1.2)$ and perform forward integration based on constant initial functions $\psi(h; x_0) = x_0, h \in [−1, 0]$, to obtain trajectories. After that, we select snapshots at $t = 0, 1, 2, \cdots, 7$ as the training dataset.

**Details for training**. For each training iteration, we randomly and independently sample 256 data points from each snapshot. Pairing between minibatch data points from adjacent time steps is performed using KPG-OT. For this task, we utilize the true coupling of only two keypoint pairs between adjacent time steps as guidance, although incorporating more keypoint pairs would enable more accurate pairing. Once data points across time steps are fully matched, we construct 256

---

**Algorithm 2** `Delay Flow Matching with Diverse Initial Functions`

```
# Input:  Data={X₀,X₁}, the source data X₀∼q₀ and the target data
X₁∼q₁; Time delay τ
# Output:  Model v(t,x,xτ;θ) for the vector field with a single
delay term xτ = x(t − τ)
Initialize Model:  v(t,x,xτ;θ) # Arbitrary neural network structure
Select Coupling Strategy:  KPG-OT/OT/Independent
Select Interpolation Method:  Linear/Geodesic/CSpline # Consider
the linear interpolation below
Select Clustering Method:  Gaussian Mixture Model/DBSCAN
Cluster the source and target dataset:
    X₀⁽¹⁾,...,X₀⁽ᴹ⁾ = Clustering_Method(X₀)
    X₁⁽¹⁾,...,X₁⁽ᴺ⁾ = Clustering_Method(X₁)
Initialize Diverse Initial Functions for Different Clusters:
    Select different constants Cₘₙ, for x₀ ∈ X₀⁽ᵐ⁾, x₁ ∈ X₁⁽ⁿ⁾:
        ψ*ₘₙ(t;x₀) = x₀ + Cₘₙ × t for t ∈ [−τ,0]
While not converge:
    Sample x₀ ⊂ X₀, x₁ ⊂ X₁ # Generate batch data
    Optimizer.zero_grad()
    x₀,x₁ = Coupling_Strategy(x₀, x₁) # Construct coupling
    Identify the cluster membership of x₀:  Iₓ₀
    Identify the cluster membership of x₁:  Iₓ₁
    t = torch.rand(batchsize)  # Randomly sample t ∈ [0,1]
    x(t) = (1 − t) × x₀ + t × x₁ # Sample state at t based on the linear
interpolation
    x(t − τ) = (1 − t + τ) × x₀ + (t − τ) × x₁ for t ∈ [τ,1] # Sample delay state
at t − τ based on the linear interpolation
    x(t − τ) = x₀ + C_{Iₓ₀Iₓ₁} × (t − τ) for t ∈ [0,τ) # Sample delay state at
t − τ based on the diverse initial functions
    Loss = { v[t,x(t),x(t − τ);θ]−(x₁−x₀) }.pow(2).mean()
    Loss.backward()
    Optimizer.step()
return Model
```

---

transport trajectories using cubic spline (CSpline) interpolation. These trajectories provide the states, delayed states, and vector fields at various time steps, enabling the computation of the training objective Eq. (16), which is then optimized via backpropagation. Further details on the experimental setup can be found in Table 5.

**Details for testing**. After training, we evaluate the performance of KP-DFM(C) and KP-CFM on both interpolation and extrapolation tasks. Specifically, we randomly select 100 initial points from the initial distribution $\mathcal{U}(0.2, 1.2)$ and generate predicted trajectories by performing forward integration on the learned vector field, obtaining the predicted trajectories and snapshots at various time steps. For interpolation, we compute the mean 2-Wasserstein distance $W_2$ between the predicted and ground-truth distributions at $t = 0.5, 1.5, 2.5, \cdots, 6.5$. Additionally, since the true dynamics are known, we calculate the error between the predicted and ground-truth trajectories at these points, quantified by the Mean Squared Error (MSE) and Mean Relative Error (MRE). For extrapolation, we similarly compute the $W_2$ distance between the predicted and ground-truth distributions at $t = 8, 9, 10$, along with the trajectory-wise error using MSE and MRE.

**Results**. As shown in Table 7 and 8, KP-DFM(C) significantly outperforms CFM in both interpolation and extrapolation tasks, achieving much lower prediction errors for both distributions and individual trajectories, demonstrating superior representational capability. Furthermore, as illustrated in Table 6, CFM fails to reconstruct the underlying dynamics, whereas KP-DFM(C) successfully captures the damped oscillatory dynamics, even when the time delay $\tau$ deviates from the ground truth.

Table 6: Additional results on biological autoregulation motif based on CFM with keypoint-guided coupling (KP-CFM), as well as DFM with constant initial functions and keypoint-guided coupling (KP-DFM(C)). For DFM, the generation results are illustrated with various time delays, specifically $\tau = 0.6, 0.8, 1.0$ (ground truth), $1.2, 1.4$.

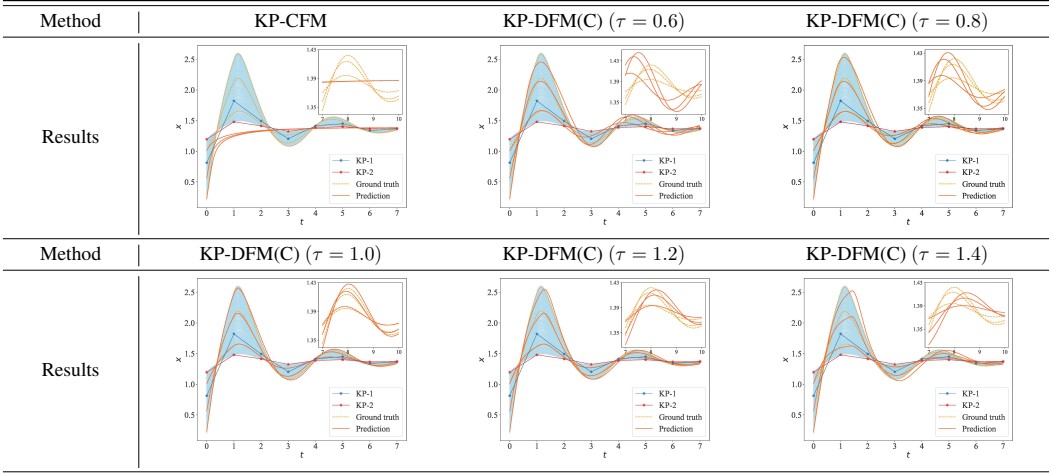

Table 7: Comparison of the interpolation prediction error between KP-CFM and KP-DFM(C) with different time delay $\tau$ on the biological autoregulation dataset.

| $\tau$ | 0(CFM) | 0.6 | 0.8 | 1.0 | 1.2 | 1.4 |
|---|---|---|---|---|---|---|
| $W_2$ | 0.1842 | 2.82e-2 | 1.55e-2 | **1.53e-2** | 2.27e-2 | 4.37e-2 |
| MSE | 9.21e-2 | 1.51e-3 | 4.42e-4 | **3.45e-4** | 5.36e-4 | 2.68e-3 |
| MRE | 0.117 | 2.22e-2 | 1.23e-2 | **9.47e-3** | 1.14e-2 | 2.57e-2 |

## D.2 SPIRAL DDE

**Dataset generation**. We consider the following 2-d DDE:

$$\dot{\boldsymbol{x}}(t) = \boldsymbol{A}\tanh[\boldsymbol{x}(t) + \boldsymbol{x}(t - \tau)], \tag{53}$$

where $\tau = 0.5$ and $\boldsymbol{A} = [[-1, 20], [-30, -1]] \in \mathbb{R}^{2 \times 2}$. As shown in Fig. 3, the dynamics exhibit trajectory crossings. We sample $1,000$ initial points from the uniform distribution $\mathcal{U}(0.8, 1) \times \mathcal{U}(0, 0.1)$ and perform forward integration based on constant initial functions $\boldsymbol{\psi}(h; \boldsymbol{x}_0) = \boldsymbol{x}_0, h \in [-0.5, 0]$, to obtain trajectories. After that, snapshots at intervals of $0.05$ within $t \in [0, 0.6]$ are selected as training dataset.

**Details for training**. In each training iteration, 256 data points are randomly and independently sampled from each snapshot. Pairing between the minibatch data points at adjacent time steps is performed using OT. After matching data points across time steps, 256 transport trajectories are

Table 8: Comparison of the extrapolation prediction error between KP-CFM and KP-DFM with different time delay $\tau$ on the biological autoregulation dataset.

| $\tau$ | 0(CFM) | 0.6 | 0.8 | 1.0 | 1.2 | 1.4 |
|---|---|---|---|---|---|---|
| $W_2$ | 2.13e-2 | 2.25e-2 | 8.67e-3 | **3.42e-3** | 4.55e-3 | 1.23e-2 |
| MSE | 5.51e-4 | 6.31e-4 | 1.28e-4 | **1.46e-5** | 4.38e-5 | 2.13e-4 |
| MRE | 1.59e-2 | 1.68e-2 | 6.75e-3 | **2.26e-3** | 3.72e-3 | 9.37e-3 |

constructed using CSpline interpolation. These trajectories provide the states, delayed states, and vector fields at various time steps, enabling the computation of the training objective in Eq. (16), which is subsequently optimized via backpropagation. Additional details on the experimental setup are provided in Table 5.

**Details for testing**. After training, we assess the performance of OT-DFM(C) and OT-CFM on the interpolation task. Specifically, $100$ initial points are randomly sampled from the initial distribution $\mathcal{U}(0.8, 1) \times \mathcal{U}(0, 0.1)$, and predicted trajectories are generated via forward integration of the learned vector field, yielding trajectories and snapshots at various time steps. The mean $W_2$ distance between the predicted and ground-truth distributions is computed at intervals of $0.01$ within $t \in [0, 0.6]$. Moreover, we calculate the error between the predicted and ground-truth trajectories at these points, quantified by MSE and MRE.

**Results**. As shown in Table 10, OT-DFM(C) significantly outperforms CFM in the interpolation prediction task, achieving substantially lower errors for prediction of both distributions and individual trajectories. Additionally, as illustrated in Table 9, CFM collapses around the trajectory crossing area, while OT-DFM(C) successfully captures the spiral dynamics, even when the time delay $\tau$ deviates from the ground truth.

Table 9: Additional results on Spiral DDE based on CFM with OT coupling (OT-CFM), as well as DFM with constant initial functions and OT coupling (OT-DFM(C)). For DFM, the generation results are illustrated with various time delays, specifically $\tau = 0.35, 0.40, 0.45, 0.50$ (ground truth), $0.55$.

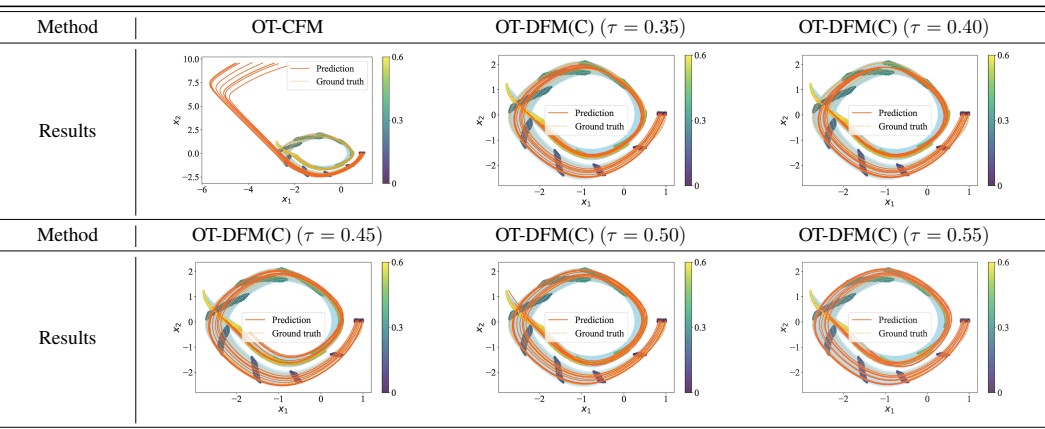

Table 10: Comparison of the interpolation prediction error between OT-CFM and OT-DFM with different time delay $\tau$ on the spiral DDE dataset.

| $\tau$ | 0(CFM) | 0.35 | 0.40 | 0.45 | 0.50 | 0.55 |
|---|---|---|---|---|---|---|
| $W_2$ | 3.95 | 0.33 | 0.43 | 0.42 | **0.20** | 0.45 |
| MSE | 15.81 | 0.07 | 0.13 | 0.12 | **0.03** | 0.14 |
| MRE | 13.14 | 1.25 | 1.66 | 1.80 | **0.68** | 1.96 |

### D.3 MOUSE HEMATOPOIESIS DATASET

**Data preprocessing**. To efficiently apply various trajectory inference methods to scRNA-seq data from mouse hematopoiesis, we first project the data into a low-dimensional space. Specifically, we use the reduced two force-directed layouts (SPRING) space for this dataset with batch correction, following the same preprocessing pipeline as TIGON (Sha et al., 2024). Besides, we use the DBSCAN clustering algorithm to categorize all cells at Day 6 into two distinct clusters, corresponding to the Neu fate and Mo fate, respectively.

**Selection of initial functions**. During training, we match the sampled minibatch cells at Day 2 and 6 using OT. For each matched cell pairs $(\boldsymbol{x}_0, \boldsymbol{x}_1)$, where $\boldsymbol{x}_0$ is sampled from data of Day 2 and $\boldsymbol{x}_1$ is sampled from data of Day 6, the initial function is defined as:

$$\frac{\mathrm{d}\boldsymbol{\psi}^*(t; \boldsymbol{x}_0)}{\mathrm{d}t} = \begin{cases} [1, 0], & \text{if } \boldsymbol{x}_1 \text{ is in Neu cluster,} \\ [0, 1], & \text{if } \boldsymbol{x}_1 \text{ is in Mo cluster.} \end{cases} \tag{54}$$

**Details of training.** During training, we align Day 2 and Day 6 with $t = 0$ and $t = 1$, respectively. For testing, the distribution of Day 4 is predicted at $t = 0.5$. In this task, the time delay is set to $\tau = 1$. In each training iteration, we randomly and independently sample 128 cells from each snapshot at Day 2 and 6. Pairing between minibatch data points at adjacent time steps is performed using OT. After matching data points across time steps, 128 transport trajectories $\gamma$ are constructed using geodesic interpolation (refer to the next paragraph). These trajectories provide the states, delayed states, and vector fields at various time steps, enabling the computation of the training objective in Eq. (16), which is subsequently optimized via backpropagation. Additional details on the experimental setup are provided in Table 5.

**Construct geodesic interpolation** $\gamma$. For each matched cell pairs, geodesic interpolation is employed to generate the interpolation trajectory (Kapuśniak et al., 2024). Here, we first build a data-induced Riemannian metric $g$ on the data manifold $\mathcal{M}$. Formally, a Riemannian metric $g$ on $\mathcal{M}$ is a smooth family of inner products on the tangent spaces of $\mathcal{M}$. Specifically, $g$ provides each $\boldsymbol{x} \in \mathcal{M}$ a positive defined symmetric bilinear form on $T_{\boldsymbol{x}}\mathcal{M}$,

$$g_{\boldsymbol{x}} : T_{\boldsymbol{x}}\mathcal{M} \times T_{\boldsymbol{x}}\mathcal{M} \to \mathbb{R}, \tag{55}$$

which induces a norm $|| \cdot ||_{g_{\boldsymbol{x}}} : T_{\boldsymbol{x}}\mathcal{M} \to \mathbb{R}$ defined by $||v||_{g_{\boldsymbol{x}}} = \sqrt{g_{\boldsymbol{x}}(v, v)}$. Based on this norm, we can calculate the length of any path connecting two points on the manifold and construct the geodesic between them. We aim to equip the ambient space $\mathbb{R}^d$ with an appropriate metric that constrains geodesics to remain close to the data manifold. Specifically, given coordinates, the Riemannian metric $g$ can be equivalently represented as a state-dependent $d \times d$ symmetric positive definite matrix $\mathbf{G}(\boldsymbol{x})$, such that $g_{\boldsymbol{x}}(u, v) = u^\top \mathbf{G}(\boldsymbol{x})v$ for any $u, v \in T_{\boldsymbol{x}}\mathcal{M}$. Intuitively, to impose manifold constraints on the transport path, $||\mathbf{G}(\boldsymbol{x})||$ is supposed to take smaller values around data points and larger values in regions far from all points in the dataset $\mathcal{D}$. Therefore, we employ the LAND metric, where $\mathbf{G}(\boldsymbol{x}; \mathcal{D}) = (\mathrm{diag}(\mathbf{h}(\boldsymbol{x}; \mathcal{D}) + \epsilon\mathbf{I})^{-1}$. Here, the components of $\mathbf{h}(\boldsymbol{x}; \mathcal{D})$ are defined as

$$h_k(\boldsymbol{x}; \mathcal{D}) = \sum_{i=1}^{N} (x_k^i - x_k)^2 \exp\left(-\frac{1}{2\sigma^2}||\boldsymbol{x} - \boldsymbol{x}^i||_2^2\right), \tag{56}$$

where $1 \leq k \leq d$ denotes the $k$-th component, the superscript $i$ represents the $i$-th data point in the training dataset $\mathcal{D}$ and $\sigma$ is the kernel bandwidth. Subsequently, we utilize a neural network $\phi_\eta$ to parameterize the geodesic interpolation path between $\boldsymbol{x}_0$ and $\boldsymbol{x}_1$ as

$$\boldsymbol{\gamma}_\eta(t, \boldsymbol{x}_0, \boldsymbol{x}_1) = (1 - t)\boldsymbol{x}_0 + t\boldsymbol{x}_1 + t(1 - t)\phi_\eta(t, \boldsymbol{x}_0, \boldsymbol{x}_1). \tag{57}$$

The trajectory $\boldsymbol{\gamma}_\eta$ approximates the geodesic between any two points sampled at adjacent time steps on the Riemannian manifold $(\mathcal{M}, g)$ implies learning parameters $\eta$ to minimize the following loss function:

$$\mathcal{L}_{\mathbf{G}}(\eta) := \mathbb{E}_{\pi(\boldsymbol{x}_0, \boldsymbol{x}_1)} \int_0^1 \dot{\boldsymbol{\gamma}}_\eta(t, \boldsymbol{x}_0, \boldsymbol{x}_1)^\top \mathbf{G}(\boldsymbol{\gamma}_{\eta,t}; \mathcal{D})\dot{\boldsymbol{\gamma}}_\eta(t, \boldsymbol{x}_0, \boldsymbol{x}_1)\mathrm{dt}, \tag{58}$$

where $\pi$ represents the OT coupling. The optimal parameter is obtained as:

$$\eta^* = \arg\min_\eta \mathcal{L}_{\mathbf{G}}(\eta), \tag{59}$$

By substituting $\eta^*$ into $\boldsymbol{\gamma}_\eta$ in Eq. (57), the resulting trajectory $\boldsymbol{\gamma}_{\eta^*}$ provides an approximation of the geodesic interpolation on the data manifold.

## D.4 IPSCS DATASET

**Data Preprocessing**. To apply trajectory inference methods to the iPSC scRNA-seq data efficiently, we first reduce the data to a 4-dimensional space using Principal Component Analysis (PCA),

following the preprocessing pipeline of TIGON (Sha et al., 2024). For cells after differentiation (from Day 3), we categorize them into two groups corresponding to the M and En fates using a Gaussian Mixture Model (GMM) with two components. For Day 2, we model the distribution with a single-component GMM and denote its mean as $\boldsymbol{\mu}_0$. As described above, for Day 5, the distribution is modeled with a two-component GMM, where the means of the M and En fates are denoted as $\boldsymbol{\mu}_{11}$ and $\boldsymbol{\mu}_{12}$, respectively.

**Selection of initial functions**. During training, we match the sampled minibatch cells at Day 2, 4 and 5 using OT. For each matched triplet $(\boldsymbol{x}_0, \boldsymbol{x}_1, \boldsymbol{x}_2)$, where $\boldsymbol{x}_0$ is sampled from the Day 2 dataset, $\boldsymbol{x}_1$ from the Day 4 dataset, and $\boldsymbol{x}_2$ from the Day 5 dataset, the initial function is defined as:

$$\frac{\mathrm{d}\boldsymbol{\psi}^*(t; \boldsymbol{x}_0)}{\mathrm{d}t} = \begin{cases} \boldsymbol{\mu}_{11} - \boldsymbol{\mu}_0, & \text{if } \boldsymbol{x}_2 \text{ is in M component}, \\ \boldsymbol{\mu}_{12} - \boldsymbol{\mu}_0, & \text{if } \boldsymbol{x}_2 \text{ is in En component}. \end{cases} \tag{60}$$

**Details of training.** During training, we align Day 2, 4 and 5 with $t = 0$, $t = 2$ and $t = 3$, respectively. For testing, the distribution of Day 3 is predicted at $t = 1$. In this task, the time delay is set to $\tau = 3$. In each training iteration, we randomly and independently sample 128 cells from each snapshot at Day 2, 4, and 5. Pairing between minibatch data points at adjacent time steps is performed using OT. After matching data points across time steps, 128 transport trajectories $\gamma$ are constructed using CSpline interpolation. These trajectories provide the states, delayed states, and vector fields at various time steps, enabling the computation of the training objective in Eq. (16), which is subsequently optimized via backpropagation. Additional details on the experimental setup are provided in Table 5.

## D.5 MNIST DATASET

**Dataset Generation**. We propose a *Semi-paired Image-to-Image Translation* task using the MNIST dataset. The source domain consists of the original MNIST images, while the target domain comprises their corresponding negative images. Specifically, the pixel values $X$ of each image in the original MNIST dataset are inverted by replacing them with $1 - X$, which flips the brightness. The objective is to map each image to its corresponding negative counterpart, as demonstrated in Table 11.

**Minibatch coupling**. To provide partial supervision, $10\%$ of the training data are paired with their negative counterparts as keypoints. During training, minibatches are independently sampled from the source and target distributions, and KPG-OT coupling (KP-) is applied. Specifically, for each batch, we sample the same number of images from the source distribution and the target distribution. Out of these, around $10\%$ image pairs are selected from the keypoint set, while the remaining images are independently sampled from the respective distributions. It is important to note that the negative counterparts of the remaining images sampled from the initial distribution are not guaranteed to appear in the target batch. Subsequently, we perform KPG-OT pairing on the remaining images in the batch based on the selected keypoint pairs, which are then used for further training.

## D.6 CIFAR-10 DATASET

**Source distribution**. The source distribution for $\boldsymbol{x}_0$ is defined as follows: with a probability of $50\%$, $\boldsymbol{x}_0$ is sampled as $\boldsymbol{x}_0 = torch.randn(3, 32, 32)/4 - 0.5$, and with the remaining $50\%$ probability, it is sampled as $x_0 = torch.randn(3, 32, 32)/4 + 0.5$, following the setting in Zhu & Lin (2024). A sample drawn from the source distribution is shown in Table 12.

**Trainable initial functions**. Designing appropriate initial values for high-dimensional images in image generation tasks is a particularly challenging problem. To address this, we employ torch.nn.Embedding to automatically learn the initial functions. Specifically, the source distribution consists of two modules, corresponding to two Gaussian distributions, while the target distribution consists of ten modules, corresponding to ten image categories. Therefore, for each pair of modules $(m, n)$, where $m \in \{0, 1\}$ and $n \in \{0, 1, \cdots, 9\}$, we need to design the time derivative $C_{mn}$ of the initial function (Eq. (22)). To achieve this, we begin by defining Emb=torch.nn.Embedding(20, $3 \times 32 \times 32$), which maps 20 discrete indices to corresponding tensors of the same shape as the image. Then, for each pair $(m, n)$, we define the initial function as:

$$\frac{\mathrm{d}\boldsymbol{\psi}^*_{mn}(t; \boldsymbol{x}_0)}{\mathrm{d}t} = \text{Emb}(10m + n), \ \boldsymbol{\psi}^*_{mn}(0; \boldsymbol{x}_0) = \boldsymbol{x}_0, \tag{61}$$

Table 11: Samples of generation process and results on MNIST data based on CFM with keypoint-guided coupling (KP-CFM), as well as DFM with constant initial functions and keypoint-guided coupling (KP-DFM(C)). For DFM, the generation results are illustrated with various time delays, specifically $\tau = 0.125, 0.25, 0.5$, and $1.0$. In each image, the first and last columns represent samples obtained from the source data and their negative counterparts, respectively, while the 10 intermediate columns depict the generation process.

| Method | KP-CFM | KP-DFM(C) ($\tau = 0.125$) | KP-DFM(C) ($\tau = 0.25$) |
|---|---|---|---|
| Generation |  |  |  |

| Method | KP-CFM | KP-DFM(C) ($\tau = 0.5$) | KP-DFM(C) ($\tau = 1.0$) |
|---|---|---|---|
| Generation |  |  |  |

where $h \in [-\tau, 0]$. This approach allows the derivative of the initial functions across different modules to adaptively adjust during training, thereby enabling more flexible and efficient learning.

**Experimental setup details**. For the image generation tasks, both CFM and DFM are trained following the configurations outlined in Tong et al. (2023a;b). In particular, we utilize a UNet architecture with the following structures and training parameters:

- channels $= 128$,
- depth $= 2$ ,
- channels multiple $= [1, 2, 2, 2]$,
- heads $= 4$,
- heads channels $= 64$,
- attention resolution $= 16$,
- dropout $= 0.1$,
- batch size per gpu $= 128$, gpus $= 1$,
- Adam optimizer with $\beta_1 = 0.9$, $\beta_2 = 0.999$, $\epsilon = 10^{-8}$, and no weight decay,
- learning rate $= 2 \times 10^{-4}$,
- gradient clipping with norm $= 1.0$,
- exponential moving average weights with decay $= 0.9999$.

Table 12: Samples from the source distribution and generation results based on CFM and DFM(D) using independent coupling (I-) or OT coupling (OT-) for adaptive NFE on transporting the Gaussian mixture model with 2 components to the CIFAR-10 dataset.

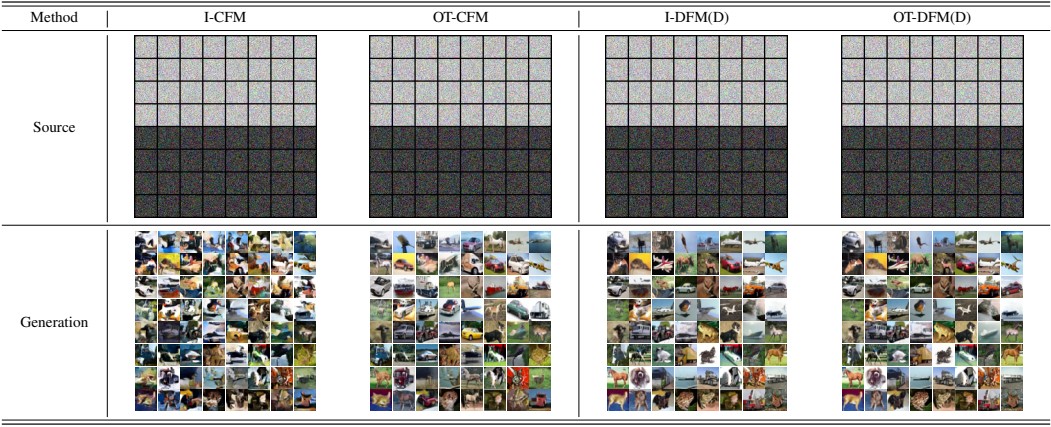

Table 13: Samples from the source distribution and generation results based on CFM and DFM(D) using independent coupling (I-) or OT coupling (OT-) for adaptive NFE on transporting the Gaussian mixture model with 2 components to the CIFAR-10 dataset.

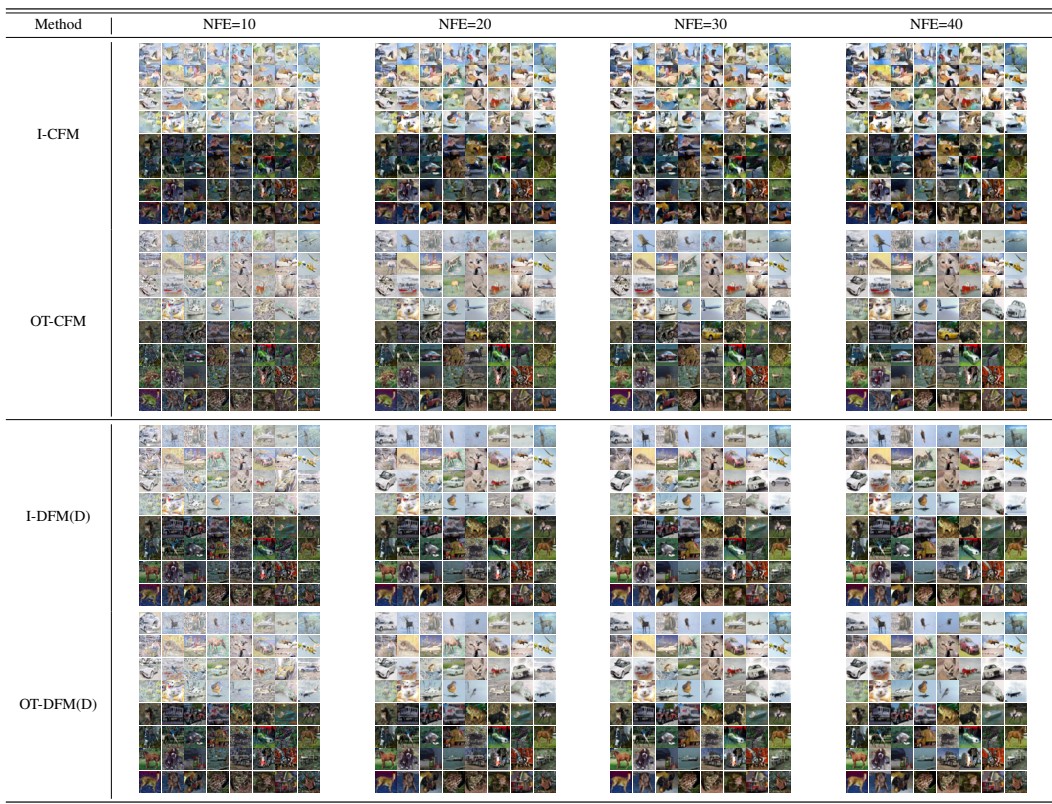

## E  TRAINING AND INFERENCE TIME OF DFM

In this section, we compare the training and inference times of DFM and CFM. Through detailed analysis and illustrative examples, we show that the training cost of DFM is only marginally higher

than that of CFM. In terms of inference efficiency, when facing trajectory intersections or distributional heterogeneity, DFM achieves substantially faster inference because it produces transport paths that are effectively "straighter" than those of CFM.

**Training time**. Similar to CFM, we formulate a regression objective to efficiently train the vector field of DFM (see Eq. (16)). The key difference is that DFM additionally requires specifying the initial functions and sampling delayed states. Note that the initial functions are selected and fixed before training and therefore introduce no additional computational cost. The only extra cost arises from delay sampling: in CFM, given sampled initial and target points, we construct an interpolation path and sample the state $x$ at time $t$. In DFM, after constructing the same interpolation path, we must sample both the state $x$ at time $t$ and the state $x_\tau$ at time $t - \tau$. This additional sampling makes DFM slightly more expensive to train than CFM, though it still remains highly efficient in practice. On the 2-d Gaussian mixture dataset, we record the per-iteration training time of both models. Table 14 reports the mean training time per iteration together with the corresponding standard deviation (over 1,000 iterations). DFM requires a slightly longer training time than CFM, but the difference is very small (DFM: $2.014_{\pm 0.109}$ s/iter vs CFM: $1.986_{\pm 0.118}$ s/iter).

**Inference time**. Compared to CFM, DFM possesses a stronger representational capacity, enabling it to more accurately approximate the target transport map and the associated trajectories. As a result, during inference, DFM typically follows a much "straighter" transport path, leading to significantly improved inference efficiency. This advantage becomes particularly pronounced in scenarios involving trajectory intersections or distributional heterogeneity. We illustrate this difference using two examples. As shown in Fig. 5, on the 2-d Gaussian mixture dataset, the keypoint-guided optimal transport induces trajectory intersections along the transport process. DFM can directly approximate the linear transport paths and therefore requires only a very small number of Euler discretization steps to achieve accurate generation. In contrast, CFM inevitably induces substantial trajectory bending near the crossing regions, necessitating much more Euler steps to obtain reasonable results. To quantify the inference efficiency, we employ an ODE solver with adaptive step size to compute the generation trajectories of both CFM and DFM. As reported in Table 14, DFM requires only $0.0091_{\pm 0.0062}$ s of average inference time per sample, substantially lower than the $0.0442_{\pm 0.0153}$ s required by CFM. The 1-d example in Fig. 6 illustrates the inference behavior of DFM and CFM under distributional heterogeneity. Near the singularity, CFM exhibits pronounced trajectory bending, which forces the model to take more discretization steps during inference. In contrast, DFM produces substantially straighter trajectories, achieving accurate generation with essentially a single step. Consequently, DFM attains higher inference efficiency along with improved accuracy in this setting.

**Curvature of the transport paths**. To assess the straightness of transport trajectories produced by DFM and CFM, we generate 1,000 transport paths on the 2-d Gaussian mixture dataset using the trained DFM and CFM, and compute the absolute curvature at the midpoint of each trajectory. The results, reported in Table 14, show that DFM yields trajectories with consistently small curvature ($0.0102_{\pm 0.0133}$), indicating substantially straighter paths, whereas CFM produces trajectories with much larger curvature ($4.3303_{\pm 3.9377}$), reflecting pronounced bending. This observation further supports the higher inference efficiency of DFM compared with CFM.

**Generation quality across different steps**. In Table 14, we report generation performance of CFM and DFM(C) on the 2-d Gaussian mixture dataset under different numbers of Euler discretization steps. We evaluate generation quality using the $W_2$ distance between the generated and target distributions. Moreover, because the keypoint-guided coupling strategy induces an optimal transport map ($X \to -X$), we additionally compute the mean squared error (MSE) between the generated target points and their ground-truth counterparts. The results show that DFM(C) consistently outperforms CFM and achieves high-quality generation with only a small number of steps. On the CIFAR-10 image generation task, we further verify that DFM achieves accurate generation with small NFE (see Table 13), and that its generation quality at low NFE is significantly higher than that of CFM (see Table 4).

***Additional remarks***. *It is important to emphasize that our DFM framework is not designed solely for image generation, where faster inference and straighter transport paths are clear advantages. DFM is also motivated by a broader range of scientific problems—such as inferring single-cell differentiation trajectories from snapshot data or recovering delay-driven dynamical systems—where overly straight trajectories are not only unhelpful but can even be detrimental. Consider single-cell trajectory inference as an example. A fundamental requirement in this setting is that the inferred trajectories*

*are supposed to remain on the data manifold, ensuring that the recovered differentiation paths reflect biologically plausible intermediate states. Excessively straight transport trajectories tend to leave the manifold, producing intermediate cell states that lack biological meaning and leading to large inference errors. As illustrated in Fig. 4, although MIOFlow (Fig. 4 (b)) infers straighter trajectories than OT-DFM(D) (Fig. 4 (f)), its paths deviate substantially from the gene-expression manifold, resulting in significantly lower trajectory inference accuracy (see Table 2).*

Table 14: Comparison of CFM and DFM(C) on the 2-d Gaussian mixture dataset in terms of training time, inference time, transport-trajectory curvature, and generation quality across different discretization steps.

| Method | CFM | DFM |
|---|---|---|
| Training time | $1.986_{\pm 0.118}$ s/iter | $2.014_{\pm 0.109}$ s/iter |
| Inference time | $0.0442_{\pm 0.0153}$ s/sample | $0.0091_{\pm 0.0062}$ s/sample |
| Curvature | $4.3303_{\pm 3.9377}$ | $0.0102_{\pm 0.0133}$ |
| 1-Step $W_2$ | 0.1561 | 0.1569 |
| 1-Step MSE | 0.0820 | 0.0211 |
| 2-Step $W_2$ | 0.5107 | 0.1325 |
| 2-Step MSE | 0.4030 | 0.0127 |
| 4-Step $W_2$ | 0.4531 | 0.0830 |
| 4-Step MSE | 0.4123 | 0.0057 |
| Adaptive-Step $W_2$ | 0.2884 | 0.0494 |
| Adaptive-Step MSE | 1.0729 | 0.0024 |

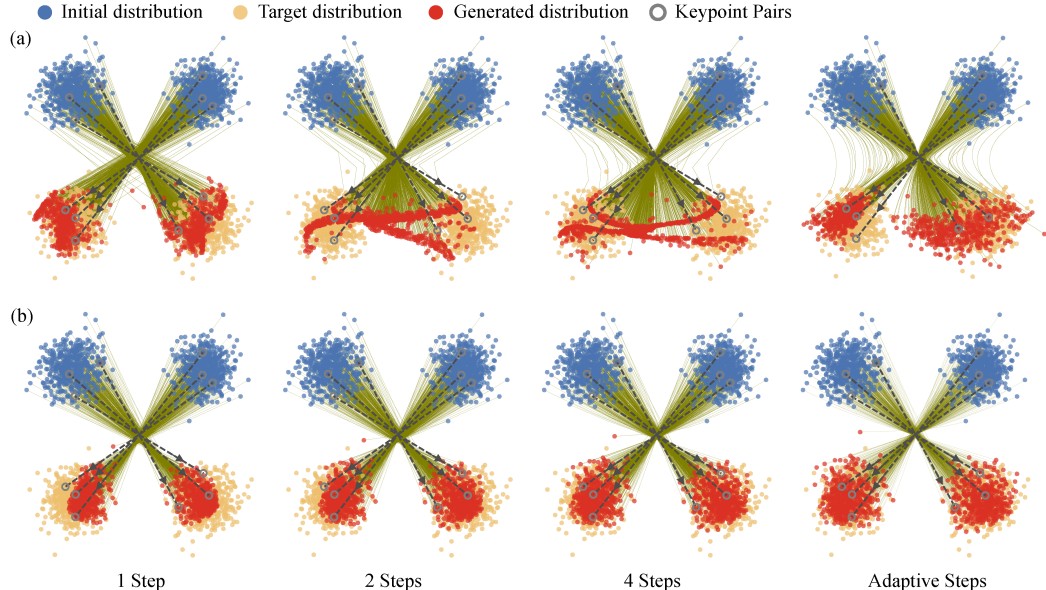

Figure 5: Comparison of CFM (a) and DFM (b) on the Gaussian mixture dataset under different numbers of sampling steps. Since DFM allows trajectory crossings, its generated paths are straighter, enabling accurate generation with very few sampling steps and substantially lower inference time. In contrast, CFM exhibits pronounced bending near trajectory-crossing regions, which requires more sampling steps, leads to slower inference, and results in lower-quality generation.

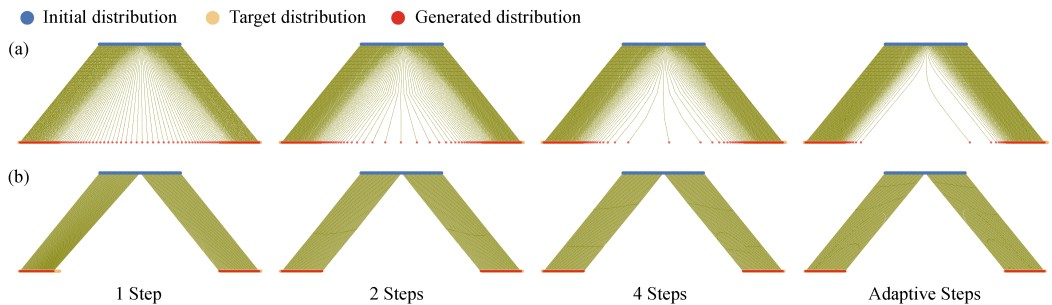

Figure 6: Comparison of CFM (a) and DFM (b) on the 1-d synthetic dataset with distributional heterogeneity under different numbers of sampling steps. CFM exhibits a singularity near 0, resulting in inaccurate generation and pronounced trajectory bending in the local neighborhood around the singularity. This necessitates more sampling steps and slows down inference. In contrast, DFM, by employing diverse initial functions, produces straighter trajectories for the two subsets from the source to the target distribution, achieving accurate generation with very few sampling steps.

# F  ABLATION STUDY

In this section, we conduct a systematic ablation study. Recall that, relative to the FM, our DFM framework additionally incorporates the delay term and the initial functions. We conduct sensitivity analyses and discuss selection strategies for both components to demonstrate the robustness of our method to their choice. Moreover, our framework admits different path interpolation schemes, coupling strategies, and the clustering procedure used to determine the initial functions, for which we also provide a detailed examination.

## F.1  SENSITIVITY ANALYSIS AND SELECTION STRATEGY FOR THE TIME DELAY $\tau$

To evaluate the effect of time delay $\tau$ selection on DFM performance, we perform a sensitivity analysis using both synthetic and real-world datasets.

First, we examine how different values of $\tau$ impact the reconstruction of delay dynamical systems. For the biological autoregulation motif, where the true time delay is $\tau = 1$, we conduct extensive experiments with $\tau$ values ranging from $0.6$ to $1.4$. As shown in Table 6, 7 and 8, while the prediction errors for both the distribution and trajectories increase as the delay deviates from the true value, KP-DFM(C) consistently captures the damped oscillation pattern across a wide range of delay values. In contrast, CFM fails to capture this behavior accurately. For the spiral DDE, with a true delay of $\tau = 0.5$, we experiment with delays from $0.35$ to $0.55$. Although the prediction errors become larger when the delay does not match the true value, OT-DFM(C) consistently outperforms CFM, achieving significantly better results, as shown in Table 10. Additionally, as illustrated in Table 9, OT-DFM(C) successfully captures the spiral dynamics across different delay values, whereas CFM exhibits divergence at the intersections of the trajectories.

Furthermore, we explore the effect of different time delay values on image generation performance. Using the MNIST dataset, we compare the generated images for $\tau = 0.125, 0.25, 0.5$, and $1.0$. As shown in Table 11, the generated images from DFM significantly outperform those generated by CFM, with FID scores also notably lower than CFM's, as summarized in Table 3. Upon examining the impact of varying time delays, we find that performance is least favorable at $\tau = 0.125$, which can be attributed to DFM's behavior approaching that of CFM as $\tau$ tends towards 0. However, for values of $\tau > 0.25$, the generation quality improves substantially.

For the choice of the delay term, we consider two scenarios: (1) the underlying system has no inherent delay; (2) the underlying system is itself a delay dynamical system.

If the system has no explicit delay structure—for example, in image generation tasks—the mere introduction of a delay term is sufficient to mitigate issues such as trajectory intersections, thereby improving generation quality and yielding results that significantly outperform ODE-based FM models. The sensitivity analysis of the delay term on the MNIST dataset further supports this claim

(see Table 3). In such cases, the key factor is simply the *presence* of a delay term, whereas the exact numerical value of $\tau$ plays a relatively minor role. If an optimal value of $\tau$ is desired, standard hyperparameter selection techniques such as grid search can be employed.

If the underlying data correspond to a genuinely delay system, then the choice of $\tau$ becomes critical for accurately capturing the system's dynamics, as illustrated in the biological autoregulation example (see Table 6, 7 and 8). Here, the closer $\tau$ is to its true value, the more accurate the trajectory reconstruction. For certain biophysical systems, prior domain knowledge often provides approximate estimates of $\tau$ (an der Heiden, 1979; Glass et al., 2021; Rihan et al., 2021). When such prior information is unavailable, we can resort to delay–identification methods from dynamical systems theory to infer the delay structure directly from data (Ma et al., 2010; 2017).

### F.2 SENSITIVITY ANALYSIS AND SELECTION STRATEGY FOR THE INITIAL FUNCTIONS

For generative frameworks based on DDEs, the initial function is an essential component. In the main text, we construct different forms of initial functions for different tasks, such as constant initial functions and diverse initial functions. For the diverse initial function, we further specify different constant time derivatives depending on the scenario (see Section 4.3.2, Eq. (22)). In this section, we demonstrate that DFM exhibits strong robustness with respect to both the choice of initial-function form and the selection of constant derivatives.

We first examine how to select the form of the initial functions. Note that diverse initial-function designs are introduced to better address transport across heterogeneous distributions. Distinct initial functions guide samples toward different target subsets. Therefore, if the source and target distributions are known to decompose into multiple modules (e.g., when single cells differentiate into several cell types), it is preferable to employ the DFM with diverse initial functions to model the generative process. If no distributional heterogeneity is present, a simple constant initial function suffices. In fact, we have theoretically established that the DFM with constant initial functions already possesses universal approximation capability for continuous transport maps between arbitrary distributions. Moreover, experiments on synthetic delay–dynamical data and the MNIST dataset further confirm that a constant initial function enables DFM to model transport maps with trajectory intersections and to capture the underlying delay-driven dynamical behavior.

Next, we examine how the selection of constant derivatives affects the performance of DFM when diverse initial functions are used. We conduct a series of comparative experiments on the real iPSCs single-cell trajectory inference task. In the main text, the initial functions employed correspond to Eq. (60), which are specifically designed based on the data structure. Here, we instead adopt a randomized procedure to generate distinct initial functions for different cell types, as follows:

$$\frac{\mathrm{d}\boldsymbol{\psi}^*(t; \boldsymbol{x}_0)}{\mathrm{d}t} = \begin{cases} \text{RandomVec1}, & \text{if } \boldsymbol{x}_2 \text{ is in M component}, \\ \text{RandomVec2}, & \text{if } \boldsymbol{x}_2 \text{ is in En component}. \end{cases} \tag{62}$$

where RandomVec1=torch.rand(4) and RandomVec2=torch.rand(4) are 4-dimensional vectors drawn independently from the uniform distribution. We randomly sampled 10 sets of initial functions, trained a separate DFM for each, and conducted the single-cell trajectory inference task. The mean and standard deviation of the inference errors are reported in Table 15 (corresponding to DFM(DR)). The results show that, compared with the specially designed initial functions (Eq. (60)), using randomized initial functions yields only a slight increase in the inference error, while still outperforming the DFM with constant initial functions and substantially surpassing the performance of CFM. This demonstrates the strong robustness of the DFM framework to the choice of initial functions. Moreover, the comparison between DFM(D) and DFM(C) further indicates that, in the presence of distributional heterogeneity, adopting diverse initial functions leads to improved generative performance, thereby reinforcing the selection strategy discussed above.

### F.3 ABLATION STUDY ON THE QUALITY OF CLUSTERING

For DFM with diverse initial functions, the design of these functions depends on the clustering results of the source and target samples. We examine two scenarios in which the clustering results are suboptimal and analyze how this affects model performance: (1) the number of clusters is smaller than the true number of underlying classes, and (2) the number of clusters exceeds the true number of classes.

Table 15: Trajectory inference results on the qPCR iPSC dataset with different initial functions. DFM(DR) denotes the DFM model equipped with diverse initial functions, where the corresponding constant time derivatives of the initial functions are randomly sampled.

| Method | $W_2$(L) | MMD(L) | $W_2$(F) | MMD(F) |
|---|---|---|---|---|
| OT-CFM | 0.579 | 0.492 | 0.226 | 0.030 |
| OT-DFM(C) | 0.553 | 0.447 | 0.234 | 0.041 |
| OT-DFM(D) | 0.532 | 0.399 | 0.213 | 0.027 |
| OT-DFM(DR) | $0.545\pm_{0.007}$ | $0.441\pm_{0.036}$ | $0.218\pm_{0.008}$ | $0.028\pm_{0.007}$ |

When the number of clusters is smaller than the true number of classes, multiple distinct subsets share the same initial function. In this case, the distributional heterogeneity among these subsets cannot be fully resolved. Nonetheless, the presence of the delay term mitigates the resulting singularities, and the overall generation quality remains substantially better than that of CFM. We illustrate this with a 1-d example in which the task is to transport $\mathcal{U}(-1, 1)$ to the two-component mixture $\frac{1}{2}\mathcal{U}(-3, -2) + \frac{1}{2}\mathcal{U}(2, 3)$. When the two modules are correctly identified and assigned distinct initial functions, the generated distribution (Fig. 7 (c)) closely matches the target. However, if the two modules are incorrectly grouped into a single cluster, the resulting generation (Fig. 7 (b)) exhibits the singularity problem near 0 in the source domain, causing a degradation in the generation quality. Nevertheless, the delay term still alleviates this issue, yielding results that remain superior to those of CFM.

When the number of clusters exceeds the true number of classes, a single true class may be split into two subclusters. In this situation, the over-segmentation provides sufficient refinement to resolve the heterogeneity-induced singularities. As shown in Fig. 7 (d), further subdividing one module of the target distribution does not negatively impact the final generation quality.

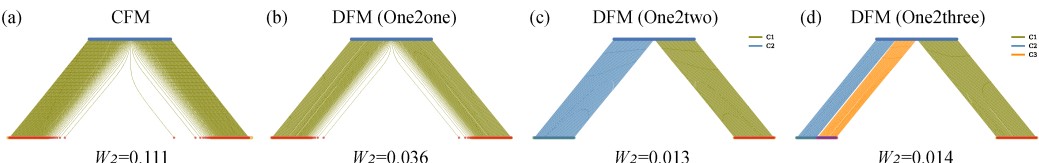

Figure 7: The generated results of CFM and DFM under different numbers of clusters, with the squared 2-Wasserstein distance between the generated and target distributions.

## F.4 SELECTION OF OTHER COMPONENTS IN THE DFM FRAMEWORK

In this section, we outline the rationale behind our choices of path–interpolation methods and coupling strategies. Across different experiments, we adopt different interpolation schemes and coupling strategies, highlighting the flexibility of the DFM framework and its ability to accommodate a variety of modeling choices. Both the selection of interpolation methods and the choice of coupling strategies are determined by the characteristics of each task and align with prior work in related domains.

Specifically, for path interpolation, we use linear interpolation when no prior geometric constraints are present, as it yields the most direct transport path and improves generative efficiency (Liu et al., 2023; Esser et al., 2024). For tasks such as single-cell trajectory inference, where paths are required to remain on the underlying data manifold, we adopt geodesic interpolation, consistent with prior work (Huguet et al., 2022; Kapuśniak et al., 2024). For continuous transport involving three or more distributions, paths connecting multiple intermediate points must be constructed, for which we employ cubic spline interpolation (Pooladian et al., 2024).

For the choice of coupling strategy, independent coupling or optimal-transport coupling is typically used when no prior constraints are available (e.g., in image generation tasks). In contrast, for certain biological systems such as single-cell gene expression, many work has shown that their evolution tends to follow energy-efficient transport patterns (Bunne et al., 2024). In such cases, employing

the optimal-transport coupling strategy is the most appropriate choice and has been widely adopted in single-cell trajectory inference (Kapuśniak et al., 2024; Sha et al., 2024). Moreover, in many scenarios, partial keypoint correspondences are known a priori, and the coupling strategy must respect these constraints. A line of research has proposed keypoint-guided optimal transport (Gu et al., 2022; 2023), which preserves keypoint correspondences while still minimizing the overall transport cost. Therefore, when partial prior coupling information is available, keypoint-guided optimal transport constitutes the most suitable strategy.

## G  COMPARISON OF DFM WITH OTHER ODE-BASED GENERATIVE METHODS

In Section 5, we discuss a series of works that modify ODE-based generative frameworks to overcome FM's limitations in handling trajectory intersections and distributional heterogeneity. Here, we focus on two representative methods: Augmented Bridge Matching (AugBM) (De Bortoli et al., 2023) and Switched Flow Matching (SFM) (Zhu & Lin, 2024). AugBM allows trajectory intersections during transport by augmenting the velocity field with information from the initial point. SFM introduces an additional switching signal as a conditioning variable and employs switching ODEs to enable transport between different modules of the source and target distribution, thereby alleviating singularities caused by distributional heterogeneity.

As discussed in Section 5, these ODE-based modifications typically address only one type of limitation and cannot simultaneously resolve both. We illustrate this with two simple examples. In Fig. 8 (a-c), we compare the generation results of CFM, DFM(C), and SFM trained under the keypoint-guided optimal transport strategy on the 2-d Gaussian mixture dataset. While SFM can alleviate issues caused by distributional heterogeneity, its representation capacity is still weaker than DFM and cannot perfectly resolve trajectory intersections. Notably, although SFM allows trajectories to cross between different mixture components, trajectories within the same component still cannot intersect. This is because points sampled from the same component share the same switching signal during generation, resulting in a single ODE that forbids intra-component trajectory crossing. Consequently, SFM cannot precisely realize the optimal transport mapping for this task ($X \to -X$). In Fig. 8 (d-f), we compare CFM, DFM(D), and AugBM on the 1-d example with distributional heterogeneity. Although AugBM allows trajectory intersections during transport, it still suffers from singularities when transporting between heterogeneous distributions, preventing it from achieving precise distribution transport.

We further highlight the distinctions between these ODE-based extensions and our proposed DFM framework. The ODE-based methods remain confined to the ODE generative paradigm, typically addressing their limitations either by augmenting the vector field with additional conditioning variables or by designing highly specialized transport paths and procedures. In contrast, DFM departs from the ODE framework and leverages the vector field with a delay term to realize distributional transport. Notably, the inclusion of the delay term naturally overcomes the limitations inherent in ODE-based approaches within the original phase space, without requiring auxiliary conditioning variables or carefully engineered transport trajectories. Moreover, for transport tasks whose underlying dynamics involve intrinsic delays, such as the gene autoregulation system in Section 6.2, ODE-based generative frameworks are incapable of recovering the true dynamics, whereas DFM, based on DDEs, can accurately model such delayed dynamics.

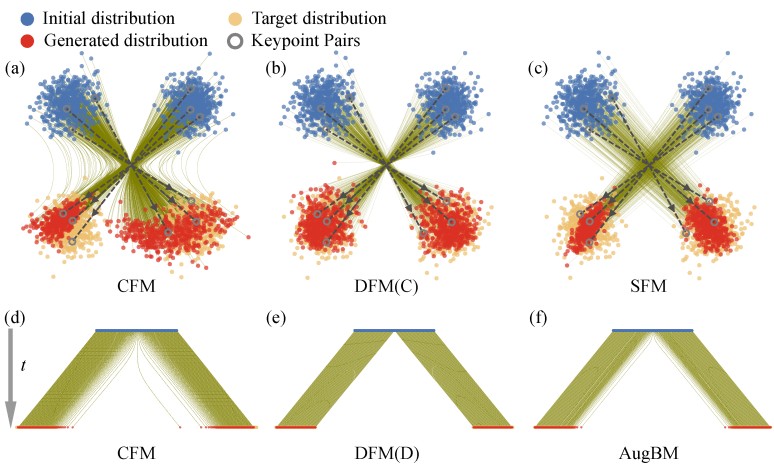

Figure 8: (a–c) Comparison of generation results from CFM (a), DFM(C) (b), and SFM (c) on the 2-d Gaussian mixture dataset. (d–f) Comparison of generation results from CFM(d), DFM(D) (e), and AugBM (f) on the 1-d dataset with distributional heterogeneity.

