# OpenReview forum: "Delay Flow Matching"
_ICLR.cc/2026/Conference — ICLR 2026 Poster_

### Official Review · Reviewer_LNpe · 2025-10-31

**Soundness:** 3
**Presentation:** 3
**Contribution:** 3
**Rating:** 6
**Confidence:** 3

**Summary:**

The paper proposes Delay Flow Matching (DFM) which is a framework that replaces the role of ODEs within standard flow matching, with flows derived from delay differential equations.
This is motivated from the fact that ODEs can fail to represent certain distribution transport plans as they cannot represent flows with intersections (relevant to real world problems such as reconstructing trajectories of single-cell data)
From this, the paper shows how the standard flow machine recipe can be generalised to work with flows from delay differential equations that overcome the previous issues. The paper then provide empirical validation of their framework on different datasets of image data and single-cell data, demonstrating compelling results.

**Strengths:**

* The core idea of replacing flow from ODEs with flows from delay differential equations is very interesting due to the properties such flows are afforded. In particular, this address key limitations of standard flow matching around trajectories not being able to intersect.

* The derivation and setup of DFM is nicely presented and formulated, as well as the theoretical justification that standard flow matching cannot handle common transport plans

* The paper provides good empirical evidence to support DFM across a range of modalities.

**Weaknesses:**

* The paper makes very limited evaluation of the profile of computational cost of training and inference (e.g. FLOPs and wall-clock time instead of just NFEs) for DFM in comparison with flow matching which makes it hard to see whether the increase in performance is worth it in cases such as image generation

* The experimental setup in the image generation differ from the standard setup - i.e. the choice of the prior distribution is made to highlight regimes where DFMs are expected to perform better, which makes it hard to mentally compare against other flow matching approaches.

**Questions:**

* Can you provide further details about the computational profile of DFM compared with flow matching

* How restrictive is the design of the initial function. For example, this requires prior information about the structural of the problem. For cases such as text to image generation, how should we design the initial function - i.e thinking about where we want to generate based on a prompt instead of the fixed classes in the dataset.

* Moreover, do you have experiments where you compare DFM with flow matching where the setup is closer to the standard setup - i.e. the prior is a isotropic Gaussian.

---

> ### Author Response · Authors · 2025-11-21
>
> We sincerely appreciate your valuable comments and constructive suggestions. To address your concerns, we have added analyses of DFM’s training and inference time, including comparisons with CFM. We have also included a sensitivity analysis of the initial function and provided guidelines for its selection. Additionally, we present comparisons between DFM and CFM on a standard image generation task. We kindly remind the reviewer that all modifications and additions in the revised manuscript are summarized in the “General Response” for convenient reference.
>
> **Response to the weaknesses and questions:**
>
> ```
> W1: The paper makes very limited evaluation of the profile of computational cost of training and inference (e.g. FLOPs and wall-clock time instead of just NFEs) for DFM in comparison with flow matching which makes it hard to see whether the increase in performance is worth it in cases such as image generation.
> Q1: Can you provide further details about the computational profile of DFM compared with flow matching?
> ```
>
> **Response**:
>
> Many thanks for your valuable suggestion. In Appendix E, we provide a detailed comparison of training and inference time between DFM and CFM. Our experiments show that although DFM incurs slightly higher training cost than CFM, it typically produces significantly *straighter* transport paths—particularly in scenarios involving trajectory intersections or distribution heterogeneity. This leads to substantially shorter inference times while maintaining high generative quality.
>
> Specifically, in Appendix E, We include comparisons of DFM and CFM across several examples, evaluating training and inference time, trajectory curvature, and generation quality under varying discretization steps (or NFE) (See Table 14, Figure 5, Figure 6, Table 13, and Table 4). The results consistently show that the paths generated by our DFM have lower curvature than CFM, enabling higher-quality sample generation in fewer steps, thereby allowing DFM to achieve both excellent efficiency and improved generation quality. *Please refer to Page 27–28, Lines 1435–1498 of the revised manuscript.*
>
> Additional Remark: It is important to note that our DFM framework is not designed solely for image generation. Our goal is to develop a broadly applicable framework for a variety of scientific problems, including single-cell trajectory inference and delay dynamical system recovery. For many scientific problems, such as the single-cell trajectory inference (see Section 6.3 of the main text), an essential requirement is that transport trajectories remain on the underlying data manifold to ensure biological interpretability. In such cases, overly straight trajectories or using too few discretization steps may cause deviations from the data manifold, leading to substantial degradation in inference accuracy. *We provide a detailed discussion and illustrations of this issue on Page 27–28, Lines 1487–1498.* Under this setting, inference efficiency is no longer the primary concern.

---

> ### Author Response · Authors · 2025-11-21
>
> ```
> W2: The experimental setup in the image generation differ from the standard setup - i.e. the choice of the prior distribution is made to highlight regimes where DFMs are expected to perform better, which makes it hard to mentally compare against other flow matching approaches.
> Q3: Moreover, do you have experiments where you compare DFM with flow matching where the setup is closer to the standard setup - i.e. the prior is a isotropic Gaussian.
> ```
>
> **Response**:
>
> We sincerely thank the reviewer for the comment. As stated in the main text, unlike standard image generation tasks, we use a more complex source distribution—a two-component Gaussian mixture. This setup is also employed in prior work [1] to assess performance under distributional heterogeneity. It allows us to more effectively assess the generative capability of the generative framework when both the source and target distributions are multimodal, thereby highlighting the model’s ability to handle more complex transport scenarios. Our experiments show that DFM performs substantially better than CFM under this setting (see Table 4).
>
> For standard image generation tasks, we further evaluate DFM(D) and CFM on the MNIST dataset, generating samples from an isotropic standard Gaussian distribution to the target data distribution. As shown in the table below, compared with CFM (which corresponds to the special case of DFM with $\tau=0$), introducing the delay term yields a slight but not substantial improvement in FID. This is likely because distributional heterogeneity is relatively mild in this standard setting. Nonetheless, DFM consistently matches or exceeds the performance of CFM across a range of settings, and it substantially outperforms CFM in more complex image generation scenarios as well as in standard single-cell trajectory inference tasks. Combined with its theoretical universal approximation property, these results highlight the advantages of the DDE-based generative framework.
>
> Table : Comparison of FID between CFM and DFM with different time delay on the standard MNIST generation task.
>
> |$\tau$|CFM($\tau=0$)|$\tau=0.25$|$\tau=0.5$|$\tau=0.75$|$\tau=1$|
> |:---:|:---:|:---:|:---:|:---:|:---:|
> |FID|$5.6678$|$5.4643$|$5.5824$|$5.6026$|$5.6471$|
>
> ```
> Q2: How restrictive is the design of the initial function. For example, this requires prior information about the structural of the problem. For cases such as text to image generation, how should we design the initial function - i.e thinking about where we want to generate based on a prompt instead of the fixed classes in the dataset.
> ```
>
> **Response**:
>
> Many thanks for your valuable question. In fact, DFM demonstrates strong robustness with respect to the choice of the initial functions and imposes minimal restrictions or constraints on its selection. In Appendix F2, we provide the sensitivity analysis of the initial functions and corresponding selection strategies. We first discuss under which circumstances DFM should adopt diverse initial functions and when a constant initial function is more appropriate. By validating our approach on real single-cell datasets, we then show that for DFM(D), even when the initial functions are randomly selected, the generative performance remains stable and consistently surpasses that of CFM (see Table 15). This demonstrates the robustness of DFM with respect to the choice of initial functions. *The detailed experimental results and analysis can be found on Page 31, Lines 1620-1667 of the revised manuscript.*
>
> The question regarding generating based on the prompt is indeed very interesting, but it goes beyond the scope of the present work. In fact, when using DFM for image generation, we incorporate an embedding layer that allows the model to adaptively adjust the initial function during training. This mechanism could potentially be extended to prompt-based generation by learning a mapping from the prompt to the initial function via a neural network, thereby guiding the generative process of DFM. Of course, the effectiveness of this approach remains to be validated in future work. Overall, we think that this is a very intriguing direction, and we plan to explore it further in our future research.
>
> [1] Zhu Q, Lin W. Switched flow matching: eliminating singularities via switching ODEs[C]//Proceedings of the 41st International Conference on Machine Learning. 2024: 62443-62475.

---

> > ### Comment · Reviewer_LNpe · 2025-11-26
> >
> > Thank you for addressing my questions. I will maintain my score supporting the acceptance of the paper.

---

> > > ### Author Response · Authors · 2025-11-27
> > > **Thank you for supporting acceptance**
> > >
> > > Thank you for your feedback and for confirming that our revisions have addressed your questions.
> > >
> > > We are grateful for the opportunity to improve our work. Having substantively addressed all the points you raised, we believe the manuscript now aligns more closely with the "accept" (Rating 8) category. We would be delighted if you found the improvements significant enough to warrant a score that more clearly reflects this.
> > >
> > > Thank you again for your time and guidance throughout this process.

---

### Official Review · Reviewer_7LHR · 2025-11-01

**Soundness:** 3
**Presentation:** 3
**Contribution:** 2
**Rating:** 4
**Confidence:** 3

**Summary:**

The paper proposes a new generative-model framework based on the idea of using delay‐differential equations (DDEs) instead of ordinary differential equations (ODEs) for the “flow‐matching” paradigm. Traditional flow matching (FM) uses an ODE to learn a vector field transporting samples from a simple base distribution (e.g., Gaussian) to a target data distribution. The authors identify three limitations of standard FM: inability to model trajectory intersections, and difficulty transferring between heterogeneous distributions. To address this, they introduce Delay Flow Matching (DFM): rather than modelling $\dot{x}(t)=v(x(t),t)$, they incorporate delay terms $\dot{x}(t)=v(x(t),x(t-\tau),t)$ in a neural-parametrised vector field. They further provide a theoretical universal approximation guarantee.

**Strengths:**

1. It clearly articulated the limitation of classic FM models.

2. The authors introduce the delay differential equation (DDE) dynamics.

3. They provide the universal approximation claim that DFM can approximate any continuous tansfer map, providing the theoretical pinning to the method.

4. The experiments include a wide range of data, including synthetic, biological, and natural images.

**Weaknesses:**

1. There are several papers that addressed the problem of non-intersecting trajectories in rectified flow / flow matching [1, 2, 3, 4]. I think it would be good to review and compare with those related works.

2. It seems to me that the proposed framework might be computationally more expensive than the classical FM. What is the trade-off between efficiency and quality?

3. I feel the FID value is too high in Table 3 for $\tau = 0$ (CFM).

4. The FID values for OT-CFM on CIFAR-10 seem to be higher than what earlier works reported.

References:
[1] Park et al. Constant Acceleration Flow, 2024
[2] Zhang et al. Towards Hierarchical Rectified Flow, 2025
[3] Chen et al. Gaussian Mixture Flow Matching Models, 2025
[4] Guo et al. Variational Rectified Flow Matching, 2025

**Questions:**

1. Could you provide the training time and inference time for DFM and other baseline models?

2. Could you add more baselines (using the papers mentioned above) in the experiments?

3. Could you add a section to discuss related work?

---

> ### Author Response · Authors · 2025-11-21
>
> We feel great thanks for your professional review work on our article. According to your valuable suggestions, we have added a comprehensive summary of related work, included comparisons between DFM and additional baseline methods, and provided a detailed analysis comparing the training and inference time of DFM with that of the classical FM.  We kindly remind the reviewer that all modifications and additions in the revised manuscript are summarized in the “General Response” for convenient reference.
>
> **Response to the weaknesses and questions:**
>
> ```
> W1: There are several papers that addressed the problem of non-intersecting trajectories in rectified flow / flow matching. I think it would be good to review and compare with those related works.
> Q3: Could you add a section to discuss related work?
> ```
>
> **Response**:
>
> We sincerely thank the reviewer for the valuable suggestion. We acknowledge that several ODE-based FM extensions have been proposed, which partially address the limitations of classial FM. We add a Related Work section (Section 5) to summarize existing ODE-based generative methods that address the problem of trajectory intersections and distributional heterogeneity. We further provide the comparison between these variants and our DDE–based generative framework, highlighting the remaining limitations of ODE-based methods and the conceptual advantages of DFM. *Please refer to Page 7–8, Lines 374–397 of the revised manuscript.*
>
> ```
> Q2: Could you add more baselines (using the papers mentioned above) in the experiments?
> ```
>
> **Response**:
>
> Many thanks for your comment. In the Related Work section, we not only summarize existing ODE-based generative methods that address trajectory intersections and distributional heterogeneity, but also analyze their remaining limitations. These include the inability to simultaneously handle both trajectory intersections and distributional heterogeneity, confinement within the ODE framework, reliance on additional latent variables to augment the phase space, the need for carefully designed transport procedures, and the inability to model distributional changes guided by delay dynamics. We further highlight the advantages of our DFM framework. In contrast to ODE-based approaches, DFM departs from the ODE paradigm and introduces a fundamentally new DDE-based generative framework. By modeling the vector field directly in the original phase space—without auxiliary latent variables or specially engineered transport paths—DFM naturally accommodates both trajectory intersections and distributional heterogeneity within a unified mechanism. Moreover, DFM inherently captures transport dynamics governed by intrinsic delays, a capability that is completely beyond the reach of existing ODE-based methods. *Please refer to Page 7–8, Lines 386–397 of the revised manuscript.*
>
> In Appendix G, we include comparisons with additional baseline methods to further validate the limitations of these ODE-based variants discussed above and to demonstrate the advantages of DFM. Among the various ODE-based FM variants, we select two representative methods—Augmented Bridge Matching (AugBM)[1] and Switched Flow Matching (SFM)[2]—for direct comparison. AugBM augments the vector field with initial-point information to allow trajectory crossing, whereas SFM introduces a switching signal to enable precise transport between heterogeneous distributions. Through experiments on illustrative examples, we show that these baseline methods still suffer from inherent limitations, while our DFM framework is able to simultaneously overcome these issues. *Details can be found on Page 32–33, Lines 1724–1756 of the revised manuscript.*

---

> ### Author Response · Authors · 2025-11-21
>
> ```
> W2: It seems to me that the proposed framework might be computationally more expensive than the classical FM. What is the trade-off between efficiency and quality?
> Q1: Could you provide the training time and inference time for DFM and other baseline models?
> ```
>
> **Response**:
>
> We sincerely thank the reviewer for the valuable question. In Appendix E, we provide a detailed comparison of training and inference time between DFM and CFM. Our experiments show that although DFM incurs slightly higher training cost than CFM, it typically produces significantly *straighter* transport paths—particularly in scenarios involving trajectory intersections or distribution heterogeneity. This leads to substantially shorter inference times while maintaining high generative quality.
>
> Specifically, in Appendix E, We include comparisons of DFM and CFM across several examples, evaluating training and inference time, trajectory curvature, and generation quality under varying discretization steps (or NFE) (See Table 14, Figure 5, Figure 6, Table 13, and Table 4). The results consistently show that the paths generated by our DFM have lower curvature than CFM, enabling higher-quality sample generation in fewer steps, thereby allowing DFM to achieve both excellent efficiency and improved generation quality. *Please refer to Page 27–28, Lines 1435–1498 of the revised manuscript.*
>
> Additional Remark: It is important to note that our DFM framework is not designed solely for image generation. Our goal is to develop a broadly applicable framework for a variety of scientific problems, including single-cell trajectory inference and delay dynamical system recovery. For many scientific problems, such as single-cell trajectory inference (see Section 6.3 of the main text), an essential requirement is that transport trajectories remain on the underlying data manifold to ensure biological interpretability. In such cases, overly straight trajectories or using too few discretization steps may cause deviations from the data manifold, leading to substantial degradation in inference accuracy. *We provide a detailed discussion and illustrations of this issue on Page 27–28, Lines 1487–1498.* Under this setting, inference efficiency is no longer the primary concern.
>
>
> ```
> W3: I feel the FID value is too high in Table 3 for CFM.
> W4: The FID values for OT-CFM on CIFAR-10 seem to be higher than what earlier works reported.
> ```
>
> **Response**:
>
> Many thanks for your question. In fact, due to our specific task design, the semi-paired image-to-image task on MNIST dataset is inherently challenging, as all constructed interpolation trajectories intersect at a single point. This makes the generation task particularly difficult and results in relatively high FID scores for all methods. In particular, CFM, due to its inherent limitation of prohibiting trajectory intersections, performs poorly under such settings, resulting in very high FID scores. However, under this setting, our comparative experiments show that DFM achieves significantly lower FID scores than CFM, which clearly demonstrates the advantage of DFM in this scenario.
>
> For the generation task on the CIFAR-10 dataset, as stated in the main text, different from standard image generation tasks, we use a more complex source distribution—a two-component Gaussian mixture. This setup allows us to more effectively assess the generative capability of the generative framework when both the source and target distributions are multimodal, thereby highlighting the model’s ability to handle more complex transport scenarios. On one hand, it is reasonable that the FID is slightly higher than in standard generation tasks due to the increased complexity of the problem. On the other hand, this example has been studied in prior work [2], and the FID of CFM reported there closely matches our experimental results, further confirming the reliability of our results in Table 4.
>
>
> [1] Valentin De Bortoli, Guan-Horng Liu, Tianrong Chen, Evangelos A Theodorou, and Weilie Nie. Augmented bridge matching. arXiv preprint arXiv:2311.06978, 2023.
>
> [2] Zhu Q, Lin W. Switched flow matching: eliminating singularities via switching ODEs[C]//Proceedings of the 41st International Conference on Machine Learning. 2024: 62443-62475.

---

> ### Author Response · Authors · 2025-11-26
>
> We would like to thank the reviewer once again for pointing out the limitations in our previous manuscript and for raising valuable questions. In response, we have conducted a systematic revision of the manuscript.
>
> The revised manuscript includes a comprehensive summary of related work, with representative methods compared against DFM, further highlighting DFM’s advantages over ODE-based FM variants. We have also provided a detailed analysis, supported by concrete experiments, of DFM’s training and inference efficiency, demonstrating that it not only achieves higher generation quality but also offers superior inference efficiency compared to ODE-based FM. Regarding the relatively high FID values, we have provided explanations and compared our results with those reported in prior work on the same image generate task, confirming the reasonableness of the FID values we obtained.
>
> We have summarized all revisions in the General Response (https://openreview.net/forum?id=6lH1XblLpo&noteId=cRynrRY8x5), with the page numbers and locations of each change clearly indicated in the revised manuscript. We sincerely appreciate your careful review of the updated manuscript and hope that these additional clarifications and results will be taken into consideration in your reassessment of our work.
>
> If you have any further questions or need additional clarifications, please feel free to contact us. Your feedback is highly valuable, and we are eager to address any remaining issues to improve our work.
>
> Thank you for your time and effort in reviewing our revised manuscript.

---

### Official Review · Reviewer_qSLq · 2025-11-02

**Soundness:** 3
**Presentation:** 2
**Contribution:** 2
**Rating:** 4
**Confidence:** 2

**Summary:**

This paper proposes Delay Flow Matching (DFM), a framework that incorporates a delay term and appropriate initial functions to enable the model to capture delay dynamics, handle trajectory intersections, and transfer between heterogeneous distributions. The paper theoretically proves the universal approximation capability of DFM and demonstrates its effectiveness on a series of synthetic and real-world tasks through various experiments.

**Strengths:**

This work seems to be the first to combine DDE with FM. The delay term enables the model to capture delay dynamics and handle trajectory intersections, while the design of appropriate initial functions allows it to handle heterogeneous distributions. The paper provides a compelling theoretical analysis, demonstrating the limitations of FM and proving the expressive power of DFM. The effectiveness of DFM is validated through various experiments.

**Weaknesses:**

1. Lack of a Systematic Ablation Study: The DFM framework introduces multiple components, including the presence of a delay term, the choice of initial function (constant vs. diverse), the path interpolation method (linear vs. CSpline vs. geodesic), and the coupling strategy (OT-, KP-, I-). While the paper does conduct some comparisons across different experiments, these analyses are scattered and do not form a single, comprehensive ablation study.

2. Missing discussion and contextualization of related work: The paper fails to discuss the recent research progress in handling trajectory intersections of FM as well as FM for heterogeneous data. A prime example is Switched Flow Matching (SFM), whose core idea is highly similar to this work's: to handle heterogeneous distributions, SFM "switches" between different ODEs based on clustering, whereas this paper "switches" the initial function. However, the paper lacks both a discussion of this conceptual relationship and a necessary experimental comparison.

3. Lack of specific guidance for the selection of tau: Tau is a critical parameter of the proposed algorithm, exerting a significant impact on the algorithm’s performance. While the paper presents experiments with substantially different tau values (e.g., tau = 1 or tau = 3), it fails to offer specific guidance on how to select tau in practice (although provides a sensitivity analysis for tau)

4. Lack of details on efficiency and resource consumption: A major pain point of FM lies in its slow inference speed. While the paper designs experiments primarily focusing on the algorithm’s accuracy, it fails to provide detailed comparisons with other models regarding training and inference time, which are valuable for practical implementation considerations.

5. For image translation experiments, the generated images are in wrong contrast and rather blurry, for which FID is not sufficient to justify the performance. More image quality index should be considered, especially for paired case. Furthermore, the image translation experiments are performed in low-dimensional (such as MINST and Cifar), how the method can be scalable to  high dimension (>=256)
image generation problems?

**Questions:**

1. For the experiments, various adaption with KP, OT and Conditional FM are used, the corresponding  cost function is not precised.
2.  note that the performance of I-DFM(D) in Table 4 outperforms that of OT-DFM(D). However, only OT-DFM was utilized in the preceding sections of the paper. Could you explain the reason for this inconsistency? Additionally, how should we reasonably tune each component of the proposed algorithm in practice?

2. The diverse initial function appears to be built on the clustering of the dataset. If the clustering results are not particularly optimal (e.g., an excessive or insufficient number of clusters), to what extent would this affect the model’s performance?

3. All image generation tasks in the paper are designed for scenarios with trajectory intersections and heterogeneous data. If DFM were applied to a standard image generation benchmark task, what performance would you anticipate it to achieve? Would the "Delay" mechanism still provide advantages in such a scenario?

---

> ### Author Response · Authors · 2025-11-21
>
> We would like to thank the reviewer for the valuable feedback and helpful suggestions, which help a lot to improve the quality of our manuscript. In the revised manuscript, we have included a systematic ablation study of all key components in DFM, provided guidelines for selecting each component, summarized related work, and added a detailed analysis of the training and inference efficiency of DFM. We kindly remind the reviewer that all modifications and additions in the revised manuscript are summarized in the “General Response” for convenient reference.
>
> **Response to the weaknesses**
>
> ```
> W1: Lack of a Systematic Ablation Study.
> ```
>
> **Response**:
>
> We sincerely thank the reviewer for the valuable suggestion. In Appendix F of the revised manuscript, we have added a single, comprehensive ablation study that thoroughly analyzes and validates the robustness of our DFM framework with respect to all its components. We also provide practical guidelines for selecting each module in real applications.
>
> Specifically, Appendix F now includes a systematic set of ablation experiments that cover:
>
> (F.1) a sensitivity analysis of the time delay $\tau$ and guidelines for its selection;
>
> (F.2) a sensitivity analysis of the initial functions and corresponding selection strategies;
>
> (F.3) an ablation study on how clustering quality affects the generative performance of DFM;
>
> (F.4) additional analysis of the choices related to other DFM components, including the construction of interpolation paths, the employed coupling strategies, and the rationale for selecting specific strategies in different tasks.
>
> These ablations collectively demonstrate that the proposed DFM framework exhibits strong robustness with respect to all of its components.
>
> *Please refer to Page 28–32, Lines 1500–1723 of the revised manuscript.*
>
>
>
> ```
> W2: Missing discussion and contextualization of related work.
> ```
>
> **Response**:
>
> Many thanks for your suggestion. We add a dedicated Related Works section (Section 5) to summarize existing ODE-based generative methods that address the problem of trajectory intersections and distribution heterogeneity. We further provide a detailed comparison between these variants and our DDE–based generative framework, highlighting the remaining limitations of ODE-based methods and the conceptual advantages of DFM. *Please refer to Page 7–8, Lines 374–397 of the revised manuscript.*
>
> Besides, we also conduct comparisons with additional baseline methods. Among the various ODE-based Flow Matching variants discussed in the Related Works section, we select two representative methods—Augmented Bridge Matching (AugBM) and Switched Flow Matching (SFM)—for direct comparison. AugBM augments the vector field with initial-point information to allow trajectory crossing, whereas SFM introduces a switching signal to enable precise transport between heterogeneous distributions. Through experiments on illustrative examples, we show that these baseline methods still suffer from inherent limitations, while our DFM framework is able to simultaneously overcome these issues. *Details can be found on Page 32–33, Lines 1724–1756 of the revised manuscript.*
>
> ```
> W3: Lack of specific guidance for the selection of tau.
> ```
>
> **Response**:
>
> In Appendix F1, we provide a detailed discussion on how the delay term $\tau$ should be chosen in practical applications under two different scenarios: (1) when the underlying system has no inherent delay, and (2) when the underlying system is itself a delay dynamical system. *Please refer to Page 30, Lines 1605-1619 of the revised manuscript.*
>
> ```
> W4: Lack of details on efficiency and resource consumption.
> ```
>
> **Response**:
>
> Many thanks for your comment. In Appendix E, we provide a detailed comparison of training and inference time between DFM and CFM. Our experiments show that although DFM incurs slightly higher training cost than CFM, it typically produces significantly *straighter* transport paths—particularly in scenarios involving trajectory intersections or distribution heterogeneity. This leads to substantially shorter inference times while maintaining high generative quality.
>
> We include comparisons of DFM and CFM across several examples, evaluating training and inference time, trajectory curvature, and generation quality under varying discretization steps (or NFE) (See Table 14, Figure 5, Figure 6, Table 13, and Table 4). The results consistently show that the paths generated by our DFM have lower curvature than CFM, enabling higher-quality sample generation in fewer steps. *Please refer to Page 27–28, Lines 1435–1498 of the revised manuscript.*

---

> ### Author Response · Authors · 2025-11-21
>
> ```
> W5: For image translation experiments, the generated images are in wrong contrast and rather blurry, ···, how the method can be scalable to high dimension (>=256) image generation problems?.
> ```
>
> **Response**:
>
> We sincerely thank the reviewer for the valuable suggestion. In fact, due to our specific task design, the semi-paired image-to-image transport setting is inherently challenging, as all constructed interpolation trajectories intersect at a single point. This makes the generation task particularly difficult and results in relatively high FID scores for all methods. However, under this setting, our comparative experiments show that DFM achieves significantly lower FID scores than CFM, which clearly demonstrates the advantage of DFM in this scenario.
>
> Moreover, under the keypoint-guided OT strategy, our primary objective is to generate high-quality samples while preserving the predefined pairing structure. For this reason, prior work [1,2] on semi-paired image-to-image translation has also widely adopted FID as a standard evaluation metric. We fully agree that developing more task-specific quality metrics would provide a more comprehensive evaluation of generative performance. We plan to explore and design additional specialized evaluation metrics in future work.
>
> Furthermore, due to constraints on time and computational resources, we did not evaluate DFM on higher-dimensional image data. Nevertheless, through rigorous theoretical analysis and extensive experiments across multiple tasks and domains, we have thoroughly demonstrated the significant advantages of DFM in handling trajectory intersections and distributional heterogeneity. In future work, we plan to evaluate the generative performance of DFM on higher-resolution images.
>
> It is important to note that our DFM framework is not designed solely for image generation. Our goal is to develop a broadly applicable framework for a variety of scientific problems, including single-cell trajectory inference and delay dynamical system recovery. We have also validated the effectiveness and applicability of DFM in these tasks through comprehensive experiments.
>
>
>
> **Response to the questions**
>
> ```
> Q1: For the experiments, various adaption with KP, OT and Conditional FM are used, the corresponding cost function is not precised.
> ```
>
> **Response**:
>
> Many thanks for your comment. We first clarify that, regardless of the chosen coupling strategy or interpolation path, DFM is always trained using the loss function in Eq. (16) of the main text, while CFM is trained using the loss function in Eq. (8). The choice of different coupling strategies or interpolation paths only affects the distribution of the latent variables $q(z)=\pi(x_{0},x_1)\mathcal{P}(\gamma;x_{0},x_1)$, whiere the coupling strategy determines $\pi$, and the interpolation path determines $\mathcal{P}$, and does not alter the form of the loss function used to train the vector field.
>
> Furthermore, for the various coupling strategies mentioned in the manuscript, we have provided explicit implementation details in the main text. For example, the coupling strategy for KPG-OT is obtained by minimizing Eq. (5), while the coupling strategy for OT is obtained by minimizing the objective in Eq. (4).
>
>
>
> ```
> Q2: Note that the performance of I-DFM(D) in Table 4 outperforms that of OT-DFM(D). However, only OT-DFM was utilized in the preceding sections of the paper. Could you explain the reason for this inconsistency? Additionally, how should we reasonably tune each component of the proposed algorithm in practice?
> ```
>
> **Response**:
>
> We sincerely thank the reviewer for the valuable question. In Appendix F of the revised manuscript, we provide practical guidelines for selecting each component of DFM in real applications. We also thoroughly analyze and validate the robustness of our DFM framework with respect to all its components. *Please refer to Page 28–32, Lines 1500–1723 of the revised manuscript.*
>
> Specifically, in Appendix F4, we analyze and explain why different coupling strategies and interpolation paths are selected for different tasks, demonstrating the rationale behind these choices and their consistency with prior work. We also provide practical guidelines on how to select appropriate components in real-world applications.
> In Appendix F1, we discuss in detail how to choose the time delay $\tau$ under different practical scenarios.
> In Appendix F2, we examine when different initial functions should be used and empirically validate the robustness of DFM with respect to the choice of initial functions.

---

> ### Author Response · Authors · 2025-11-21
>
> ```
> Q3: The diverse initial function appears to be built on the clustering of the dataset. If the clustering results are not particularly optimal (e.g., an excessive or insufficient number of clusters), to what extent would this affect the model’s performance?
> ```
>
> **Response**:
>
> We sincerely thank the reviewer for the question. In Appendix F3, through detailed experiments and analysis, we investigate how clustering quality affects the generative performance of DFM. Using an example involving distributional heterogeneity, our ablation study shows that an insufficient number of clusters can lead to a moderate degradation in performance; however, the generation quality still remains superior to that of CFM. In contrast, using an excessive number of clusters does not substantially affect the generative performance of DFM. *Please refer to Page 31–32, Lines 1668–1698 of the revised manuscript.*
>
>
> ```
> Q4: All image generation tasks in the paper are designed for scenarios with trajectory intersections and heterogeneous data. If DFM were applied to a standard image generation benchmark task, what performance would you anticipate it to achieve? Would the "Delay" mechanism still provide advantages in such a scenario?
> ```
>
> **Response**:
>
> Many thanks for your comment. As stated in the main text, unlike standard image generation tasks, we use a more complex source distribution—a two-component Gaussian mixture. This setup is also employed in prior work [3] to assess performance under distributional heterogeneity. It allows us to more effectively assess the generative capability of the generative framework when both the source and target distributions are multimodal, thereby highlighting the model’s ability to handle more complex transport scenarios. Our experiments show that DFM performs substantially better than CFM under this setting (see Table 4).
>
> For standard image generation tasks, we further evaluate DFM(D) and CFM on the MNIST dataset, generating samples from an isotropic standard Gaussian distribution to the target data distribution. As shown in the table, compared with CFM (which corresponds to the special case of DFM with $\tau=0$), introducing the delay term yields a slight but not substantial improvement in FID. This is likely because distributional heterogeneity is relatively mild in this standard setting. Nonetheless, DFM consistently matches or exceeds the performance of CFM across a range of settings, and it substantially outperforms CFM in more complex image generation scenarios as well as in standard single-cell trajectory inference tasks. Combined with its theoretical universal approximation property, these results highlight the advantages of the DDE-based generative framework.
>
> Table : Comparison of FID between CFM and DFM with different time delay on the standard MNIST generation task.
>
> |$\tau$|CFM($\tau=0$)|$\tau=0.25$|$\tau=0.5$|$\tau=0.75$|$\tau=1$|
> |:---:|:---:|:---:|:---:|:---:|:---:|
> |FID|$5.6678$|$5.4643$|$5.5824$|$5.6026$|$5.6471$|
>
>
> [1] Gu X, Yang L, Sun J, et al. Optimal transport-guided conditional score-based diffusion model[J]. Advances in Neural Information Processing Systems, 2023, 36: 36540-36552.
>
> [2] Mustafa A, Mantiuk R K. Transformation consistency regularization–a semi-supervised paradigm for image-to-image translation[C]//European conference on Computer Vision. Cham: Springer International Publishing, 2020: 599-615.
>
> [3] Zhu Q, Lin W. Switched flow matching: eliminating singularities via switching ODEs[C]//Proceedings of the 41st International Conference on Machine Learning. 2024: 62443-62475.

---

> ### Author Response · Authors · 2025-11-26
>
> We sincerely thank the reviewer again for the valuable comments and suggestions, which have greatly helped us improve the overall quality of the manuscript.
>
> Based on your comments, we have made substantial revisions to the paper. In the updated manuscript, we have added a series of systematic ablation studies (including sensitivity analyses on the time delay $\tau$ and the initial functions, ablations on the impact of clustering quality, and analyses of the design rationale behind each component of DFM). We have also included a more comprehensive summary of related work, along with representative comparisons and analyses between prior methods and DFM. In addition, we have provided detailed guidance on the selection of key components in DFM (including the time delay $\tau$, the initial functions, the coupling strategy, and the path interpolation), as well as a thorough comparison of the training and inference efficiency of DFM versus ODE-based FM. We have further included experimental results on standard image generation tasks, showing that DFM achieves comparable generation quality to ODE-based FM, while significantly outperforming it on more complex generation tasks. Together, these additional experiments and analyses demonstrate the robustness of DFM and underscore its performance advantages over ODE-based FM approaches.
>
> All revisions have been summarized in the General Response (https://openreview.net/forum?id=6lH1XblLpo&noteId=cRynrRY8x5), and for each comment we have clearly indicated the corresponding page numbers and locations in the revised manuscript. We sincerely appreciate your time and effort in reviewing our revised manuscript, and we kindly hope that you will take these additional experiments, analyses, and clarifications into consideration when evaluating the updated manuscript.
>
> If you have any additional questions or concerns, please feel free to let us know. Your insights are highly valuable to us, and we are happy to address any further issues that could help improve our work.
>
> Thank you again for your time and thoughtful review.

---

### Official Review · Reviewer_vrRC · 2025-11-02

**Soundness:** 3
**Presentation:** 4
**Contribution:** 3
**Rating:** 6
**Confidence:** 3

**Summary:**

The paper extends Flow Matching to Delay Differential Equations via Delay Flow Matching (DFM), where the drift $u(t,x,x_\tau)$ depends on both the current state and a delayed past state. This lets the model capture trajectory intersections, delay dynamics, and heterogeneous source/target supports while remaining simulation-free to train. The authors instantiate variants with constant vs diverse initial functions and provide empirics on synthetic delay systems, single-cell trajectories, and image tasks, showing consistent gains over ODE-FM baselines.

**Strengths:**

The paper is original in extending flow matching to delay differential equations and in using tailored initial-function designs, including constant and diverse variants, to cope with heterogeneous supports and keypoint-guided transport. The theoretical development is solid as the paper establishes a universal-approximation result for continuous transport maps together with a well-motivated conditional training objective. The empirical evaluation is broad, spanning synthetic delayed systems, single-cell trajectory inference, and image generation, with consistent improvements on standard metrics. The writing is generally clear and well structured.

**Weaknesses:**

1) The practical significance is less clear in settings where the sole goal is high-quality final samples and intermediate trajectories are irrelevant. In such cases, the added machinery of delays and historical states may not translate into noticeably better samples versus strong ODE/SDE baselines.

2) Relatedly, some benefits attributed to the framework (e.g., handling heterogeneous classes) might also be achieved with common conditioning mechanisms such as classifier-free guidance, where the network ingests class or conditioning vectors directly. It is unclear whether the proposed approach would outperform such simpler alternatives.

3) The paper is a little imprecise at times, using phrases like "approximately push-forward" without a precise definition, which makes it hard to interpret the guarantees.

**Questions:**

1) How does the required time-step (or number of function evaluations) for stable and accurate integration of DDEs compare to ODEs in your experiments? Do DDEs typically demand finer discretization?

2) Many practical generative systems rely on guidance (e.g., classifier-free guidance). How does your framework integrate with these mechanisms? Does introducing delays change how guidance is applied?

3) Your approach allows different initial function choices. How sensitive is performance to this choice in practice?

---

> ### Author Response · Authors · 2025-11-21
>
> We sincerely thank the reviewer for the insightful comments and constructive suggestions, which have greatly contributed to improving the quality of our manuscript. Below, we provide detailed responses to each of the reviewer’s comments. We kindly remind the reviewer that all modifications and additions in the revised manuscript are summarized in the “General Response” for convenient reference.
>
> **Response to the weaknesses**
>
> ```
> W1: The practical significance is less clear in settings where the sole goal is high-quality final samples and intermediate trajectories are irrelevant. In such cases, the added machinery of delays and historical states may not translate into noticeably better samples versus strong ODE/SDE baselines.
> ```
>
> **Response**:
>
> We sincerely thank the reviewer for the comment. In many tasks, generating high-quality final samples is inherently tied to the transport process, making it difficult to evaluate generation quality in isolation. For example, in classical image generation tasks, one must learn a generative process that transports an initial noise distribution to image distributions of multiple semantic categories. Such transport between heterogeneous distributions naturally encounters singularities, which can lead to semantically ambiguous samples and degraded generation quality [1]. As demonstrated in several experiments in the main text, using diverse initial functions within the DFM framework significantly improves the quality of generated samples under such settings.
>
> Moreover, our DFM framework is not designed solely for image generation. A key motivation is to address scientific problems such as single-cell trajectory inference, where the goal is to infer the distribution of gene expression states at different time during the differentiation. In these applications, the intermediate transport trajectory is crucial. Ignoring the transport path would produce trajectories lacking biological interpretability and introduce substantial errors.
>
> ```
> W2: Relatedly, some benefits attributed to the framework (e.g., handling heterogeneous classes) might also be achieved with common conditioning mechanisms such as classifier-free guidance. It is unclear whether the proposed approach would outperform such simpler alternatives.
> ```
>
> **Response**:
>
> Many thanks for your valuable comment. We discuss this issue from two aspects:
>
> 1.	Our proposed DFM framework can naturally integrate with guidance-based methods, allowing for further extensions. We provide a detailed explanation in our response to Q2.
>
> 2.	Existing guidance-based methods still have limitations. For example, while they can handle distributional heterogeneity, they do not allow trajectories to cross arbitrarily, which prevents accurate modeling of certain transport strategies. DFM, by introducing delay terms, fundamentally overcomes this limitation. To illustrate this, we consider Switched Flow Matching (SFM), which introduces a switching signal as guidance to achieve precise transport between heterogeneous distributions. In Appendix G, by comparing the generative results of SFM and DFM on a 2-d Gaussian mixture dataset with key-point guidance (Figure 8(b,c)), we show that while SFM can mitigate issues caused by distributional heterogeneity, its representation capacity remains limited and cannot perfectly handle trajectory intersections. Specifically, although SFM allows trajectories to cross between different mixture components, trajectories within the same component cannot intersect because points sampled from the same component share the same switching signal during generation, resulting in a single ODE that forbids intra-component trajectory crossing (Figure 8(c)). Consequently, SFM cannot precisely realize the optimal transport mapping for this task ($X \to -X$).In contrast, our DFM framework, by incorporating delay terms, not only achieves precise transport between heterogeneous distributions but also accurately models transport strategies involving trajectory intersections (Figure 8(b)), demonstrating its advantage over guidance-based methods.
>
> *Details about the comparison with additional baseline methods can be found on Page 32–33, Lines 1724–1756.*
>
>
> ```
> W3: The paper is a little imprecise at times, using phrases like "approximately push-forward" without a precise definition, which makes it hard to interpret the guarantees.
> ```
>
> **Response**:
>
> Many thanks for the valuable suggestion. We acknowledge that our previous presentation of the theorem lacked rigor and precision. We have revised the statement of the universal approximation theorem and the corresponding proof using more rigorous mathematical language to eliminate any potential confusion or ambiguity.  *For the revised version, please refer to Page 7, Lines 331–336 of the revised manuscript.*
>
> [1] Zhu Q, Lin W. Switched flow matching: eliminating singularities via switching ODEs, International Conference on Machine Learning. 2024.

---

> > ### Author Response · Authors · 2025-11-21
> >
> > ```
> > Q3: Your approach allows different initial function choices. How sensitive is performance to this choice in practice?
> > ```
> >
> > **Response**:
> >
> > Many thanks for your comment. In Appendix F2, we provide the sensitivity analysis of the initial functions and corresponding selection strategies. We first discuss under which circumstances DFM should adopt diverse initial functions and when a constant initial function is more appropriate. By validating our approach on real single-cell datasets, we then show that for DFM(D), even when the initial functions are randomly selected, the generative performance remains stable and consistently surpasses that of CFM (see Table 15). This demonstrates the robustness of DFM with respect to the choice of initial functions. *The detailed experimental results and analysis can be found on Page 31, Lines 1620-1667.*

---

> ### Author Response · Authors · 2025-11-21
>
> **Response to the questions**
>
> ```
> Q1: How does the required time-step ... discretization?
> ```
>
> **Response**:
>
> We thank the reviewer for the comment. We provide a comprehensive discussion from two aspects:
>
> (1)Training with linear interpolation paths:
>
> In general, DFM tends to produce straighter transport trajectories than CFM due to its stronger representation capacity, especially in scenarios involving trajectory intersections or distributional heterogeneity. This leads to more efficient inference, as DFM can achieve accurate generative results with fewer Euler discretization steps (or NFEs) than CFM.
>
> In Appendix E, we include comparisons between DFM and CFM across multiple examples, evaluating training and inference time, trajectory curvature, and generation quality under different discretization levels (or NFEs). The results consistently show that the paths generated by our DFM have lower curvature than CFM, enabling higher-quality sample generation in fewer steps. (see Table 14, Figure 5, Figure 6, Table 13, and Table 4). *These analyses and results are presented in Page 27–28, Lines 1435–1486 of the revised manuscript.*
>
> (2)Nonlinear interpolation paths:
>
> For many scientific problems, such as single-cell trajectory inference (see Section 6.3 of the main text), an essential requirement is that transport trajectories remain on the underlying data manifold to ensure biological interpretability. In such cases, overly straight trajectories or using too few discretization steps may cause deviations from the data manifold, leading to substantial degradation in inference accuracy. *We provide a detailed discussion and illustrations of this issue on Page 27–28, Lines 1487–1498.* Under this setting, inference efficiency and the number of discretization steps are no longer the primary concerns.
>
> ```
> Q2: Many practical generative systems rely on guidance ... how guidance is applied?
> ```
>
> **Response**:
>
> We sincerely thank the reviewer for raising this question. In fact, our DFM framework can naturally integrate with these guidance-based mechanisms. Moreover, the way guidance is incorporated into the vector field with the delay term is analogous to its implementation in conventional ODE vector fields. We provide the following analysis to illustrate this point in detail.
>
> We note that classifier-free guidance (CFG), originally developed for diffusion models, has recently been incorporated into the Flow Matching framework [1] by jointly training an unconditional vector field $u\_t(x)$ and a conditional vector field $u\_t(x|y)$, and then constructing the guided vector field. A similar strategy can be adopted to introduce CFG into our DFM framework. Specifically, for the probability path conditioned on $y$, we have:
> $$p(x,t|\psi,y)=\int p(x,t|z,\psi,y)q(z|y)dz,$$
> $$p(x,t;x\_{\tau},t-\tau|\psi,y)=\int p(x,t;x\_{\tau},t-\tau|z,\psi,y)q(z|y)dz,$$
> We can define the vector field conditioned on $y$ by marginalizing over the conditional vector field $u(t,x,x\_{\tau}|z,y)$ via
> $$u(t,x,x\_{\tau}|y)=\int u(t,x,x\_{\tau}|z,y)\frac{p(x,t;x\_{\tau},t-\tau|z,\psi,y)q(z|y)}{p(x,t;x\_{\tau},t-\tau|\psi,y)}dz,$$
> which can be proved to generate the probability path $p(x,t|\psi,y)$ (The proof follows exactly the same procedure as that of Proposition 4.1 in the main text, with the following substitutions: $p(x,t|\psi)\to p(x,t|\psi,y), p(x,t;x\_{\tau},t-\tau|\psi)\to p(x,t;x\_{\tau},t-\tau|\psi,y), p(x,t|z,\psi)\to p(x,t|z,\psi,y), p(x,t;x\_{\tau},t-\tau|z,\psi)\to p(x,t;x\_{\tau},t-\tau|z,\psi,y), q(z)\to q(z|y), u(t,x,x\_{\tau})\to u(t,x,x\_{\tau}|y), u(t,x,x\_{\tau}|z)\to u(t,x,x\_{\tau}|z,y)$). Then, we can train the parameterized vector field conditioned on $y$ by minimizing the following objective:
> $$\mathcal{L}(\theta)=\mathbb{E}\_{t, q(y), q(z|y), q^{\circ}(\psi), p(x,t;x\_{\tau},t-\tau|z,\psi,y)} ||v(t,x, x\_{\tau}|y;\theta) - u(t,x,x\_{\tau}|z,y)||^2,$$
> which can be proved to be equivalent to minimizing the following objective:
> $$\tilde{\mathcal{L}}(\theta)=\mathbb{E}\_{t, q(y), q^{\circ}(\psi), p(x,t;x\_{\tau},t-\tau|\psi,y)} ||v(t,x, x\_{\tau}|y;\theta) - u(t,x,x\_{\tau}|y)||^2.$$
> The proof is identical to that of Proposition 4.2 in the main text, except that the vector fields and distributions are replaced by their counterparts conditioned on $y$.
>
> Based on this, we can train the guided flows of the delay vector field using the loss function $\mathcal{L}(\theta)$ , where the conditioning variable $y$ is replaced by $y=\emptyset$ with probability $0<P<1$, corresponding to the null-conditioning case (similar to [1]). In this way, the unconditional and conditional vector fields with the delay term are trained jointly, and interpolating between them yields the guided flows associated with the delay vector field, thereby enabling the use of CFG within the DFM framework.
>
>
> [1]. Zheng, Q., Le, M., Shaul, N., Lipman, Y., Grover, A., and Chen, R. T. Guided flows for generative modeling and decision making. arXiv preprint, 2023

---

### Official Review · Reviewer_KnhQ · 2025-11-08

**Soundness:** 3
**Presentation:** 2
**Contribution:** 3
**Rating:** 8
**Confidence:** 4

**Summary:**

The paper proposed to learn vector field that uses dynamics with memory (a dependence on a past state) to overcome two core limitations of standard flow matching: difficulty to handle trajectory crossings and difficulty with heterogeneous source/target supports. This “delay” preserves sample identity at intersections, enabling mappings like Gaussian point-wise negation and keypoint-consistent image translations that ODE flows struggle with. Both theoretical support and experimental validations are provided for the claims.

**Strengths:**

1.	The proposed approach is novel and and enables learning more flexible vector fields using flow matching than is possible with existing approaches.
2.	Experimental Observations are also interesting and clearly show the benefits of DFM.

**Weaknesses:**

1.	The writing could be made clearer. The paper is dense in terms of information content. A simple and concrete running example to illustrate each of the section’s content would be helpful for following the paper.

2.	The claim that the described limitations of ODE based FM are inherent and unavaoidable is too strong. With the right choice of interpolant and coupling, those limitations could be addressed as in [1,2]

[1] Shrestha, Sagar, and Xiao Fu. "Diversified Flow Matching with Translation Identifiability." Forty-second International Conference on Machine Learning.

[2] Albergo, Michael S., and Eric Vanden-Eijnden. "Building normalizing flows with stochastic interpolants." arXiv preprint arXiv:2209.15571 (2022).
However, ODE based FMs do require special design for different types of constraints and might not be as universal as the proposed DFM. The authors are recommended to include such discussions.

**Questions:**

1.	Could the authors discuss realizations of initial functions and the path gamma for the two examples in Figure 1 and (intuitive) explanations of how the learnt vector field circumvent the described issues?

2.	Could the authors provide algorithms to summarize different DFM methods in the appendix ?

3.	Would it be more precise to say that the universal approximation theorem states the existence of delayed vector field that can approximate any mapping, rather than DFM can approximate any mapping? Because the flow matching objective for identifying those vector fields might still need to be modified (with different choice of coupling and gamma) for different cases ?

---

> ### Author Response · Authors · 2025-11-21
>
> We sincerely appreciate your positive feedback, valuable comments, and constructive suggestions. We hope that the following responses, together with the revisions made to the manuscript, adequately address your concerns and further improve the clarity and rigor of our work. We kindly remind the reviewer that all modifications and additions in the revised manuscript are summarized in the “General Response” for convenient reference.
>
> **Response to the weaknesses and questions**
>
> ```
> W1: The writing could be made clearer. The paper is dense in terms of information content. A simple and concrete running example to illustrate each of the section’s content would be helpful for following the paper.
> Q1: Could the authors discuss realizations of initial functions and the path gamma for the two examples in Figure 1 and explanations of how the learnt vector field circumvent the described issues?
> ```
>
> **Response**:
>
> Many thanks for your valuable suggestions. To improve the clarity and readability of the manuscript, and to better illustrate the key components of the DFM pipeline, we have added two simple running examples that explicitly illustrate how DFM operates. These two examples respectively present the problems of trajectory intersections and distribution heterogeneity during the transport process (*see Figure 2*). In Section 6.1, we provide a detailed explanation of the training procedure of DFM on these examples, including the choice of the coupling strategies, the design of the initial functions, and the construction of the interpolation paths. Furthermore, for each example, we not only present the resulting inference results but also construct specific DDEs that offer intuitive explanations for why the vector field with a delay term naturally resolves the problem of trajectory intersections and distributional heterogeneity.
> (*Please see Page 8, Lines 406–424 of the revised manuscript.*)
>
> ```
> W2: The claim that the described limitations of ODE based FM are inherent and unavaoidable is too strong. With the right choice of interpolant and coupling, those limitations could be addressed.
> ```
>
> **Response**:
>
> Many thanks for your comments. We agree that our earlier claim was too strong, and we acknowledge that several ODE-based FM extensions have partially addressed these limitations. To ensure a more rigorous and precise presentation, we revise the discussion of ODE-based FM limitations to clarify that these conclusions apply specifically to the classical FM framework (*see Page 4, Lines 186–188*).
>
> In addition, we add a dedicated Related Works section (Section 5) to systematically review the existing ODE-based FM variants that address the problem of trajectory intersections and distribution heterogeneity. We further analyze the remaining limitations of these variants, which helps highlight the conceptual advantages of the proposed DFM framework (*see Page 7–8, Lines 374–397*).
>
> Moreover, we conduct experiments comparing two representative ODE-based FM extensions with DFM and provide intuitive demonstrations of the scenarios in which these methods still struggle relative to DFM. *These results are summarized on Page 32–33, Lines 1724–1756 of the revised manuscript.*
>
> ```
> Q2: Could the authors provide algorithms to summarize different DFM methods in the appendix?
> ```
>
> **Response**:
>
> Many thanks for your helpful suggestion. To provide a clear summary of the different DFM methods, we add pseudocode for both DFM(C) and DFM(D) in Appendix C. *Please refer to Page 19, Lines 1009–1013 of the revised manuscript.*
>
>
> ```
> Q3: Would it be more precise to say that the universal approximation theorem ... for different cases ?
> ```
>
> **Response**:
>
> We sincerely appreciate your careful and insightful comment. We realize that our previous statement of the universal approximation property may cause some confusion or ambiguity. We have now refined the theorem to provide a more rigorous and accurate formulation. Specifically, instead of claiming that “DFM can approximate any mapping”, the revised statement clarifies that it is the flow map induced by an appropriate vector field with a single delay term that possesses the universal approximation property for any continuous transport map between distributions. The revised theorem is stated as follows.
>
> (Universal approximating capability of DDE's flow map). For any given $\epsilon>0$ and continuous transport map $F: \mathbb{R}^d\to \mathbb{R}^d$, which push-forward the source distribution $q_0$ to the target distribution $q_1$, if there exists a neural network $f(x;\theta)$ such that $||f(x;\theta)-[F(x)-x]||<\epsilon$, for all $x\in \mathbb{R}^d$, then we can construct a vector field with a single delay term $v(t,x,x_{\tau};\theta)$, where the corresponding flow map $G(x;\theta)$, under the constant initial function condition, satisfies $||G(x;\theta)-F(x)||<\epsilon$, for all $x\in \mathbb{R}^d$.
>
> *Please refer to Page 7, Lines 331–336*

---

### Author Response · Authors · 2025-11-21
**General Response**

We express our sincere gratitude to all the reviewers for their professional evaluation of our article, which has greatly contributed to enhancing the quality of our manuscript. In response to the reviewers’ valuable comments and questions, we have made substantial additions and revisions to the manuscript. All modifications are highlighted in color in the updated version. The main changes are summarized as follows:

**1. Revision of the theorem:**

We have refined the statement of the universal approximation theorem under the DFM framework to make it more rigorous and accurate.

*Please refer to Page 7, Lines 331–336 of the revised manuscript*.

**2. Addition of illustrative examples:**

To better assist readers in understanding the overall process of DFM, we add two simple illustrative examples to specifically demonstrate it. These two examples respectively present the problems of trajectory intersections and distribution heterogeneity during the transport process (see Figure 2). In Section 6.1, we provide a detailed explanation of the training procedure of DFM on these examples, including the choice of the coupling strategies, the design of the initial functions, and the construction of the interpolation paths. Moreover, for both examples, we not only present the inference results but also construct specific DDEs to intuitively illustrate why the delay vector field naturally resolves trajectory crossing and distribution heterogeneity.

*Please see Page 8, Lines 406–424 of the revised manuscript.*


**3. Pseudocode:**

To summarize different DFM methods, we add pseudocode for both DFM(C) and DFM(D) in Appendix C.

*Please see Page 19, Lines 1009–1013 of the revised manuscript.*

**4. Summary of related works:**

We add a dedicated Related Works section (Section 5) to summarize existing ODE-based generative methods that address the problem of trajectory intersections and distribution heterogeneity. We further provide a detailed comparison between these variants and our DDE–based generative framework, highlighting the remaining limitations of ODE-based methods and the conceptual advantages of DFM.

*Please refer to Page 7–8, Lines 374–397 of the revised manuscript.*

**5. Comparison with additional baseline methods:**

Among the various ODE-based Flow Matching variants discussed in the Related Works section, we select two representative methods—Augmented Bridge Matching (AugBM) and Switched Flow Matching (SFM)—for direct comparison. AugBM augments the vector field with initial-point information to allow trajectory crossing, whereas SFM introduces a switching signal to enable precise transport between heterogeneous distributions. Through experiments on illustrative examples, we show that these baseline methods still suffer from inherent limitations, while our DFM framework is able to simultaneously overcome these issues.

*Details can be found on Page 32–33, Lines 1724–1756 of the revised manuscript.*

**6. Comprehensive ablation studies:**

We add a systematic set of ablation experiments in Appendix F, covering:

(F.1) a sensitivity analysis of the time delay $\tau$ and guidelines for its selection;

(F.2) a sensitivity analysis of the initial functions and corresponding selection strategies;

(F.3) an ablation study on how clustering quality affects the generative performance of DFM;

(F.4) additional analysis of the choices related to other DFM components, including the construction of interpolation paths, the employed coupling strategies, and the rationale for selecting specific strategies in different tasks.

These ablations collectively demonstrate that the proposed DFM framework exhibits strong robustness with respect to all of its components.

*Please refer to Page 28–32, Lines 1500–1723 of the revised manuscript.*

**7. Training and inference time analysis:**

In Appendix E, we provide a detailed comparison of training and inference time between DFM and CFM. Our experiments show that although DFM incurs slightly higher training cost than CFM, it typically produces significantly *straighter* transport paths—particularly in scenarios involving trajectory intersections or distribution heterogeneity. This leads to substantially shorter inference times while maintaining high generative quality.

We include intuitive comparisons of DFM and CFM across several examples, evaluating training and inference time, trajectory curvature, and generation quality under varying discretization steps (or NFE) (See Table 14, Figure 5, Figure 6, Table 13, and Table 4).

*Please refer to Page 27–28, Lines 1435–1498 of the revised manuscript.*



Finally, we thank all the reviewers again for your valuable and insightful comments. We hope that our general response as well as the individual responses for each reviewer adequately address the reviewers’ concerns.

---

### Author Response · Authors · 2025-12-02
**Overview of Authors’ Rebuttal for the Area Chair**

Dear Area Chair,

We sincerely appreciate your time and efforts in reviewing our submission. In this work, we introduce Delay Flow Matching (DFM), a new generative modeling framework based on delay differential equations. By incorporating delay terms into the vector field, DFM fundamentally overcomes several limitations inherent to existing ODE-based generative frameworks—for example, the prohibition of trajectory crossing during transport, the emergence of singularities when the source and target distributions exhibit strong heterogeneity, and the inability to model transport dynamics that intrinsically involve delays. We further provide a rigorous proof that the proposed DFM framework possesses universal approximation capability for continuous transport maps between arbitrary distributions. This establishes that DFM has a strictly stronger representational capacity than conventional ODE-based generative models, enabling it to capture a broader and more expressive class of distributional transformations.

To the best of our understanding, we have carefully and comprehensively addressed all concerns raised by the reviewers. Below we provide a brief summary of the main issues highlighted during the review process and our corresponding responses.

**1. Training and Inference Time**

Reviewer vrRC (Q1), Reviewer qSLq (W4), Reviewer 7LHR (W2 & Q1), and Reviewer LNpe (W1 & Q1) asked whether our DFM framework requires longer training or inference time, or finer discretization, compared to ODE-based CFM. Through both empirical evidence and analysis, we show that DFM not only achieves better generative quality than CFM but can also generate samples more efficiently.

During training, DFM only requires sampling one additional delay term per step, without introducing any other computational overhead. As a result, its average per-step training time is only marginally higher than that of CFM (see Table 14). During inference, our experiments demonstrate that DFM often produces ***straighter*** transport trajectories—particularly in scenarios where trajectory crossing or distribution heterogeneity occurs. Consequently, DFM requires significantly fewer discretization steps and shorter inference time to generate high-quality samples (see Table 14).

In Appendix E, we evaluate and compare DFM and CFM across examples in terms of training and inference time, trajectory curvature, and generation quality under varying discretization steps. The results consistently show that DFM yields lower-curvature trajectories, enabling higher-quality sample generation in fewer steps and ultimately offering both improved efficiency and better performance. *For detailed results and analysis, please refer to Page 27–28, Lines 1435–1498 of the revised manuscript.*

**2. Systematic Ablation Study**

Reviewer vrRC (Q3) and Reviewer LNpe (Q2) asked about the impact of the choice of the initial function on DFM’s performance. Reviewer qSLq further noted that the previous manuscript lacked a systematic ablation study (W1) and specifically inquired about the effect of the clustering quality (Q3). In response, we added a comprehensive ablation study in Appendix F that thoroughly examines and validates the robustness of all key components of DFM, including the delay term (F.1), the initial functions (F.2), and the clustering quality (F.3). Across all ablation settings, the results consistently demonstrate that the proposed DFM framework is highly robust with respect to each of its components. *For detailed results and analysis, please refer to Page 28–32, Lines 1500–1723 of the revised manuscript.*

**3. Summary of Related Work and Additional Comparisons with Other Baselines**

Reviewer qSLq (W2) and Reviewer 7LHR (W1 & Q2 & Q3) noted that several variants of ODE-based FM have been proposed to alleviate trajectory intersections and distribution heterogeneity, and asked for a more systematic summary and comparison with our method. In response, we added a dedicated **Related Works** section (Section 5) that reviews existing ODE-based approaches addressing these issues. We also provide a detailed comparison between these variants and our DDE-based generative framework, highlighting the remaining limitations of ODE-based methods as well as the conceptual advantages of DFM. *Please refer to Page 7–8, Lines 374–397 of the revised manuscript.*

In Appendix G, we further selected two representative ODE-based methods—one designed to mitigate trajectory intersections and another to address distribution heterogeneity—and conducted direct empirical comparisons against DFM. The results show that these ODE-based variants still suffer from notable limitations in these settings, whereas DFM effectively overcomes both challenges. *For detailed results and analysis, please refer to Page 32–33, Lines 1724–1756 of the revised manuscript.*

---

> ### Author Response · Authors · 2025-12-02
> **Overview of Authors’ Rebuttal for the Area Chair ((Continuation))**
>
> **4. Guidance on Selecting Each Component of DFM**
>
> Reviewer qSLq (W3 & Q1 & Q2) pointed out that the previous manuscript lacked guidance on how to choose the components of our framework and requested clarification on the rationale for using different coupling strategies and interpolation paths across experiments. In Appendix F, we provide practical guidelines for selecting each component of DFM in real applications, including the delay term, the initial functions, the coupling strategy, and the interpolation path. In Appendix F.4, we further analyze and explain why different coupling strategies and interpolation paths are chosen for different tasks, demonstrating the rationale behind these choices and their consistency with prior work.
>
> **5. Performance on Standard Image Generation Tasks**
>
> Reviewer qSLq (Q4) and Reviewer LNpe (W2 & Q3) asked about the performance of DFM on image generation tasks when the initial distribution is an isotropic standard Gaussian. In the main experiments, we adopted a more complex initial distribution—a two-component Gaussian mixture proposed in prior work—to better highlight DFM’s advantages in handling distribution heterogeneity. This setting allows us to more effectively evaluate the generative capability of the model when both the source and target distributions are multimodal, thereby emphasizing DFM’s ability to manage more challenging transport scenarios. Under this setup, DFM demonstrates clear advantages over CFM.
>
> For standard image generation tasks, we additionally conducted experiments on the MNIST dataset using a standard Gaussian prior. The results show that DFM achieves slightly better generation quality than CFM across different delay settings, though the gap is smaller due to the weaker heterogeneity between source and target distributions in this case. Nonetheless, DFM consistently matches or surpasses CFM’s performance across all tested configurations.
>
> **6. Revision of Theorem Statements**
>
> Reviewer KnhQ (W2 & Q3) and Reviewer vrRC (W3) pointed out that the statement of our theorem is a little imprecise. We have revised the formulation of the universal approximation theorem and its corresponding proof using more rigorous mathematical language to eliminate potential confusion or ambiguity (*see Page 7, Lines 331–336*). In addition, we clarified the discussion of the limitations of ODE-based FM to emphasize that these conclusions apply specifically to the classical FM framework (*see Page 4, Lines 186–188*).
>
> **7. Integration with Classifier-Free Guidance**
>
> Reviewer vrRC (W2 & Q2) asked whether DFM can be combined with classifier-free guidance. In our response, we formally show that the DFM framework naturally supports integration with guidance-based mechanisms, and that incorporating guidance into the delay vector field follows an analogous formulation to its implementation in ODE-based vector fields. We also provide a concrete training procedure that enables DFM to learn the conditional vector field based on the conditioning variable $y$.
>
> **8. Illustrative Examples**
>
> As suggested by Reviewer KnhQ (W1 & Q1), to better illustrate the DFM workflow, we have added two simple running examples that explicitly illustrate how DFM operates in Section 6.1. *Please see Page 8, Lines 406–424.*
>
> **9. Pseudocode**
>
> As suggested by Reviewer KnhQ (Q2), to summarize different DFM methods, we add pseudocode for both DFM(C) and DFM(D) in Appendix C. *Please see Page 19, Lines 1009–1013.*
>
> Finally, we would like to reiterate that DFM is a fundamentally new DDE-based generative framework that overcomes many intrinsic limitations of ODE-based approaches. By incorporating delay terms, DFM addresses trajectory-crossing and distribution-heterogeneity issues directly in the original phase space, unlike other FM variants that rely on additional latent variables or specially designed transport processes.
>
> Our goal is to develop a broadly applicable methodology suitable for a wide range of scientific problems—not only image generation, but also tasks such as single-cell trajectory inference, recovery of delay dynamical systems, and other scientific applications that can benefit from generative modeling. We have demonstrated DFM’s strong performance and advantages across these standard tasks.
>
> Moreover, DFM can naturally integrate with many established techniques—such as keypoint-guided optimal transport and classifier-free guidance—highlighting its strong extensibility and compatibility with existing generative modeling practices.
>
> While several reviewers have not yet responded to our rebuttal, we believe that the substantial additional experiments and analyses we have provided effectively address the core concerns shared across their reviews. We would be very grateful if you could consider our detailed responses together with the revised manuscript, and we fully trust your fair and independent reassessment of our work.
>
> Best regards,
>
> The authors of Paper “Delay Flow Matching”

---

### Meta-Review · Area_Chair_WGrt · 2025-12-27

**Summary:**

The paper proposes Delay Flow Matching (DFM), a novel approach that integrates Delay Differential Equations (DDE) into the flow matching framework. The central innovation is a velocity field that depends on past states, enabling the flow paths to intersect—a feature not possible with classical Ordinary Differential Equations (ODE) paths. This introduces a significant degree of freedom for constructing transport maps between source and target distributions. The authors demonstrate the method's theoretical strength by proving its universal approximation and provide experimental evidence of its effectiveness.

Reviewers Feedback
Strengths: Reviewers liked the novel approach, the resulting increased flexibility for the flow matching framework, and the positive outcomes observed in the experimental setup.
Concerns: Key concerns included insufficiently clear writing, a lack of self-containedness, a potentially overstated critique of ODE-FM limitations, and questionable significance specifically in the pure generation setting. The paper was also criticized for omitting simple baselines (e.g., adding conditioning mechanisms to the velocity field) and lacking contextualization with related work (e.g., switched flow matching). Finally, reviewers wondered whether the delay mechanism offers a benefit in a standard image generation task.

The authors addressed the main drawbacks in their rebuttal and edited the manuscript accordingly. Overall it seems this paper passes the bar for ICLR acceptance.

**Reviewer Concerns:**

Please see above.

**Reviewer Scores:**

Please see above.

---

### Decision · Program_Chairs · 2026-01-26

Accept (Poster)